# Density Ratio-Free Doubly Robust Proxy Causal Learning

**Bariscan Bozkurt**[1]  **Houssam Zenati**[1]  **Dimitri Meunier**[1]  **Liyuan Xu**[2]

**Arthur Gretton**[1,3]

[1]Gatsby Computational Neuroscience Unit, University College London,
[2]Secondmind, [3]DeepMind.
{bariscan.bozkurt.23, h.zenati, dimitri.meunier.21}@ucl.ac.uk
liyuan9988@gmail.com  arthur.gretton@gmail.com

## Abstract

We study the problem of causal function estimation in the Proxy Causal Learning (PCL) framework, where confounders are not observed but proxies for the confounders are available. Two main approaches have been proposed: outcome bridge-based and treatment bridge-based methods. In this work, we propose two kernel-based doubly robust estimators that combine the strengths of both approaches, and naturally handle continuous and high-dimensional variables. Our identification strategy builds on a recent density ratio-free method for treatment bridge-based PCL; furthermore, in contrast to previous approaches, it does not require indicator functions or kernel smoothing over the treatment variable. These properties make it especially well-suited for continuous or high-dimensional treatments. By using kernel mean embeddings, we propose the first density-ratio free doubly robust estimators for proxy causal learning, which have closed form solutions and strong uniform consistency guarantees. Our estimators outperform existing methods on PCL benchmarks, including a prior doubly robust method that requires both kernel smoothing and density ratio estimation.

## 1 Introduction and related works

Estimating the effects of interventions-also referred to as *treatments*-on outcomes is a central goal of causal learning. This task is particularly challenging in observational settings, where randomized experiments are not feasible. A major difficulty stems from *confounding variables* that influence both the treatment and the outcome. In practice, the commonly made unconfoundedness (or ignorability) assumption [1]—that there are no unobserved confounders—often fails to hold, as it is unrealistic to expect that all relevant confounders can be accounted for. As a result, one may instead assume that the observed covariates serve as *proxies* for latent unmeasured confounders.

One classical line of work for addressing unobserved confounding is *instrumental variable* (IV) regression, which assumes access to instruments that affect the treatment but are independent of the unobserved confounders [2, 3, 4]. A more recent and promising direction is the Proxy Causal Learning (PCL), which leverages two auxiliary variables: (i) a treatment proxy, denoted by $Z$, which is causally related to the treatment, and (ii) an outcome proxy, denoted by $W$, which is causally related to the outcome. The associated causal graph is illustrated in Figure (1). In this setup, a bidirectional arrow indicates that either causal direction between the two variables is plausible, or that they may share an unobserved common cause. Within this framework, Miao et al. [5] showed that the causal effect can be identified via an *outcome bridge function*, without the need

39th Conference on Neural Information Processing Systems (NeurIPS 2025).

to explicitly recover the latent confounders. This contrasts with approaches such as Kuroki and Pearl [6], Louizos et al. [7], Lee et al. [8], which attempt to estimate the latent confounders directly.

Specifically, Miao et al. [5] demonstrate that an outcome bridge function—a function of the outcome proxy $W$ and the treatment—whose conditional expectation equals the regression function, can be integrated over the distribution of $W$ to recover the average causal effect. Building on this idea, several methods have been proposed to estimate the outcome bridge function and the causal effect, utilizing sieve expansions [10], reproducing kernel Hilbert spaces (RKHS) [11, 12], neural networks [13, 14] and minimax learning [15, 16]. In particular, Mastouri et al. [11] proposed two kernel-based methods for efficiently estimating the outcome bridge function: a two-stage regression approach called *kernel proxy variable* (KPV) and a one-step estimator based on maximum moment restrictions [17] called *proximal maximum moment restriction* (PMMR). Although outcome bridge function-based methods have received widespread attention, an alternative line of work focuses on identifying causal effects using ideas inspired by inverse propensity score (IPS) models [18, 19, 20]. The seminal work by Cui et al. [21] introduced a complementary identification strategy based on a *treatment bridge function*—a function of the treatment proxy $Z$ and the treatment. This approach, however, is limited to binary treatments and relies on an indicator function over the treatment of interest in the identification formula. Moreover, it requires density

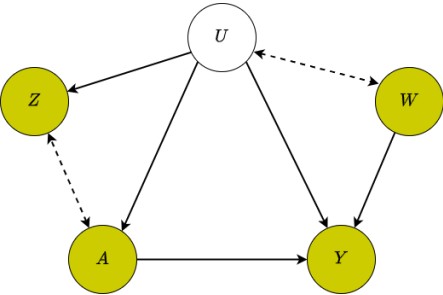

Figure 1: An illustrative causal graph for proxy causal learning (PCL), consistent with Assumption (2.2) [5]. Observed variables are shown as yellow nodes: $A$ denotes the treatment, $Y$ denotes the outcome, $Z$ denotes the treatment proxy, and $W$ denotes the outcome proxy. The unobserved confounder $U$ is depicted as a white node. Dotted bi-directional arrows indicate potential bidirectional causality (ambiguous directionality) or the existence of a shared latent ancestor between variables. For other causal models that satisfy this assumption, see Table A.1 in [9]

ratio estimation to recover the treatment bridge function. Extensions to continuous treatments have been proposed by Wu et al. [22] and Deaner [10]. In particular, Wu et al. [22] generalized the binary treatment framework of Cui et al. [21] by replacing the indicator function with kernel smoothing. However, density ratio estimation remains a significant challenge, particularly in high-dimensional settings. To overcome the limitations posed by indicator functions, kernel smoothing, and explicit density ratio estimation, Bozkurt et al. [23] introduced a density ratio-free identification strategy based on a novel formulation of the treatment bridge function called *kernel alternative proxy* (KAP). Their method simplifies the least-squares objective to bypass the explicit density ratio estimation, following ideas similar to those in Kanamori et al. [24], and it also avoids both kernel smoothing and indicator function. As a result, it is more scalable for settings with continuous and high-dimensional treatments.

In practice, the treatment- and outcome- bridge approaches each have different and complementary strengths. Bozkurt et al. [23] illustrate in experiments that the relative performance of each method depends on the informativeness of the proxies [see Section (13) in 23]. Specifically, when the outcome proxy $W$ is more informative about the unobserved confounder $U$ than the treatment proxy $Z$, their treatment bridge-based method outperforms outcome bridge-based alternatives. Conversely, when $Z$ is more informative about $U$ than $W$, outcome bridge-based methods such as those of Mastouri et al. [11] and Singh [12] yield better performance. However, in practice, it is often difficult to know in advance which scenario applies. This motivates the development of a *doubly robust* (DR) approach for the PCL setting that combines the strengths of both classes of methods in a way that the resulting approach will remain robust, even if one of the methods fails to identify the causal effect.

DR methods are well established in fully observed causal settings due to their appealing property of consistency if either the outcome model or the propensity score model is correctly specified [25, 26]. Such estimators have been generalized to a wide range of functional estimation tasks [27, 28], and are often grounded in the theory of efficient influence functions [29, 30, 31]. These techniques allow for bias reduction and valid inference even in high-dimensional or nonparametric settings [32, 33].

In the PCL setting, the first doubly robust (DR) approach was introduced by Cui et al. [21], who combined their novel treatment bridge function-based method with the earlier outcome bridge

function-based method of Miao et al. [5]. Their method is built on the theory of efficient influence functions and achieves the semiparametric efficiency bound. However, as noted earlier, it is limited to the binary treatment case. An extension to continuous treatments was proposed by Wu et al. [22], who replaced the indicator function over the treatment of interest with a kernel smoothing technique, resulting in a nonparametric DR estimation procedure. Nonetheless, learning the treatment bridge component of their DR estimator still requires explicit density ratio estimation. Additionally, the use of kernel smoothing can introduce further difficulties when the treatment is high-dimensional.

In this work, we develop two novel doubly robust estimators for PCL. Specifically: (i) our first DR estimator integrates the two-stage regression method KPV of Mastouri et al. [11] with the treatment bridge function formulation KAP of [23], and (ii) our second DR estimator leverages the maximum moment restriction-based algorithm PMMR in Mastouri et al. [11] with the same treatment bridge strategy KAP. Both estimators retain consistency as long as either the outcome bridge function or the treatment bridge function is correctly specified, thus achieving the doubly robust property.

Our key contributions are: (i) We propose two novel doubly robust algorithms for causal effect estimation in the PCL setting that circumvent the need of explicit density ratio estimation; (ii) We leverage conditional mean embeddings (CMEs) to derive simple, closed-form estimators using matrix and vector operations—scalable to continuous and high-dimensional treatments; (iii) We establish uniform consistency of our proposed estimators, which is stronger than the pointwise convergence typical in doubly robust causal learning; and (iv) We empirically demonstrate that our methods outperform existing PCL baselines in challenging scenarios.

The remainder of the paper is organized as follows: Section (2) introduces the problem setup and our doubly robust identification result. Section (3) presents our nonparametric estimation algorithms. Section (4) outlines the consistency results for our proposed methods. Numerical experiments are reported in Section (5), and we conclude in Section (6). Our implementation code is available on GitHub[1].

## 2    Problem setting and identification

We consider the problem of estimating causal effects, defined as counterfactual outcomes under hypothetical interventions. Let $A \in \mathcal{A}$ denote the treatment and $Y \in \mathcal{Y} \subset \mathbb{R}$ the observed outcome. An unobserved confounder $U \in \mathcal{U}$ affects both $A$ and $Y$. Our goal is to estimate the *dose-response curve*, as formalized in the following definition.

**Definition 2.1.** *The dose-response is defined as $\theta_{ATE}(a) = \mathbb{E}[\mathbb{E}[Y \mid A = a, U]]$, representing the counterfactual mean outcome if the entire population received treatment $a$. The subscript ATE signifies that its semiparametric counterpart is the average treatment effect (ATE).*

The key difficulty in estimating dose-response stems from the fact that the confounding variable $U$ is unobserved in most of the applications. To address this issue, proxy causal learning setup assumes the availability of two proxy variables: $Z$, which serves as a proxy for the treatment, and $W$, which acts as a proxy for the outcome. The underlying causal graph is depicted in Figure (1). This graphical model admits the following conditional independence assumptions which we leverage throughout:

**Assumption 2.2.** *(Conditional Independencies). The following conditional independence statements are implied by the causal graph illustrated in Figure (1): i-) $Y \perp Z|U, A$ (Conditional Independence for Y), ii-) $W \perp Z|U, A$ and $W \perp A|U$ (Conditional Independence for W).*

In the following subsections, we review two main approaches to estimate the dose-response curve in PCL setting: (i) outcome bridge function methods [5, 11, 12, 13, 34], and (ii) treatment bridge function methods [10, 21, 22, 23, 35]. These methods rely on completeness Assumptions (2.3) and (2.5) that formalize the requirement that the proxy variable be sufficiently informative about the unobserved confounders, highlighting the importance of collecting rich sets of proxy variables in observational studies to mitigate unobserved confounding. While completeness is generally not testable (even when all relevant variables are observed), it is known to hold for a wide range of semiparametric and nonparametric models [5]. For an extended discussion on completeness within the PCL framework, we refer the reader to Miao et al. [5, Section (S.2)].

---

[1]`https://github.com/BariscanBozkurt/Doubly-Robust-Kernel-Proxy-Variable-Algorithm`

## 2.1 Outcome bridge function-based identification

To identify the dose-response, Miao et al. [5], Wang Miao and Tchetgen [34] show that the causal effect is nonparametrically identifiable under the model in Figure (1), given the following completeness assumption [13, Assumption 2]:

**Assumption 2.3.** *(Completeness Assumption). For any square-integrable function $\ell : \mathcal{U} \to \mathbb{R}$ and for any $a \in \mathcal{A}$, $\mathbb{E}[\ell(U) \mid Z, A = a] = 0$ almost surely if and only if $\ell(U) = 0$ almost surely.*

The completeness condition ensures that the proxy variable $Z$ exhibits sufficient variation relative to the unobserved confounder $U$, enabling the identification of the treatment effect.

The theorem below identifies $\theta_{\text{ATE}}(a)$ via an outcome bridge function.

**Theorem 2.4** (Causal Identification with Outcome Bridge Function [5, 34]). *Let Assumptions (2.2) and (2.3) hold. Furthermore, suppose that there exists an outcome bridge function $h_0(w, a)$ satisfying*

$$\mathbb{E}[Y \mid Z, A] = \int h_0(w, A)p(w \mid Z, A)dw. \tag{1}$$

*Then, the dose-response can be identified by $\theta_{ATE}(a) = \mathbb{E}[h_0(W, a)]$.*

The proof of Theorem (2.4) can be found in [34, Proposition (3.1)], [13, Corollary (1)].

## 2.2 Treatment bridge function-based identification

Cui et al. [21] and Deaner [10] proposed an alternative identification results using a treatment bridge function, but its learning requires density ratio estimation. Therefore, we instead follow Bozkurt et al. [23], whose method avoids this issue as we describe it in Section (3) and forms the basis of our doubly robust estimator. The structure of this treatment bridge function-based identification is analogous to inverse propensity weighting principle [19, 20]. We adopt the following completeness assumptions to identify the dose-response via the treatment bridge function [23, Assumption 3.3]:

**Assumption 2.5.** *(Completeness Assumption). For any square integrable function $\ell : \mathcal{U} \to \mathbb{R}$, for all $a \in \mathcal{A}$, $\mathbb{E}[\ell(U) \mid W, A = a] = 0$ $p(W)$ almost surely if and only if $\ell(U) = 0$ $p(U) - a.e.$*

The completeness condition ensures that the proxy variable $W$ exhibits enough variation in relation to the unobserved confounder $U$, enabling the identification of treatment effect.

**Theorem 2.6** (Causal identification with treatment bridge function [23]). *Let Assumptions (2.2) and (2.5). Furthermore, suppose that there exists a treatment bridge function $\varphi_0(z, a)$ such that*

$$\mathbb{E}\left[\varphi_0(Z, a) \mid W, A = a\right] = \frac{p(W)p(a)}{p(W, a)}, \quad \forall a \in \mathcal{A}, \tag{2}$$

*then the dose-response can be identified by $\theta_{ATE}(a) = \mathbb{E}\left[Y\varphi_0(Z, a) \mid A = a\right]$.*

Theorem (2.6) is proved in [23, Theorem 3.4]. The existence of the treatment bridge function $\varphi_0$ relies on Assumption (2.3), along with mild regularity and integrability conditions. In particular, note that Assumption (2.3) underpins the identification of the causal effect via the outcome bridge function. On the other hand, Assumption (2.5) ensures the existence of $h_0$ and identification via the treatment bridge $\varphi_0$. Thus, these completeness assumptions serve distinct yet complementary roles. See Supplementary Material (S.M.) ( D) for further discussion on the existence of the bridge functions.

## 2.3 Doubly robust proximal identification

With two distinct identification approaches at hand, we now present a doubly robust identification result that combines outcome bridge function and treatment bridge function-based identification.

**Theorem 2.7** (Doubly robust causal identification). *The dose-response can be identified with*

$$\theta_{ATE}^{(DR)}(a; h, \varphi) \mid_{(h=h_0, \varphi=\varphi_0)} = \mathbb{E}[\varphi_0(Z, a)(Y - h_0(W, a)) \mid A = a] + \mathbb{E}[h_0(W, a)], \tag{3}$$

*where $h_0$ and $\varphi_0$ are the outcome and treatment bridge functions that satisfy Equations (1) and (2), respectively. Furthermore, $\theta_{ATE}^{(DR)}(a; h, \varphi)$ admits double robustness such that $\theta_{ATE}^{(DR)}$ identifies the dose-response curve if **either** $h$ solves Equation (1) **or** $\varphi$ solves Equation (2)—but not necessarily both.*

We prove Theorem (2.7) in S.M. (A) and relate our estimator to semiparametric efficiency theory in S.M. (B).

**Remark 2.8.** *In S.M. (B), we provide a background in semiparametric efficiency theory and efficient influence functions (EIF) [29, 30, 28], and we derive the EIF of the dose-response in the discrete treatment case. Under the mild Assumption (B.3), Theorem (B.4) shows that the EIF is $\psi_{ATE}(\mathcal{O}; a) = \varphi_0(Z, A)(Y - h_0(W, A))\frac{1}{p(A)}\mathbf{1}[A = a] + h_0(W, a) - \theta_{ATE}(a)$, where*

$\mathcal{O} = (Y, Z, W, A)$, *and $\mathbf{1}[\cdot]$ is the indicator function defined as* $\mathbf{1}[a_i = a_j] = \left\{ \begin{array}{ll} 1 & a_i = a_j, \\ 0 & otherwise. \end{array} \right.$

*Since the expectation of the influence function is zero, we note that:*

$$0 = \mathbb{E}[\psi_{ATE}(\mathcal{O}; a)] = \mathbb{E}[\varphi_0(Z, a)(Y - h_0(W, a)) \mid A = a] + \mathbb{E}[h(W, a)] - \theta_{ATE}(a),$$

*which implies the identification $\theta_{ATE}(a) = \mathbb{E}[\varphi_0(Z, a)(Y - h_0(W, a)) \mid A = a] + \mathbb{E}[h_0(W, a)]$. While this EIF-based identification is derived for discrete treatments, Theorem (2.7) shows the same form holds for continuous treatments and preserves double robustness.*

**Remark 2.9.** *Our identification result in Theorem (2.7) differs from the doubly robust identification in Cui et al. [21]. They identify the dose-response as $\mathbb{E}[q_0(Z, A)(Y - h_0(W, A))\mathbf{1}[A = a]] + \mathbb{E}[h_0(W, a)]$, where $q_0$ is the treatment bridge function solving the Fredholm integral equation $\mathbb{E}[q_0(Z, a) \mid W, A = a] = 1/p(A = a \mid W) \ \forall a \in \mathcal{A}$, and $h_0$ solves Equation (1). Unlike our formulation based on Equation (2), their identification uses the indicator function $\mathbf{1}[A = a]$. Wu et al. [22] approximate this indicator with kernel smoothing in estimation. While both methods use joint expectations over $(Y, Z, W, A)$, ours relies on conditional expectation under the distribution $p(Y, Z, W \mid A = a)$. This structural difference enables applicability of our methods to high-dimensional treatments, where the kernel-smoothing-based DR approach fails.*

## 3 Nonparametric kernel methods

We now present a nonparametric estimation algorithm for approximating the function in Equation (3) using RKHS theory and conditional mean embeddings. The next subsection reviews the necessary RKHS background, followed by our proposed algorithms' derivations.

### 3.1 Reproducing kernel Hilbert spaces

In this section, we briefly review reproducing kernel Hilbert spaces (RKHSs), as concepts such as kernels and mean embeddings are used throughout our algorithm derivations and consistency analyses. For each space $\mathcal{F} \in \{\mathcal{A}, \mathcal{W}, \mathcal{Z}\}$, we denote the associated positive semi-definite kernel by $k_{\mathcal{F}}(\cdot, \cdot) : \mathcal{F} \times \mathcal{F} \to \mathbb{R}$, which induces the RKHS $\mathcal{H}_{\mathcal{F}} \subset \{\ell : \mathcal{F} \to \mathbb{R}\}$. We denote the corresponding canonical feature map by $\phi_{\mathcal{F}}(f) = k_{\mathcal{F}}(\cdot, f) \in \mathcal{H}_{\mathcal{F}}$. The inner product and norm in the RKHS $\mathcal{H}_{\mathcal{F}}$ are denoted by $\langle \cdot, \cdot \rangle_{\mathcal{H}_{\mathcal{F}}}$ and $\|\cdot\|_{\mathcal{H}_{\mathcal{F}}}$, respectively. The tensor product space is denoted by $\mathcal{H}_{\mathcal{F}} \otimes \mathcal{H}_{\mathcal{G}}$, and for convenience, we use the shorthand notation $\mathcal{H}_{\mathcal{F}\mathcal{G}}$. This space is isometrically isomorphic to the Hilbert space of Hilbert–Schmidt operators from $\mathcal{H}_{\mathcal{G}}$ to $\mathcal{H}_{\mathcal{F}}$ [36], denoted by $S_2(\mathcal{H}_{\mathcal{G}}, \mathcal{H}_{\mathcal{F}})$. Analogously, we denote the tensor product feature map $\phi_{\mathcal{F}}(f) \otimes \phi_{\mathcal{G}}(g) \in \mathcal{H}_{\mathcal{F}} \otimes \mathcal{H}_{\mathcal{G}}$ by $\phi_{\mathcal{F}\mathcal{G}}(f, g)$. Throughout the paper, we impose the following assumption on the domains and the corresponding kernels:

**Assumption 3.1.** *We assume that (i) each $\mathcal{F} \in \{\mathcal{A}, \mathcal{W}, \mathcal{Z}\}$ is a Polish space; (ii) $k_{\mathcal{F}}(f, .)$, is continuous and bounded by $\kappa$, i.e., $\sup_{f \in \mathcal{F}} \|k_{\mathcal{F}}(f, .)\|_{\mathcal{H}_{\mathcal{F}}} \leq \kappa$, for almost every $f \in \mathcal{F}$.*

For a given distribution $p(f)$ on $\mathcal{F}$ and a kernel $k_{\mathcal{F}}$ such that $\mathbb{E}[k_{\mathcal{F}}(F, F)] < \infty$, the kernel mean embedding of $p(F)$ is defined as $\mu_F = \int_{\mathcal{F}} k_{\mathcal{F}}(\cdot, f)p(f)df \in \mathcal{H}_{\mathcal{F}}$ [37, 38]. Furthermore, for a given conditional distribution $p(F|g)$ for each $g \in \mathcal{G}$, the conditional mean embedding (CME) of $p(F|g)$ is defined as the operator $\mu_{F|G}(g) = \int_{\mathcal{F}} k_{\mathcal{F}}(\cdot, f)p(f|g)df \in \mathcal{H}_{\mathcal{F}}$ [39, 40, 41, 42, 43].

### 3.2 Doubly robust algorithm for dose-response curve estimation

We note that the doubly robust function $\theta_{\text{ATE}}^{(\text{DR})}$ consists of three components:

$$\theta_{\text{ATE}}^{(\text{DR})}(a) = \mathbb{E}[Y\varphi_0(Z, a) \mid A = a] - \mathbb{E}[\varphi_0(Z, a)h_0(W, a) \mid A = a] + \mathbb{E}[h_0(W, a)].$$

Let $\mathcal{D} = \{y_i, z_i, w_i, a_i\}_{i=1}^{t}$ be i.i.d. samples from the distribution $p(Y, Z, W, A)$ that are used to estimate $\theta_{\text{ATE}}^{(DR)}$. We develop two doubly robust algorithms for estimating the dose-response curve: (i)

combining the two-stage regression method KPV from Mastouri et al. [11] with the density ratio-free treatment bridge approach KAP from Bozkurt et al. [23], and (ii) combining the maximum moment restriction method PMMR from Mastouri et al. [11] with the same KAP strategy. Following Mastouri et al. [11] and Bozkurt et al. [23], we assume $h_0 \in \mathcal{H}_{\mathcal{W}} \otimes \mathcal{H}_{\mathcal{A}}$ and $\varphi_0 \in \mathcal{H}_{\mathcal{Z}} \otimes \mathcal{H}_{\mathcal{A}}$.

Two RKHS-based methods for estimating $h_0$ are Kernel Proxy Variable (KPV) and Proxy Maximum Moment Restriction (PMMR), both proposed by Mastouri et al. [11]. KPV estimates $\mathbb{E}[h(W, a)]$ in three steps: (i) Given samples $\{\bar{w}_i, \bar{z}_i, \bar{a}_i\}_{i=1}^{n_h} \subset \mathcal{D}$, it estimates $\mu_{W|Z,A}(z, a) = \mathbb{E}[\phi_{\mathcal{W}}(W) \mid Z = z, A = a]$ via regularized least squares. (ii) With second-stage data $\{\tilde{y}_i, \tilde{z}_i, \tilde{a}_i\}_{i=1}^{m_h} \subset \mathcal{D}$, it optimizes the sample-based counterpart of the loss $\mathcal{L}_{\text{KPV}}(h) = \mathbb{E}\left[(Y - \mathbb{E}[h(W, A) \mid Z, A])^2\right] + \lambda_{h,2}\|h\|_{\mathcal{H}_{\mathcal{WA}}}$ using the representer theorem [44] and the first-stage estimate. (iii) The dose-response at the treatment $a$ is then estimated via the sample mean $\frac{1}{t_h}\sum_{i=1}^{t_h} \hat{h}(\dot{w}_i, a)$ where $\{\dot{w}_i\}_{i=1}^{t_h} \subset \mathcal{D}$ can be considered as the third-stage samples, and $\hat{h}$ denotes the bridge function estimate. Here $n_h$, $m_h$, and $t_h$ denotes the number of samples in each stage. One may either split the $t$ training samples across stages or reuse them. In the original KPV implementation, data is split for stages one and two, and the entire dataset is used in stage three, i.e., $t_h = t$ [11].

PMMR instead uses a one-step estimation via the empirical version of the loss $\mathcal{L}_{\text{PMMR}}(h) = \sup_{\|g\| \leq 1} \mathbb{E}[(Y - h(W, A))g(Z, A)]^2 + \lambda_{\text{MMR}}\|h\|_{\mathcal{H}_{\mathcal{WA}}}$ with $g \in \mathcal{H}_{\mathcal{ZA}}$, which can be solved in closed form using the representer theorem. The entire dataset of $t$ samples is used directly for training. Dose-response estimation mirrors the same procedure as in KPV. Additional details and pseudo-code are provided in S.M. (C.1) and Algorithms(2) and (3).

Bozkurt et al. [23] propose a two-stage regression algorithm in RKHSs, called the *Kernel Alternative Proxy* (KAP), to estimate the treatment bridge function $\varphi_0$, followed by a third-stage regression to approximate the dose-response curve via $\mathbb{E}[Y\hat{\varphi}(Z, a) \mid A = a]$, where $\hat{\varphi}$ approximates $\varphi_0$. A key advantage of this approach is that it avoids explicit density ratio estimation, in contrast to the methods proposed by Wu et al. [22] and Cui et al. [21]. Specifically, KAP simplifies the regularized least squares objective as:

$$
\mathcal{L}_{\text{KAP}}(\varphi) = \mathbb{E}\left[\left(\frac{p(W)p(A)}{p(W, A)} - \mathbb{E}[\varphi(Z, A) \mid W, A]\right)^2\right] + \lambda_{2,\varphi}\|\varphi\|_{\mathcal{H}_{\mathcal{ZA}}}^2
$$

$$
= \mathbb{E}\left[\mathbb{E}[\varphi(Z, A) \mid W, A]^2\right] + \mathbb{E}\left[\frac{p(W)p(A)}{p(W, A)}\mathbb{E}[\varphi(Z, A) \mid W, A]\right] + \lambda_{2,\varphi}\|\varphi\|_{\mathcal{H}_{\mathcal{ZA}}}^2 + \text{const.}
$$

$$
= \mathbb{E}\left[\mathbb{E}[\varphi(Z, A) \mid W, A]^2\right] + \mathbb{E}_W\mathbb{E}_A\left[\mathbb{E}[\varphi(Z, A) \mid W, A]\right] + \lambda_{2,\varphi}\|\varphi\|_{\mathcal{H}_{\mathcal{ZA}}}^2 + \text{const.} \tag{4}
$$

Here, $\mathbb{E}_W\mathbb{E}_A[\cdot]$ denotes the decoupled expectation under $p(W)p(A)$, "const." refers to terms independent of $\varphi$, and $\lambda_{2,\varphi}$ is a regularization parameter. Notably, the objective in Equation (4) involves no density ratio terms. The KAP method proceeds in three stages: (i) Given i.i.d. samples $\{\bar{w}_i, \bar{z}_i, \bar{a}_i\}_{i=1}^{n_\varphi} \subset \mathcal{D}$, the first-stage regression estimates the conditional mean embedding $\mu_{W|Z,A}(z, a) = \mathbb{E}[\phi_{\mathcal{W}}(W) \mid Z = z, A = a]$ via regularized least squares. (ii) Using this estimate to approximate the conditional mean $\mathbb{E}[\varphi(Z, A) \mid W, A] = \langle \varphi, \mu_{W|Z,A}(z, a) \otimes \phi_{\mathcal{A}}(a)\rangle_{\mathcal{H}_{\mathcal{ZA}}}$, the second-stage regression minimizes the empirical counterpart of Equation (4) using second-stage samples $\{\tilde{w}_i, \tilde{a}_i\}_{i=1}^{m_\varphi} \subset \mathcal{D}$. A closed-form solution is derived via the representer theorem [44]. (iii) With the estimate $\hat{\varphi}$, the dose-response is estimated via kernel ridge regression to approximate $\mathbb{E}[Y\hat{\varphi}(Z, a) \mid A = a]$ using third-stage samples $\{\dot{y}_i, \dot{z}_i, \dot{a}_i\}_{i=1}^{t_\varphi} \subset \mathcal{D}$. Here, $n_\varphi$, $m_\varphi$, and $t_\varphi$ denote the sample sizes for the first, second, and third stages, respectively, which may be obtained by splitting or reusing the original $t$ training samples. In the published implementation, data is split for the first and second stages, while all $t$ samples are used in the third stage, i.e., $t_\varphi = t$. Further details are provided in S.M. (C.2), and the algorithm is outlined in Algorithm (4).

As a final step to estimate $\theta_{\text{ATE}}^{(\text{DR})}(a)$, we develop a procedure for estimating $\mathbb{E}[\varphi_0(Z, a)h_0(W, a) \mid A = a]$, completing the doubly robust algorithm when combined with estimators for $\mathbb{E}[Y\varphi_0(Z, a) \mid A = a]$ and $\mathbb{E}[h_0(W, a)]$. Using the properties of the tensor product, we notice that

$$
\mathbb{E}[\varphi_0(Z, a)h_0(W, a) \mid A = a] \approx \mathbb{E}[\hat{\varphi}(Z, a)\hat{h}(W, a) \mid A = a]
$$

$$
= \mathbb{E}\left[\left\langle\hat{\varphi}, \phi_{\mathcal{Z}}(Z) \otimes \phi_{\mathcal{A}}(a)\right\rangle_{\mathcal{H}_{\mathcal{Z}} \otimes \mathcal{H}_{\mathcal{A}}}\left\langle\hat{h}, \phi_{\mathcal{W}}(W) \otimes \phi_{\mathcal{A}}(a)\right\rangle_{\mathcal{H}_{\mathcal{W}} \otimes \mathcal{H}_{\mathcal{A}}} \mid A = a\right]
$$

$$= \mathbb{E}\left[\left\langle \hat{\varphi} \otimes \hat{h}, (\phi_{\mathcal{Z}}(Z) \otimes \phi_{\mathcal{A}}(a)) \otimes (\phi_{\mathcal{W}}(W) \otimes \phi_{\mathcal{A}}(a))\right\rangle_{\mathcal{H}_{\mathcal{Z}} \otimes \mathcal{H}_{\mathcal{A}} \otimes \mathcal{H}_{\mathcal{W}} \otimes \mathcal{H}_{\mathcal{A}}} \mid A = a\right]. \quad (5)$$

As derived in S.M. (C), the expectation in Equation (5) can be approximated using a combination of kernel ridge regression and the properties of the inner products in RKHSs. In essence, the approximation procedure requires learning the conditional mean embedding $\mu_{Z,W|A}(a) = \mathbb{E}\left[\phi_{\mathcal{Z}}(Z) \otimes \phi_{\mathcal{W}}(W) \mid A = a\right]$. We approximate this term with vector-valued kernel ridge regression. In particular, under the regularity condition $\mathbb{E}[g(Z,W) \mid A = \cdot] \in \mathcal{H}_{\mathcal{A}}$ for all $g \in \mathcal{H}_{\mathcal{Z}\mathcal{W}}$, there exists a Hilbert-Schmidt operator $C_{Z,W|A} \in \mathcal{S}_2(\mathcal{H}_{\mathcal{A}}, \mathcal{H}_{\mathcal{Z}\mathcal{W}})$ such that $\mathbb{E}\left[\phi_{\mathcal{Z}}(Z) \otimes \phi_{\mathcal{W}}(W) \mid A = a\right] = C_{Z,W|A}\phi_{\mathcal{A}}(a)$. This operator is learned by minimizing the regularized least-squares function:

$$\hat{\mathcal{L}}_{DR}^c(C) = \frac{1}{t}\sum_{i=1}^t \|\phi_{\mathcal{Z}}(z_i) \otimes \phi_{\mathcal{W}}(w_i) - C\phi_{\mathcal{A}}(a_i)\|_{\mathcal{H}_{\mathcal{Z}\mathcal{W}}}^2 + \lambda_{\text{DR}}\|C\|_{\mathcal{S}_2(\mathcal{H}_{\mathcal{A}}, \mathcal{H}_{\mathcal{Z}\mathcal{W}})}^2,$$

where $\{z_i, w_i, a_i\}_{i=1}^t \subset \mathcal{D}$ denote observations from training set, $t$ is the number of given training samples, and $\lambda_{\text{DR}}$ is the regularization parameter for the vector-valued kernel ridge regression. The minimizer is given by $\hat{\mu}_{Z,W|A}(a) = \hat{C}_{Z,W|A}\phi_{\mathcal{A}}(a) = \sum_i \xi_i(a)\phi_{\mathcal{Z}}(z_i) \otimes \phi_{\mathcal{W}}(w_i)$ where $\xi_i(a) = \left[(\boldsymbol{K}_{AA} + t\lambda_{\text{DR}}\boldsymbol{I})^{-1}\boldsymbol{K}_{Aa}\right]_i$, $\boldsymbol{K}_{AA}$ is the kernel matrix over $\{a_i\}_{i=1}^t$, $\boldsymbol{K}_{Aa}$ is the kernel vector between training points $\{a_i\}_{i=1}^t$ and the target treatment $a$. Using this learned conditional mean embedding, we show that the approximation of Equation (5) results in the following closed-form:

$$\mathbb{E}[\hat{\varphi}(Z,a)\hat{h}(W,a) \mid A = a] \approx \sum_{i=1}^t \xi_i(a)\hat{\varphi}(z_i, a)\hat{h}(w_i, a). \quad (6)$$

We summarize the full procedure in the pseudo-code presented in Algorithm (1). We name our methods **Doubly Robust Kernel Proxy Variable** (DRKPV) and **Doubly Robust Proxy Maximum Moment Restriction** (DRPMMR). Specifically, DRKPV uses the KPV algorithm (summarized in Algorithm 2), while DRPMMR employs the PMMR method (summarized in Algorithm 3).

---

**Algorithm 1** DRKPV / DRPMMR Algorithms

---

**Input**: Training samples $\{y_i, w_i, z_i, a_i\}_{i=1}^t$,
**Parameters:** Regularization parameter $\lambda_{\text{DR}}$ and the parameters of Algorithms (2 or 3) and (4).
**Output**: Doubly robust dose-response estimation $\hat{\theta}_{\text{ATE}}^{(\text{DR})}(a)$ for all $a \in \mathcal{A}$.

1: Collect outcome bridge and dose-response curve estimates, $\hat{h}$ and $\hat{\theta}_1(\cdot)$, with either Algorithm (2, KPV) or (3, PMMR).
2: Collect treatment bridge and the dose-response curve estimates, $\hat{\varphi}$ and $\hat{\theta}_2(\cdot)$, with Algorithm (4).
3: Let $\hat{\theta}_3(\cdot)$ be given by $\hat{\theta}_3(\cdot) = \sum_{i=1}^t \xi_i(\cdot)\hat{\varphi}(z_i, \cdot)\hat{h}(w_i, \cdot)$, where $\xi_i(\cdot) = \left[(\boldsymbol{K}_{AA} + t\lambda_{\text{DR}}\boldsymbol{I})^{-1}\boldsymbol{K}_{A(\cdot)}\right]_i$.
4: For each $a \in \mathcal{A}$, return the doubly robust dose-response estimation as $\hat{\theta}_{\text{ATE}}^{(\text{DR})}(a) = \hat{\theta}_1(a) + \hat{\theta}_2(a) - \hat{\theta}_3(a)$.

---

## 4 Consistency results

In the S.M. (E), we present non-asymptotic uniform consistency guarantees for our proposed doubly robust dose-response curve algorithms, DRKPV and DRPMMR. Theorems (4.1) and (4.2) below are a consequence of our non-asymptotic uniform consistency and demonstrate that our estimators converge to the true causal function.

**Theorem 4.1.** *Suppose Assumptions (3.1), (E.1), (E.2), (E.5), (E.6), (E.8),(E.11), and (E.12) hold. Then, for the DRKPV algorithm with given training samples $\{y_i, w_i, z_i, a_i\}_{i=1}^t$, we obtain that $\sup_{a \in \mathcal{A}} |\hat{\theta}_{ATE}^{(DR)}(a) - \theta_{ATE}(a)| \to 0$ with $t \to \infty$ almost surely by reducing the regularizer $\lambda_{DR}$ and the regularizers of KPV and KAP algorithms at appropriate rates.*

**Theorem 4.2.** *Suppose Assumptions (3.1), (E.1), (E.2), (E.5), (E.6), (E.8-2),(E.16), and (E.17) hold. Then, for DRPMMR algorithm with given training samples $\{y_i, w_i, z_i, a_i\}_{i=1}^t$, we obtain that $\sup_{a \in \mathcal{A}} |\hat{\theta}_{ATE}^{(DR)}(a) - \theta_{ATE}(a)| \to 0$ with $t \to \infty$ almost surely by reducing the regularizer $\lambda_{DR}$ and the regularizers of PMMR and KAP algorithms at appropriate rates.*

The precise high probability finite-sample bounds on the error in the supremum norm and the corresponding optimal regularization parameters are given in Theorems (E.27) and (E.28). Note that both KPV and KAP involve multiple regression stages, each potentially using different sample sizes (i.e., $n_h$, $m_h$, $t_h$, $n_\varphi$, $m_\varphi$, $t_\varphi$). These samples are derived from the original training set $\{y_i, w_i, z_i, a_i\}_{i=1}^{t}$, either through data splitting or reuse across stages. In particular, KPV uses $n_h$, and $m_h$, samples from the training set for its first- and second-stages, respectively, and $t_h$ samples from training set to estimate the dose response. Similarly, KAP uses $n_\varphi$, $m_\varphi$, and $t_\varphi$ samples from the training set for its first-, second-, and third-stages, respectively. We detail the data-splitting procedure in S.M. (F.2). Consequently, the convergence rate in Theorem (E.27) depends on the sizes of these stage-specific subsets as well as the total training size $t$. In contrast, PMMR uses the full dataset of size $t$, and the convergence rate for DRPMMR Theorem (E.28) depends on $t$ and the sample sizes used in the KAP stages. A comprehensive summary of the consistency results for KPV, PMMR, and KAP appears in S.M. (E), and complete proofs for methods' convergence are provided in S.M. (E.3).

**Remark 4.3.** *In Theorems (4.1) and (4.2), we establish uniform consistency of DRKPV and DRPMMR, which ensures control of estimation error across the entire treatment domain. These results hold under smoothness and effective RKHS dimension assumptions.*

- *We note that while kernel ridge regression provides optimal rates in Sobolev–Matérn classes [45], the rates of convergence are slower for larger input dimensions [45, 46]. Our results therefore apply in high-dimensional settings under dimension-dependent smoothness assumptions, but addressing increasing-dimension regimes is left for future work. For further details, see Remark (E.29).*
- *Furthermore, while our estimators are consistent and derived from the EIF, they are not classical one-step estimators [47, 48] and do not automatically achieve local efficiency or asymptotic normality [28]. This remains an interesting topic for future work and is discussed further in Remark (E.30).*

## 5 Numerical experiments

In this section, we assess the performance of our proposed estimators for dose-response curve estimation using both synthetic and real-world datasets. We benchmark our methods against several recent state-of-the-art PCL algorithms, including Proximal Kernel Doubly Robust (PKDR) [22], Kernel Negative Control (KNC) [12], Kernel Proxy Variable (KPV) [11], Proximal Maximum Moment Restriction (PMMR) [11], and Kernel Alternative Proxy (KAP) [23]. Except for experiments involving PKDR, we use a Gaussian kernel of the form $k_\mathcal{F}(f_i, f_j) = \exp(-\|f_i - f_j\|_2^2/(2l^2))$ for each $\mathcal{F} \in \{\mathcal{W}, \mathcal{Z}, \mathcal{A}\}$, where $l$ denotes the kernel bandwidth. The bandwidth is selected using the median heuristic based on pairwise distances. For PKDR, we follow the original implementation by Wu et al. [22] and use the Epanechnikov kernel. We determine the regularization parameter $\lambda_{\text{DR}}$ by utilizing the closed-form expression for leave-one-out cross-validation (LOOCV) in kernel ridge regression. Either LOOCV or a held-out validation set are applied for the regularization terms in the treatment and outcome bridge methods, in line with prior approaches in [11, 12, 23]. We provide additional experimental details—including ablation studies on hyperparameter selection, and scalability analysis with Nyström approximation—in the S.M. (F).

**Synthetic Low Dimensional:** We adopt the synthetic data generation process from Wu et al. [22] which simulates a confounded, nonlinear, and noisy treatment–outcome relationship:

$$U_1 \sim \mathcal{U}[-1, 2], \;\; U_2 \sim \mathcal{U}[0, 1] - \mathbf{1}[0 \le U_1 \le 1], \;\; W = [U_2 + \mathcal{U}[-1, 1], U_1 + \mathcal{N}(0, 1)]$$
$$Z = [U_2 + \mathcal{N}(0, 1), U_1 + \mathcal{U}[-1, 1]], \;\; A := U_1 + \mathcal{N}(0, 1)$$
$$Y := 3\cos(2(0.3U_2 + 0.3U_1 + 0.2) + 1.5A) + \mathcal{N}(0, 1).$$

Here, $\mathcal{U}[a, b]$ denotes the uniform distribution on the interval $[a, b]$, and $\mathcal{N}(\mu, \sigma^2)$ is the Gaussian distribution with mean $\mu$ and variance $\sigma^2$. We run experiments using training sets of sizes 500, 1000, and 2000. Figure (2a) presents the mean squared error (MSE) of different PCL benchmark methods, averaged over 30 independent runs. Our proposed methods, DRKPV and DRPMMR, consistently outperform competing algorithms and demonstrate improved accuracy with increasing data.

**dSprite:** We use the *Disentanglement testing Sprite dataset* (*dSprite*) dataset [49], a collection of $64 \times 64$ grayscale images characterized by latent variables: *scale*, *rotation*, *posX*, and *posY*. Originally designed for disentanglement studies [50], it was recently adapted by Xu et al. [13] as a benchmark for proxy causal learning. In this setup, the treatment is a *high-dimensional vector* obtained by flattening each image and adding Gaussian noise. The target causal function is defined as $\theta_{\text{ATE}}(A) = ((\text{vec}(B)^\top A)^2 - 3000)/500$, where $A \in \mathbb{R}^{4096}$ and $B \in \mathbb{R}^{64 \times 64}$, with entries of $B$ given

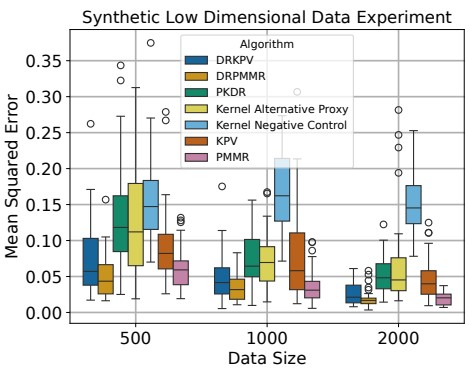

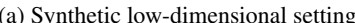

(a) Synthetic low-dimensional setting

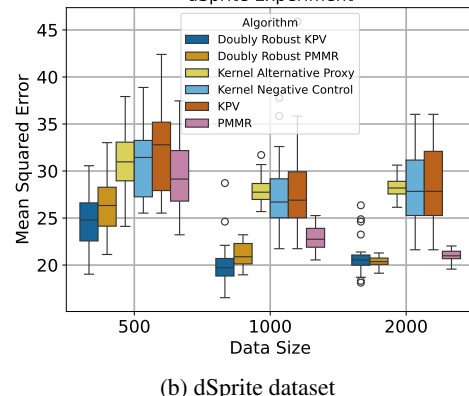

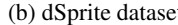

(b) dSprite dataset

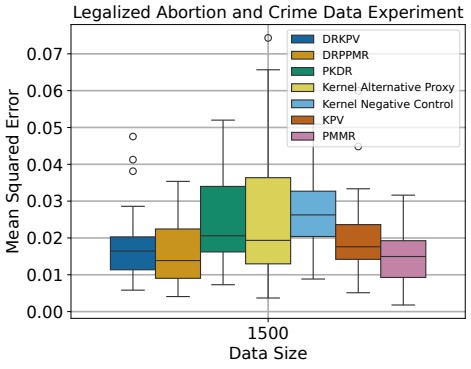

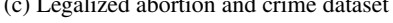

(c) Legalized abortion and crime dataset

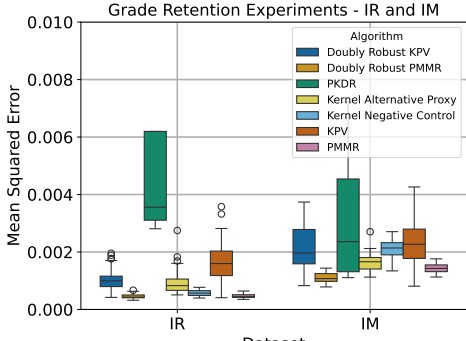

(d) Grade retention and cognitive outcome datasets

Figure 2: Dose-response curve estimation across various datasets and algorithms: *DRKPV* and *DRPMMR* (Ours), PKDR [22], KAP [23], KNC [12], KPV [11], and PMMR [11]. (a) Synthetic low-dimensional setting, (b) dSprite dataset, (c) legalized abortion and crime dataset, and (d) grade retention and cognitive outcome datasets.

by $B_{ij} = |32 - j|/32$. The outcome is generated via $Y = 12(posY - 0.5)^2\theta_{\text{ATE}}(A) + \epsilon$, where $\epsilon \sim \mathcal{N}(0, 0.5)$. The treatment proxy $Z \in \mathbb{R}^3$ comprises the latent variables *scale*, *rotation*, and *posX*. The outcome proxy $W$ is another *dSprite* image sharing the same *posY* value as the treatment image, while the other latent factors are fixed to *scale* = 0.8, *rotation* = 0, and *posX* = 0.5. We run our evaluations using training set sizes of 500, 1000, and 2000. Figure (2b) reports the MSE results averaged across 30 independent runs. Since the implementation of Wu et al. [22] does not support high-dimensional treatments, we omit PKDR. Our methods outperform all others on this high-dimensional benchmark.

**Legalized Abortion and Crime:** We evaluate our methods on the Legalized Abortion and Crime dataset [51], following the preprocessing and setup from [11, 22, 23, 52]. We use the version of the dataset available from the GitHub repository of Mastouri et al. [11][2]. In the causal graph, the treatment $A$ is the effective abortion rate, and the outcome $Y$ is the murder rate. The treatment proxy $Z$ is the generosity of aid to families with dependent children, while the outcome proxy $W$ include beer consumption per capita, the logarithm of the prisoner population per capita, and the presence of a concealed weapons law. Figure (2c) reports the MSE results averaged over 30 runs, where each of the 10 data files[2] is evaluated using three different random runs (leading to different data-splits for different regression stages in KPV and KAP). Our DRKPV and DRPMMR outperform their non-doubly robust counterparts (KPV and PMMR) and all other baselines.

**Grade Retention:** We evaluate the effect of grade retention on long-term cognitive development using data from the ECLS-K panel study [10, 53], following the setup of Mastouri et al. [11][2]. The

---

[2]https://github.com/yuchen-zhu/kernel_proxies

treatment variable $A$ indicates whether a student was retained in grade, and the outcome variable $Y$ corresponds to cognitive test scores in math and reading measured at age 11. The treatment proxy $Z$ consists of the average scores from 1st/2nd and 3rd/4th grade assessments, while the outcome proxy $W$ includes cognitive and behavioral test scores recorded during kindergarten. Figure (2d) reports the MSE results averaged across 30 realizations (3 independent runs per each of the 10 dataset), and compares our method with other proxy-based approaches. On the IM (math retention) dataset, DRKPV and DRPMMR outperform their non-doubly robust counterparts (KPV and PMMR) and all baselines. On the IR (reading retention) dataset, DRKPV outperforms KPV and all other methods except PMMR, while DRPMMR performs on par with PMMR, both outperforming the rest.

**Doubly Robust Estimation in Misspecified Setting:** We further evaluate the robustness of our methods when the bridge functions are misspecified. By representer theorem [44], the bridge functions in KPV, PMMR, and KAP can be expressed as linear combinations of feature maps in their respective RKHSs: $\hat{h} = \sum_{i=1}^{n_h} \sum_{j=1}^{m_h} \alpha_{ij} \phi_{\mathcal{W}}(w_i) \otimes \phi_{\mathcal{A}}(\tilde{a}_j)$ for KPV, $\hat{h} = \sum_{i=1}^{t} \alpha_i \phi_{\mathcal{W}}(w_i) \otimes \phi_{\mathcal{A}}(a_i)$ for PMMR, and $\hat{\varphi} = \sum_{l=1}^{m_\varphi} \sum_{t=1}^{m_\varphi} \gamma_{lt} \hat{\mu}_{Z|W,A}(\tilde{w}_t, \tilde{a}_l) \otimes \phi_{\mathcal{A}}(\tilde{a}_l)$ for KAP (see S.M. (C.1) and (C.2)). To simulate misspecification, we first train the DRKPV (or DRPMMR) in the synthetic low-dimensional setup and then perturb either the outcome (KPV/PMMR) or treatment (KAP) bridge coefficients by adding Gaussian noise. Figures (3a) and (3b) show DRKPV results averaged over five independent runs with standard deviation bands. In Figure (3a), we illustrate the result when the outcome bridge coefficients are perturbed via $\alpha_{ij} \leftarrow \alpha_{ij} + \varepsilon_{ij}$, where $\varepsilon_{ij} \sim \mathcal{N}(0, 0.2)$. Similarly, Figure (3b) presents results when the treatment bridge function is misspecified by jittering $\gamma_{ij} \leftarrow \gamma_{ij} + \varepsilon_{ij}$ with $\varepsilon_{ij} \sim \mathcal{N}(0, 0.2)$. In these plots, the term *slack prediction* denotes the empirical estimate of $\mathbb{E}[\hat{\varphi}(Z, a)\hat{h}(W, a) \mid A = a]$ in Equation (6). Despite misspecification, DRKPV recovers the true causal function, as the slack term offsets the error. Figures (3c) and (3d) show analogous experiments for DRPMMR. Once again, we perturb one of the learned bridge functions by injecting Gaussian noise $\mathcal{N}(0, 0.2)$ into its coefficients. As in the DRKPV case, DRPMMR continues to accurately recover the true causal effect, demonstrating its robustness to misspecification of bridge functions. For enhanced legibility, a larger version of this figure is provided in S.M. (F.4).

Figures (5a)–(5d) in S.M. (F.4) illustrate the same experiments under higher perturbation, where the coefficients are jittered with $\varepsilon_{ij} \sim \mathcal{N}(0, 0.5)$. Furthermore, additional robustness evaluations are provided in S.M. (F.5) where we adopt the semi-synthetic setups from [23, Section 13] based on the JobCorps dataset [54, 55]. These experiments vary the informativeness of the proxy variables to challenge the outcome and treatment bridge completeness assumptions (Assumptions (2.3) and (2.5)).

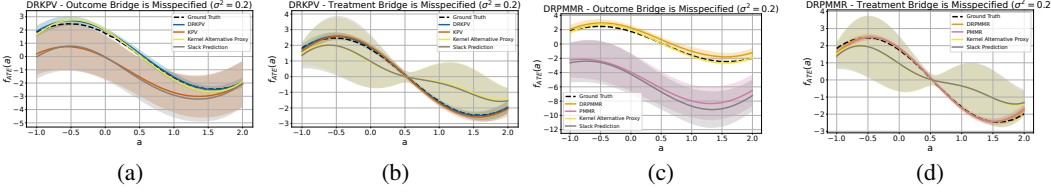

Figure 3: Experimental results in bridge function misspecifications with the synthetic low-dimensional data: (a, b) DRKPV estimates under outcome and treatment bridge misspecifications, respectively; (c, d) DRPMMR estimates under outcome and treatment bridge misspecifications, respectively.

## 6 Discussion and conclusion

We introduce two doubly robust estimators—DRKPV and DRPMMR—in the proxy causal learning framework that address unmeasured confounding without requiring explicit density ratio estimation. Built on kernel mean embeddings, both estimators have closed-form expressions and effectively combine outcome and treatment bridge functions to recover the dose-response curve. They are provably consistent and remain robust under misspecification of either bridge. Empirically, our methods outperform recent baselines on challenging benchmarks and scale well to high-dimensional treatments. However, the curse of dimensionality poses a subtle theoretical limitation when the input dimension grows with the sample size; addressing this is an interesting area for future work. A key limitation of our work is the computational cost of kernel methods; while we present an initial step toward scalability using Nyström approximation, exploring stronger scalable methods remains a primary direction for future research.

## Acknowledgments and Disclosure of Funding

Bariscan Bozkurt, Houssam Zenati, Dimitri Meunier, and Arthur Gretton are supported by Gatsby Charitable Foundation. We are grateful for the constructive feedback and insightful discussion provided by the anonymous reviewers of NeurIPS 2025.

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

## Supplementary Material for Density Ratio-Free Doubly Robust Proxy Causal Learning

Section (A) reviews our doubly robust identification result and proves Theorem (2.7). Section (B) outlines semiparametric theory and derives the efficient influence function for discrete treatments. In Section (C), we present derivations for our proposed algorithms: Doubly Robust Kernel Proxy Variable (DRKPV) and Doubly Robust Proxy Maximum Moment Restriction (DRPMMR). Section (D) discusses conditions for the existence of outcome and treatment bridge functions. Section (E) proves consistency of our methods. Section (F) details the experimental setup, including hyperparameter tuning and additional results.

## A  Doubly robust identification

Here, we prove Theorem (2.7). For completeness, we restate the theorem below.

**Theorem A.1** (Doubly robust causal identification; Replica of Theorem (2.7))**.** *The dose-response can be identified with*

$$\theta_{ATE}^{(DR)}(a; h, \varphi) \mid_{(h=h_0, \varphi=\varphi_0)} = \mathbb{E}[\varphi_0(Z, a)(Y - h_0(W, a)) \mid A = a] + \mathbb{E}[h_0(W, a)],$$

*where $h_0$ and $\varphi_0$ are the outcome and treatment bridge functions that satisfy Equations (1) and (2), respectively. Furthermore, $\theta_{ATE}^{(DR)}(a; h, \varphi)$ admits* double robustness *such that $\theta_{ATE}^{(DR)}$ identifies the dose-response curve if **either** h solves Equation (1) **or** $\varphi$ solves Equation (2)—but not necessarily both.*

*Proof.* To understand why $\theta_{ATE}^{DR}(a)$ exhibits double robustness, we consider the following two cases:

*Case i:* Suppose that $h$ is correctly specified, i.e. $h = h_0$. Then,

$$\theta_{ATE}^{(DR)}(a; h_0, \varphi) = \mathbb{E}[\varphi(Z, a)(Y - h_0(W, a)) \mid A = a] + \mathbb{E}[h_0(W, a)]$$
$$= \int \varphi(z, a) \underbrace{\left(\int (y - h_0(w, a)) p(y, w|z, a) dy dw\right)}_{=0 \text{ due to Equation (1)}} p(z|a) dz + \mathbb{E}[h_0(W, a)] = \theta_{ATE}(a)$$

where the final equality is due to Theorem (2.4).

*Case ii:* Now, suppose that $\varphi$ is correctly specified, i.e., $\varphi = \varphi_0$. Then, we note that

$$\theta_{ATE}^{(DR)}(a; h, \varphi_0) = \mathbb{E}[Y \varphi_0(Z, a) \mid A = a] - \mathbb{E}[\varphi_0(Z, a) h(W, a) \mid A = a] + \mathbb{E}[h(W, a)]$$

First off, note that the first component $\mathbb{E}[Y \varphi_0(Z, a) \mid A = a]$ identifies the dose-response due to Theorem (2.6). Next, we consider the remaining two terms

$$- \mathbb{E}[\varphi_0(Z, a) h(W, a) \mid A = a] + \mathbb{E}[h(W, a)]$$
$$= - \int \int \varphi_0(z, a) h(w, a) p(z, w|a) dz dw + \mathbb{E}[h(W, a)]$$
$$= - \int \int \varphi_0(z, a) h(w, a) p(z|w, a) p(w|a) dz dw + \mathbb{E}[h_0(W, a)]$$
$$= - \int \left(\int \varphi_0(z, a) p(z|w, a) dz\right) h(w, a) p(w|a) dw + \mathbb{E}[h(W, a)]$$
$$= - \int \left(\mathbb{E}[\varphi_0(Z, a)|w, a]\right) h(w, a) p(w|a) dw + \mathbb{E}[h_0(W, a)]$$
$$= - \int \frac{p(w)}{p(w|a)} h(w, a) p(w|a) dw + \mathbb{E}[h(W, a)] = -\mathbb{E}[h(W, a)] + \mathbb{E}[h(W, a)] = 0,$$

where the first equality in the last line above is due to Equation (2). $\square$

The following lemma is the direct implication of Theorem (2.7), which shows that the dose-response curve can also be identified with the conditional expectation of the multiplication of bridge functions.

**Lemma A.2.** *Let $h_0$ and $\varphi_0$ be outcome and treatment bridge functions satisfying Equation (1) and (2), respectively. Then, the dose-response can be identified by the following slack function $\theta_{ATE}(a) = \mathbb{E}[\varphi_0(Z,a)h_0(W,a) \mid A = a]$.*

*Proof.* As we observe in the second case of the proof of Theorem (2.7)

$$\mathbb{E}[\varphi_0(Z,a)h_0(W,a) \mid A = a] = \int \frac{p(w)}{p(w|a)} h_0(w,a)p(w|a)dw = \mathbb{E}[h_0(W,a)] = \theta_{ATE}(a).$$

$\square$

# B Derivation of efficient influence function in discrete treatment setting

## B.1 Background on efficient influence functions

Let $\mathcal{P}$ denote a nonparametric statistical model, and let $\psi : \mathcal{P} \to \mathbb{R}$ be a target functional of interest—such as the dose-response curve $\theta_{\text{ATE}}(a)$. In semiparametric efficiency theory, a statistical parameter such as the average treatment effect is viewed as a functional mapping the underlying data-generating law $p \in \mathcal{P}$ to a real number. In this section, we denote by $\mathbb{P}$ the true distribution with $\mathbb{P}\ell = \mathbb{E}[\ell]$, $\mathbb{P}_n$ the empirical distribution with $\mathbb{P}_n(\ell) := \frac{1}{n}\sum_{i=1}^n \ell(Z_i)$. The $\mathcal{L}_2(\mathbb{P})$ norm is written $\|\ell\|^2 := \int \ell^2 d\mathbb{P}$. We are interested in how the functional $\theta$ changes as the data-generating distribution $p \in \mathcal{P}$ varies. When $\theta$ is sufficiently smooth, it admits a *Gâteaux derivative*, defined as:

$$\theta'(p; H) := \frac{d}{d\epsilon}\theta(p + \epsilon H)\Big|_{\epsilon=0}$$

for $H$ in the tangent space of $\mathcal{P}$. The functional is said to be Gâteaux differentiable at $P$ if $\theta'(p; H)$ exists and is linear and continuous in $H$. The Gâteaux derivative formalizes the local sensitivity of the functional to infinitesimal perturbations of the distribution. It extends the notion of a directional derivative to the space of probability laws and provides a linear approximation of how $\theta(p)$ varies when $p$ is displaced along an admissible direction $H$ within the tangent space.

This gives rise to the *von Mises expansion* [56, 28]:

$$\theta(q) = \theta(p) + \theta'(p; q - p) + R_2(p, q)$$

where $R_2(p, q)$ is a second-order remainder depending on products or squares of differences between $q$ and $p$. The von Mises expansion plays an analogous role to a first-order Taylor expansion in finite dimensions, expressing the change in the functional as a linear term—captured by the influence function—plus a higher-order remainder. This expansion enables the definition of the *influence function*.

**Definition B.1** (Influence function [29, 28, 31]). *A measurable function $\psi(\cdot; p) \in \mathcal{L}_2(p)$ is the influence function of $\theta$ at $p$ if*

$$\theta'(p; q - p) = \int \psi(z; p)\, d(q - p)(z), \quad \text{for all } q \in \mathcal{P}$$

*with $\int \psi(z; p)dp(z) = 0$ and $\mathbb{E}_p[\psi(Z; p)^2] < \infty$.*

The influence function $\psi$ is the *Riesz representer* of the derivative $\theta'(p; \cdot)$, as $\mathcal{L}_2(p)$ is a Hilbert space. It is also referred to as the *pathwise derivative gradient* [57] or *Neyman orthogonal score* [30, 33]. Intuitively, the influence function measures how sensitive the parameter of interest is to infinitesimal changes in the data-generating distribution. It can be seen as the "directional derivative" of the functional with respect to perturbations of the underlying distribution—much like a gradient in function space. This makes it a key tool for constructing estimators that correct first-order bias and achieve minimal asymptotic variance.

To compute influence functions, we analyze *parametric submodels* $\{p_\epsilon\} \subset \mathcal{P}$ such that $p_{\epsilon=0} = p$ and $\epsilon \mapsto p_\epsilon$ is smooth. A canonical choice is the *tilted model*:

$$\frac{dp_\epsilon}{dp}(z) = 1 + \epsilon s(z), \quad \text{where } \|s\|_\infty \leq M, \; \epsilon < 1/M$$

The associated *score function* is $s_\epsilon(z) = s(z) = \frac{\partial}{\partial \epsilon} \log p_\epsilon(z)\big|_{\epsilon=0}$ [58, 31].

Applying the von Mises expansion to $p_\epsilon$, we obtain:

$$\theta(p_\epsilon) = \theta(p) + \epsilon \int \psi(z;p)s(z)\,dp(z) + R_2(p, p_\epsilon)$$

Since $R_2(p, p_\epsilon)$ is second-order, it satisfies $\frac{d}{d\epsilon} R_2(p, p_\epsilon)\big|_{\epsilon=0} = 0$, a condition known as *Neyman orthogonality* [33]. Thus,

$$\frac{d}{d\epsilon}\theta(p_\epsilon)\bigg|_{\epsilon=0} = \int \psi(z;p)s(z)\,dp(z)$$

This defines *pathwise differentiability*, and the function $\psi$ satisfying this identity is the efficient influence function.

**Definition B.2** (Pathwise differentiability [29, 28]). *A functional $\theta$ is pathwise differentiable at $p$ if for every smooth parametric submodel $\{p_\epsilon\} \subset \mathcal{P}$ with score $s_\epsilon(z)$, we have:*

$$\frac{d}{d\epsilon}\theta(p_\epsilon)\bigg|_{\epsilon=0} = \int \psi(z;p)s_\epsilon(z)\,dp(z)$$

Once $\psi$ is known, it can be used to construct an *efficient estimator*. Suppose $\widehat{p}$ estimates $p$, and let $\mathbb{P}_n$ denote the empirical measure. The *plug-in estimator* $\widehat{\theta}_{\text{pi}} := \theta(\widehat{p})$ admits the expansion:

$$\theta(\widehat{p}) - \theta(p) = -\mathbb{E}_p[\psi(Z; \widehat{p})] + R_2(\widehat{p}, p)$$

This motivates the *one-step estimator*, which corrects the plug-in estimator with an estimation of the first-order bias $\mathbb{E}_P[\psi(Z; \widehat{p})]$:

$$\widehat{\theta}^{\text{os}} := \theta(\widehat{p}) + \mathbb{P}_n[\psi(Z; \widehat{p})]$$

When analyzing one-step estimators, it is common to write the following decomposition [28]:

$$\widehat{\theta}^{\text{os}} - \theta(p) = (\mathbb{P}_n - \mathbb{P})[\psi(Z;p)] + \underbrace{(\mathbb{P}_n - \mathbb{P})[\psi(Z;\widehat{p}) - \psi(Z;p)]}_{\text{empirical process}} + \underbrace{R_2(\widehat{p}, p)}_{\text{second-order remainder}}.$$

The first term is a sample average and converges by the central limit theorem. The second term, known as the *empirical process*, vanishes at rate $o_{\mathbb{P}}(n^{-1/2})$ if the estimated influence function $\widehat{\psi}$ is $\mathcal{L}_2$-consistent (i.e $\|\widehat{\psi}(Z) - \psi(Z)\|_2 \to_{\mathbb{P}} 0$, where $\|\ell\|_2^2 = \int \ell^2 dp$.) which is the case if it lies in a Donsker class or is estimated via cross-fitting (see [28]).

The third term, $R_2(\widehat{p}, p)$, is the *second-order remainder*, which captures nonlinear errors. This term often exhibits a rate which depends on product of nuisance error rates. Therefore, under standard product-rate conditions, such as nuisance errors converging at rate $n^{-1/4}$, we obtain $R_2 = o_{\mathbb{P}}(n^{-1/2})$.

Together, these results imply that $\widehat{\theta}^{\text{os}}$ is asymptotically linear and efficient. Specifically, in nonparametric models, the best achievable precision for estimating a target parameter $\theta$ is governed by a *local asymptotic minimax bound* which is attained by this estimator.

This bound extends the classical Cramér–Rao inequality using pathwise differentiability and parametric submodels. For any smooth submodel $\{p_\epsilon\}$ with score $s(z)$, the Cramér–Rao lower bound is:

$$\frac{(\mathbb{E}[\psi(Z;p)s(z)])^2}{\mathbb{E}[s(z)^2]} \leq \mathbb{E}[\psi(Z;p)^2],$$

where $\psi$ is the influence function. Equality is achieved when $s = \psi$, yielding the *nonparametric efficiency bound*:

$$\text{var}\{\psi(Z;p)\}$$

Since the one-step estimator satisfies

$$\sqrt{n}(\widehat{\theta}^{\text{os}} - \theta) \rightsquigarrow N(0, \text{var}\{\psi(Z;p)\}),$$

it achieves this bound and is thus locally efficient.

Eventually, note that the efficient influence function (EIF) is defined relative to the model's *tangent space*, which formalizes the allowable directions of perturbation around the true distribution. The tangent space is the set of valid perturbations around the true law and thus determines which directions contribute to efficiency.

In *nonparametric models*, the tangent space is the entire Hilbert space of square-integrable, mean-zero functions:
$$\mathcal{T} = \{\ell \in \mathcal{L}_2(\mathbb{P}) : \mathbb{E}_{\mathbb{P}}[\ell(Z)] = 0\}.$$

This ensures that any regular, pathwise differentiable functional admits a *unique* influence function $\psi$ that also serves as the EIF.

In contrast, a *semiparametric model* imposes structural constraints on the distribution (e.g., conditional independence, smoothness, or dimension reduction). These constraints reduce the tangent space to a strict subspace of $\mathcal{L}_2(\mathbb{P})$. In this case, there may exist multiple functions $\psi$ satisfying the von Mises expansion, but only those that lie within the tangent space correspond to valid perturbations. Among them, the EIF is defined as the unique $\psi$ that also lies in the tangent space and minimizes variance. Therefore, careful attention must be paid to ensure that the candidate $\psi$ lies in the tangent space, as only then does it correspond to the efficiency bound [28].

## B.2 Derivation of the efficient influence function

In this section, we consider discrete treatment setting, i.e., $A \in \{0, 1, \ldots, d_{\mathcal{A}}\}$. We define the conditional expectation operator $T : \mathcal{L}_2(p_{W,A}) \to \mathcal{L}_2(p_{Z,A})$ by $T(\ell) = \mathbb{E}[\ell(W, A) \mid Z, A]$, and its adjoint $T^* : \mathcal{L}_2(p_{Z,A}) \to \mathcal{L}_2(p_{W,A})$ by $T^*(\ell) = \mathbb{E}[\ell(Z, A) \mid W, A]$, where $p_{Z,A}$ and $p_{W,A}$ denote the joint distributions of the random variables $(Z, A)$ and $(W, A)$, respectively. Here, $\mathcal{L}_2(\mathcal{F}, p)$ denotes the space of square-integrable functions on domain $\mathcal{F}$ with respect to the measure $p$. We impose the following regularity condition on the conditional mean operators following the setup in Cui et al. [21]:

**Assumption B.3.** *$T$ and $T^*$ are surjective.*

Under this setting, we derive the efficient influence function (EIF) of the dose-response curve, following the framework of [21, Theorem 3.1].

**Theorem B.4.** *(Efficient influence function) The efficient influence function of $\theta_{ATE}(a)$ where Assumption (B.3) holds, and Equation (1) holds at the true data generating law, is given by*

$$\psi_{ATE}(\mathcal{O}; a) = \varphi_0(Z, A)(Y - h_0(W, A))\frac{1}{p(A)}\mathbf{1}[A = a] + h_0(W, a) - \theta_{ATE}(a)$$

*where $\mathcal{O}$ is the collection of the variables $(Y, A, W, Z)$, and $\mathbf{1}[\cdot]$ is the indicator function. Therefore, the corresponding semiparametric local efficiency bound of $\theta_{ATE}(a)$ equals to $\mathbb{E}\left[\psi_{ATE}(\mathcal{O}; a)^2\right]$.*

**Remark B.5.** *We note that our identification result for the dose-response curve in the continuous treatment setting can be viewed as a generalization of the identification result implied by the efficient influence function in Theorem (B.4). Specifically, observe that, since the expectation of the influence function is zero, we have:*

$$0 = \mathbb{E}[\varphi_0(Z, a)(Y - h_0(W, a))\frac{1}{p(a)}\mathbf{1}[A = a] + h_0(W, a) - \theta_{ATE}(a)]$$
$$= \mathbb{E}[\varphi_0(Z, a)(Y - h_0(W, a)) \mid A = a] + \mathbb{E}[h(W, a)] - \theta_{ATE}(a)$$

*Therefore, in the discrete treatment case, the dose-response curve can be identified by*

$$\theta_{ATE}(a) = \mathbb{E}[\varphi_0(Z, a)(Y - h_0(W, a)) \mid A = a] + \mathbb{E}[h(W, a)] \tag{7}$$

*In Theorem (2.7), we show that the same representation identifies the dose-response curve in the continuous treatment case, while also exhibiting the double robustness property.*

**Remark B.6.** *We note that the formulation of our efficient influence function (EIF) differs from that of Cui et al. [21] particularly in the treatment bridge function term. In their framework, the treatment bridge function $q_0 : \mathcal{Z} \times \mathcal{A} \to \mathbb{R}$ is defined as the solution to*

$$\mathbb{E}[q_0(Z, a)|W, A = a] = \frac{1}{p(A = a \mid W)} = \frac{p(W)}{p(W, a)}, \quad \forall a \in \mathcal{A}, \tag{8}$$

*and the outcome bridge function solves (1). Under the Assumption (B.3) and assuming that Equation (1) holds under the true data generating law, Cui et al. [21] shows that the EIF of the dose-response is given by*

$$\psi^q_{ATE}(\mathcal{O}; a) = q_0(Z, A)(Y - h_0(W, A))\mathbf{1}[A = a] + h_0(W, a) - \theta_{ATE}(a),$$

*which leads to the identification formula*

$$\theta_{ATE}(a) = \mathbb{E}\left[q_0(Z, A)(Y - h_0(W, A))\mathbf{1}[A = a]\right] + \mathbb{E}[h(W, a)]. \tag{9}$$

*Instead, our identification result in Equation (7) differs from Equation (9) in two key ways: (i) We use a treatment bridge function $\varphi_0$ that solves Equation (2), which includes an additional factor of $p(a)$ in the numerator compared to Equation (8); (ii) As a result, the first term in our identification formula involves the conditional expectation $\mathbb{E}[\varphi_0(Z, a)(Y - h_0(W, a)) \mid A = a]$ over the distribution $p(Y, Z, W \mid A = a)$, whereas in Equation (9), the expectation is taken over the joint distribution $p(Y, Z, W, A)$ with an indicator function enforcing the treatment level. Because of this difference, our framework extends naturally to continuous treatments without the kernel smoothing step of Wu et al. [22], whose technique has been extensively used in the doubly-robust literature to handle continuous treatments [59, 60, 61]. Furthermore, this structural difference extends the applicability of our methods to high-dimensional treatments, where kernel-smoothing-based DR approaches are ineffective as we demonstrated with dSprite experiment in Section (5).*

*Proof of Theorem (B.4).* We consider the following [28, 31, 57] parametric sub-model for nonparametric $\mathcal{P}$ indexed by $\epsilon$ which includes the true data generating law at $\epsilon = 0$ for some mean-zero function $\ell : \mathcal{O} \to \mathbb{R}$,

$$p_\epsilon(\mathcal{O}) = p(\mathcal{O})\{1 + \epsilon\ell(\mathcal{O})\} \tag{10}$$

where $\mathcal{O} = (Y, Z, W, A)$ is the collection of variables of interest, $\|\ell\|_\infty \leq M < \infty$, and $\epsilon < 1/M$ so that $p_\epsilon(\mathcal{O}) \geq 0$. We have the score function $s_\epsilon(\mathcal{O})|_{\epsilon=0} = \frac{\partial}{\partial \epsilon} \log p_\epsilon(\mathcal{O})\big|_{\epsilon=0}$. To find the efficient influence function for $\theta_{ATE}(a)$, we need to find the curve $\psi_{ATE}$ that satisfies

$$\frac{\partial}{\partial \epsilon} \theta_{ATE}(a; p_\epsilon)\bigg|_{\epsilon=0} = \int \psi_{ATE}(\mathcal{O}; p)s_\epsilon(\mathcal{O})dp(\mathcal{O})\bigg|_{\epsilon=0} \tag{11}$$

First of all, we recall that

$$\mathbb{E}[Y \mid Z, A] = \int h(w, A)dp(w \mid Z, A)$$

and define the bridge function $h_\epsilon$ so that:

$$\mathbb{E}_{p_\epsilon}[Y \mid Z, A] = \int h_\epsilon(w, A)p_\epsilon(w \mid Z, A)dw.$$

Therefore, we have

$$\int (y - h_\epsilon(w, a))\, p_\epsilon(w, y|z, a)\, |_{\epsilon=0}\, dwdy = 0,$$

which implies that

$$\int \frac{\partial h_\epsilon(w, a)}{\partial \epsilon}\bigg|_{\epsilon=0} p(w|z, a)dw = \int (y - h(w, a))\frac{\partial log p_\epsilon(w, y|z, a)}{\partial \epsilon}p_\epsilon(w, y|z, a)\bigg|_{\epsilon=0} dwdy \tag{12}$$

Now, we recall that

$$\theta_{ATE}(a; p_\epsilon) = \int h_\epsilon(w, a)p_\epsilon(w)dw.$$

Hence,

$$\begin{aligned}
\frac{\partial \theta_{ATE}(a; p_\epsilon)}{\partial \epsilon}\bigg|_{\epsilon=0} &= \int \frac{\partial h_\epsilon(w, a)}{\partial \epsilon}\bigg|_{\epsilon=0} p(w)dw + \int h(w, a)\frac{\partial p_\epsilon(w)}{\partial \epsilon}\bigg|_{\epsilon=0} dw \\
&= \int \frac{\partial h_\epsilon(w, a)}{\partial \epsilon}\bigg|_{\epsilon=0} p(w)dw + \int h(w, a)\frac{\partial \log p_\epsilon(w)}{\partial \epsilon}\bigg|_{\epsilon=0} p(w)dw \tag{13}
\end{aligned}$$

We notice that

$$\int h(w,a)\frac{\partial \log p_\epsilon(w)}{\partial \epsilon}\bigg|_{\epsilon=0} p(w)dw = \mathbb{E}\left[(h(W,a) - \theta_{ATE}(a))\frac{\partial \log p_\epsilon(Z,Y,A,W)}{\partial \epsilon}\bigg|_{\epsilon=0}\right]. \quad (14)$$

One can show this as follows:

$$\mathbb{E}\left[(h(W,a) - \theta_{ATE}(a))\frac{\partial \log p_\epsilon(Z,Y,A,W)}{\partial \epsilon}\bigg|_{\epsilon=0}\right]$$

$$= \int h(w,a)\frac{\partial p_\epsilon(z,y,a',w)}{\partial \epsilon}\bigg|_{\epsilon=0} dp(z,y,a',w) - \theta_{ATE}(a)\int \frac{\partial \log p_\epsilon(z,y,a',w)}{\partial \epsilon}\bigg|_{\epsilon=0} dp(z,y,a',w)$$

$$= \int h(w,a)\frac{\partial \log p_\epsilon(w)}{\partial \epsilon}\bigg|_{\epsilon=0} dp(z,y,a',w) + \int h(w,a)\frac{\partial \log p_\epsilon(z,y,a'|w)}{\partial \epsilon}\bigg|_{\epsilon=0} dp(z,y,a',w)$$

$$- \theta_{ATE}(a)\underbrace{\int \frac{\partial \log p_\epsilon(w)}{\partial \epsilon}\bigg|_{\epsilon=0} dp(z,y,a',w)}_{=0} -\theta_{ATE}(a)\int \frac{\partial \log p_\epsilon(z,y,a'|w)}{\partial \epsilon}\bigg|_{\epsilon=0} dp(z,y,a',w)$$

$$= \int h(w,a)\frac{\partial \log p_\epsilon(w)}{\partial \epsilon}\bigg|_{\epsilon=0} dp(z,y,a',w) + \int (h(w,a) - \theta_{ATE}(a))\frac{\partial \log p_\epsilon(z,y,a'|w)}{\partial \epsilon}\bigg|_{\epsilon=0} dp(z,y,a',w)$$

$$= \int h(w,a)\frac{\partial \log p_\epsilon(w)}{\partial \epsilon}\bigg|_{\epsilon=0} dp(z,y,a',w)$$

$$+ \int (h(w,a) - \theta_{ATE}(a))\underbrace{\left(\int \frac{\partial \log p_\epsilon(z,y,a'|w)}{\partial \epsilon}\bigg|_{\epsilon=0} dp(z,y,a',|w)\right)}_{=0} dp(w)$$

$$= \int h(w,a)\frac{\partial \log p_\epsilon(w)}{\partial \epsilon}\bigg|_{\epsilon=0} dp(w).$$

Next, we consider the first component in Equation (13):

$$\int \frac{\partial h_\epsilon(w,a)}{\partial \epsilon}\bigg|_{\epsilon=0} p(w)dw = \int \frac{\partial h_\epsilon(w,a)}{\partial \epsilon}\bigg|_{\epsilon=0}\frac{p(w)}{p(w,a)}p(w,a)dw$$

$$= \int \frac{\partial h_\epsilon(w,a')}{\partial \epsilon}\bigg|_{\epsilon=0}\mathbf{1}[a=a']\frac{p(w)}{p(w,a')}p(w,a')dwda'$$

$$= \int \frac{\partial h_\epsilon(w,a')}{\partial \epsilon}\bigg|_{\epsilon=0}\mathbf{1}[a=a']\frac{1}{p(a')}\mathbb{E}[\varphi_0(Z,a') \mid W=w,A=a']p(w,a')dwda'$$

$$= \int \frac{\partial h_\epsilon(w,a')}{\partial \epsilon}\bigg|_{\epsilon=0}\mathbf{1}[a=a']\frac{1}{p(a')}\varphi_0(z,a')p(z,w,a')dwda'$$

$$= \int \mathbf{1}[a=a']\frac{1}{p(a')}\varphi_0(z,a')\left(\int \frac{\partial h_\epsilon(w,a')}{\partial \epsilon}\bigg|_{\epsilon=0}p(w|z,a')dw\right)p(z,a')dzda'$$

$$= \int \mathbf{1}[a=a']\frac{1}{p(a')}\varphi_0(z,a')\left(\int (y-h(w,a))\frac{\partial logp_\epsilon(w,y|z,a)}{\partial \epsilon}\bigg|_{\epsilon=0}p(w,y|z,a')dwdy\right)p(z,a')dzda'$$
$$(15)$$

$$= \int \mathbf{1}[a=a']\frac{1}{p(a')}\varphi_0(z,a')(y-h(w,a))\frac{\partial logp_\epsilon(w,y|z,a')}{\partial \epsilon}\bigg|_{\epsilon=0}p(w,y,z,a')dwdydzda'$$

$$+ \underbrace{\int \mathbf{1}[a=a']\frac{1}{p(a')}\varphi_0(z,a')(y-h(w,a))\frac{\partial logp_\epsilon(z,a')}{\partial \epsilon}\bigg|_{\epsilon=0}p(w,y,z,a')dwdydzda'}_{=0}$$

$$= \mathbb{E}\left[\mathbf{1}[A=a]\frac{1}{p(A)}\varphi_0(Z,A)(Y-h(W,A))\frac{\partial \log p_\epsilon(W,Y,Z,A)}{\partial \epsilon}\bigg|_{\epsilon=0}\right], \quad (16)$$

where equality in Equation (15) is due to Equation (12). Now, combining the results in Equations (14) and (16), we obtain

$$\frac{\partial \theta_{ATE}(a;p_\epsilon)}{\partial \epsilon}\bigg|_{\epsilon=0} =$$

$$\mathbb{E}\left[\left(\mathbf{1}[A=a]\frac{1}{p(A)}\varphi_0(Z,A)(Y-h(W,A))+h(W,a)-\theta_{ATE}(a)\right)\frac{\partial\log p_\epsilon(W,Y,Z,A)}{\partial\epsilon}\Big|_{\epsilon=0}\right],$$

which shows that the efficient influence function for $\theta_{ATE}(a)$ is given by

$$\psi_{ATE}(\mathcal{O};a)=\varphi_0(Z,A)(Y-h(W,A))\frac{1}{p(A)}\mathbf{1}[A=a]+h_0(W,a)-\theta_{ATE}(a) \qquad (17)$$

Next, we show that $\psi_{ATE}(a)$ belongs to a tangent space $\mathcal{T}_1+\mathcal{T}_2$ using Assumption (B.3), where

$\mathcal{T}_1:=\{s(Z,A)\in\mathcal{L}_2(\mathcal{Z}\times\mathcal{A},p_{Z,A})\mid\mathbb{E}[s(Z,A)]=0\}$

$\mathcal{T}_2:=\{s(Y,W\mid Z,A)\in\mathcal{L}_2(\mathcal{Z}\times\mathcal{A},p_{Z,A})^\perp\mid\mathbb{E}[(Y-h(W,A)s(Y,W\mid Z,A)\mid Z,A]\in cl(\mathcal{R}(T))]\}.$

Here, $\mathcal{L}_2(\mathcal{Z}\times\mathcal{A},p_{Z,A})^\perp$ is the orthogonal complement of $\mathcal{L}_2(\mathcal{Z}\times\mathcal{A},p_{Z,A})$, $\mathcal{R}(T)$ is the range space of the conditional mean operator $T$, and $cl(\cdot)$ denotes the closure. We decompose Equation (17) as follows:

$$\begin{aligned}&\mathbb{E}[h(W,a)-\theta_{\text{ATE}}(a)\mid Z,A]\\&+h(W,a)-\theta_{\text{ATE}}(a)-\mathbb{E}[h(W,a)-\theta_{\text{ATE}}(a)\mid Z,A]\\&+\frac{1}{p(A)}\mathbf{1}[A=a]\varphi_0(Z,A)(Y-h(W,A)).\end{aligned}$$

Notice that $\mathbb{E}[h(W,a)-\theta_{\text{ATE}}(a)\mid Z,A]\in\mathcal{T}_1$ due to Equation (1). For the remaining two terms, notice that

$$\mathbb{E}\left[h(W,a)-\theta_{\text{ATE}}(a)-\mathbb{E}[h(W,a)-\theta_{\text{ATE}}(a)\mid Z,A]\mid Z,A\right]=0 \qquad (18)$$

$$\mathbb{E}\left[\frac{1}{p(A)}\mathbf{1}[A=a]\varphi_0(Z,A)(Y-h(W,A))\mid Z,A\right]=0. \qquad (19)$$

Hence, using Equations (18) and (19) and Assumption (B.3), we observe that

$$\mathbb{E}\left[(Y-h_0(W,A))\left(h(W,a)-\theta_{\text{ATE}}(a)-\mathbb{E}[h(W,a)-\theta_{\text{ATE}}(a)\mid Z,A]\right)\mid Z,A\right]\in cl(\mathcal{R}(T)),$$

$$\mathbb{E}\left[\frac{1}{p(A)}\mathbf{1}[A=a]\varphi_0(Z,A)(Y-h(W,A))^2\mid Z,A\right]\in cl(\mathcal{R}(T)).$$

$\square$

# C  Algorithm derivations

In this section, we derive the doubly robust PCL algorithms, called doubly robust kernel proxy variable (DRKPV) and doubly robust proxy maximum moment restriction (DRPMMR). We begin by reviewing the results from Mastouri et al. [11] and Bozkurt et al. [23], which establish estimation procedures for the outcome and treatment bridge functions: namely, the Kernel Proxy Variable (KPV), Proxy Maximum Moment Restriction (PMMR), and Kernel Alternative Proxy (KAP) methods. Building on these, we introduce fully kernelized doubly robust algorithms for estimating the dose-response curve from observational data, without requiring an explicit density ratio estimation step. This property makes our methods particularly suitable for continuous and high-dimensional treatment settings.

## C.1  Outcome bridge function-based approaches

Two kernel-based methods KPV and PMMR have been proposed in [11] to estimate the outcome bridge function $h:\mathcal{W}\times\mathcal{A}\to\mathbb{R}$, which satisfies

$$\mathbb{E}[Y\mid Z,A]=\int h_0(w,A)p(w\mid Z,A)dw.$$

Once $h_0$ satisfies this equation, the dose-response curve can be identified by $\mathbb{E}[h_0(W,a)]$ [5, 11, 13], as stated in Theorem (2.4). In both approaches proposed by Mastouri et al. [11], $h_0$ is assumed to lie in the RKHS $\mathcal{H}_\mathcal{W}\otimes\mathcal{H}_\mathcal{A}$. Therefore, we note that

$$\theta_{ATE}(a)=\mathbb{E}[h_0(W,a)]=\mathbb{E}\left[\langle h_0,\phi_\mathcal{W}(W)\otimes\phi_\mathcal{A}(a)\rangle_{\mathcal{H}_\mathcal{W}\otimes\mathcal{H}_\mathcal{A}}\right]$$

$$= \langle h_0, \mathbb{E}[\phi_{\mathcal{W}}(W)] \otimes \phi_{\mathcal{A}}(a)\rangle_{\mathcal{H}_{\mathcal{W}} \otimes \mathcal{H}_{\mathcal{A}}} \approx \langle h_0, \hat{\mu}_W \otimes \phi_{\mathcal{A}}(a)\rangle_{\mathcal{H}_{\mathcal{W}} \otimes \mathcal{H}_{\mathcal{A}}},$$

where $\hat{\mu}_W = \frac{1}{t_h}\sum_{i=1}^{n}\phi_{\mathcal{W}}(\dot{w}_i)$ is the sample-based estimate of the mean embedding $\mathbb{E}[\phi_{\mathcal{W}}(W)]$. As a result, by replacing $h_0$ with its approximation $\hat{h}$ in the above expression, the dose-response can be estimated using the inner product:

$$\theta_{ATE}(a) \approx \left\langle \hat{h}, \hat{\mu}_W \otimes \phi_{\mathcal{A}}(a)\right\rangle_{\mathcal{H}_{\mathcal{W}} \otimes \mathcal{H}_{\mathcal{A}}}.$$

In the following, we review the estimation procedures of both KPV and PMMR proposed by Mastouri et al. [11]. Both methods yield closed-form solutions for the estimation of the bridge function and the causal function, which we present in the subsequent subsections.

### C.1.1 Kernel proxy variable algorithm

To solve the outcome bridge function equation in Equation (1), the KPV method aims to minimize the following regularized population loss function, assuming that $h \in \mathcal{H}_{\mathcal{W}\mathcal{A}}$:

$$\mathcal{L}_{\text{KPV}}(h) = \mathbb{E}\left[(Y - \mathbb{E}[h(W, A) \mid Z, A])^2\right] + \lambda_{h,2}\|h\|_{\mathcal{H}_{\mathcal{W}} \otimes \mathcal{H}_{\mathcal{A}}},$$

where $\lambda_{h,2}$ is the regularization constant for the second stage regression of KPV. Since this loss function involves the conditional expectation $\mathbb{E}[h(W, A) \mid Z, A]$, it cannot be directly minimized. To address this, the KPV algorithm decomposes the problem into two-stage regressions. Note that, under Assumption (3.1), the conditional expectation can be rewritten as

$$\mathbb{E}[h(W, A) \mid Z, A] = \mathbb{E}\left[\langle h, \phi_{\mathcal{W}}(W) \otimes \phi_{\mathcal{A}}(A)\rangle \mid Z, A\right]$$
$$= \langle h, \mathbb{E}[\phi_{\mathcal{W}}(W) \mid Z, A] \otimes \phi_{\mathcal{A}}(A)\rangle.$$

Thus, the first step in KPV is to approximate the conditional mean embedding $\mu_{W|Z,A}(z, a) = \mathbb{E}[\phi_{\mathcal{W}}(W) \mid Z = z, A = a]$ using vector-valued kernel ridge regression. In particular, under the regularity condition that $\mathbb{E}[g(W)|Z = \cdot, A = \cdot]$ lies in $\mathcal{H}_{\mathcal{Z}\mathcal{A}}$ for all $g \in \mathcal{H}_{\mathcal{W}}$, there exists an operator $C_{W|Z,A} \in \mathcal{S}_2(\mathcal{H}_{\mathcal{Z}\mathcal{A}}, \mathcal{H}_{\mathcal{W}})$ such that $\mu_{W|Z,A}(z, a) = C_{W|Z,A}(\phi_{\mathcal{Z}}(z) \otimes \phi_{\mathcal{A}}(a))$.

Let $\{\bar{w}_i, \bar{z}_i, \bar{a}_i\}_{i=1}^{n_h}$ be i.i.d. samples from the distribution $p(W, Z, A)$, representing the first-stage regression samples, where $n_h$ is the number of first-stage samples for KPV algorithm. The conditional expectation operator is estimated by minimizing the following regularized least-squares objective:

$$\hat{\mathcal{L}}_{KPV}^c(C) = \frac{1}{n_h}\sum_{i=1}^{n_h}\|\phi_{\mathcal{W}}(\bar{w}_i) - C(\phi_{\mathcal{Z}}(\bar{z}_i) \otimes \phi_{\mathcal{A}}(\bar{a}_i))\| + \lambda_{h,1}\|C\|_{\mathcal{S}_2(\mathcal{H}_{\mathcal{W}}, \mathcal{H}_{\mathcal{Z}\mathcal{A}})}^2,$$

where $\lambda_{h,1}$ is the regularization constant for the first stage regression of KPV. The minimizer $\hat{C}_{W|Z,A}$ has a closed form solution and is given by [11]:

$$\hat{C}_{W|Z,A}(\phi_{\mathcal{Z}}(z) \otimes \phi_{\mathcal{A}}(a)) = \hat{\mu}_{W|Z,A}(z, a) = \sum_{i=1}^{n_h}\boldsymbol{\beta}_i(z, a)\phi_{\mathcal{W}}(\bar{w}_i),$$

where

$$\boldsymbol{\beta}(z, a) = (\boldsymbol{K}_{\bar{Z}\bar{Z}} \odot \boldsymbol{K}_{\bar{A}\bar{A}} + n\lambda_1 \boldsymbol{I})^{-1}(\boldsymbol{K}_{\bar{Z}z} \odot \boldsymbol{K}_{\bar{A}a}).$$

By substituting this approximation into the sample-based version of the loss function $\mathcal{L}_{KPV}(h)$, the KPV algorithm estimates the bridge function $h_0$. Specifically, let $\{\tilde{y}_i, \tilde{z}_i, \tilde{a}_i\}_{i=1}^{m_h}$ be i.i.d. samples from the distribution $p(Y, Z, A)$, denoting the second-stage data, where $m_h$ is the number of second-stage samples for the KPV algorithm. The sample-based KPV objective is then given by

$$\hat{\mathcal{L}}_{KPV}(h) = \frac{1}{m_h}\sum_{i=1}^{m_h}\left(\tilde{y}_i - \langle h, \hat{\mu}_{W|Z,A}(\tilde{z}_i, \tilde{a}_i) \otimes \phi_{\mathcal{A}}(\tilde{a}_i)\rangle\right)^2 + \lambda_{h,2}\|h\|_{\mathcal{H}_{\mathcal{W}\mathcal{A}}}^2. \tag{20}$$

By the representer theorem [44], $h$ admits the representation

$$h = \sum_{i=1}^{n_h}\sum_{j=1}^{m_h}\alpha_{ij}\phi_{\mathcal{W}}(\bar{w}_i) \otimes \phi_{\mathcal{A}}(\tilde{a}_j), \tag{21}$$

for some coefficients $\alpha_{ij}$. Substituting this representation into the sample loss in Equation (20), KPV finds a closed-form solution for the bridge function by minimizing for the coefficients $\alpha_{ij}$. For further details, we refer the reader to [11].

Below, we summarize the pseudo-code for the KPV algorithm. Unlike the original implementation in [11], we use a more numerically stable version that optimizes fewer parameters, as presented in [62] (see Appendix F in [62]). This version of the KPV algorithm optimizes $m_h$ parameters rather than $n_h \times m_h$ parameters.

---

**Algorithm 2** Kernel Proxy Variable Algorithm [11, 62]

---

**Input**: First-stage samples $\{\bar{w}_i, \bar{z}_i, \bar{a}_i\}_{i=1}^{n_h}$, Second-stage samples $\{\tilde{y}_i, \tilde{z}_i, \tilde{a}_i\}_{i=1}^{m_h}$,
**Parameters:** Regularization parameters $\lambda_{h,1}$ and $\lambda_{h,2}$, and kernel functions $k_{\mathcal{F}}(\cdot, \cdot)$ for each $\mathcal{F} \in \{\mathcal{A}, \mathcal{Z}, \mathcal{W}\}$.
**Output**: Bridge function estimation $\hat{h}$.

1: For variables $F \in \{W, Z, A\}$ with domain $\mathcal{F}$, define the (required) first and second-stage kernel matrices/vectors as follows:

$$\boldsymbol{K}_{\bar{F}\bar{F}} = [k_{\mathcal{F}}(\bar{f}_i, \bar{f}_j)]_{ij} \in \mathbb{R}^{n_h \times n_h}, \quad \boldsymbol{K}_{\bar{F}\tilde{F}} = [k_{\mathcal{F}}(\bar{f}_i, \tilde{f}_j)]_{ij} \in \mathbb{R}^{n_h \times m_h},$$
$$\boldsymbol{K}_{\tilde{F}\tilde{F}} = [k_{\mathcal{F}}(\tilde{f}_i, \tilde{f}_j)]_{ij} \in \mathbb{R}^{m_h \times m_h}, \boldsymbol{K}_{\bar{F}f} = [k_{\mathcal{F}}(\bar{f}_i, f)]_i \in \mathbb{R}^{n_h},$$
$$\boldsymbol{K}_{\tilde{F}f} = [k_{\mathcal{F}}(\tilde{f}_j, f)]_j \in \mathbb{R}^{m_h}.$$

2: Define the following matrices and coefficient vector:

$$\boldsymbol{B} = (\boldsymbol{K}_{\bar{A}\bar{A}} \odot \boldsymbol{K}_{\bar{Z}\bar{Z}} + n_h \lambda_{h,1} \boldsymbol{I})^{-1} (\boldsymbol{K}_{\bar{A}\tilde{A}} \odot \boldsymbol{K}_{\bar{Z}\tilde{Z}}),$$
$$\boldsymbol{M} = \boldsymbol{K}_{\tilde{A}\tilde{A}} \odot \left(\boldsymbol{B}^{\top} \boldsymbol{K}_{\bar{W}\bar{W}} \boldsymbol{B}\right),$$
$$\boldsymbol{\alpha} = (\boldsymbol{M} + m_h \lambda_{h,2} \boldsymbol{I})^{-1} \tilde{\boldsymbol{Y}}.$$

3: The bridge function estimation is given by $\hat{h}(w, a) = \boldsymbol{\alpha}^{\top} \left(\boldsymbol{K}_{\tilde{A}a} \odot \left(\boldsymbol{B}^{\top} \boldsymbol{K}_{\bar{W}w}\right)\right).$
4: The dose-response estimation is given by

$$\hat{\theta}_{ATE}(a) = \sum_{i=1}^{t} \hat{h}(\dot{w}_i, a) = \sum_{i=1}^{t} \boldsymbol{\alpha}^{\top} \left(\boldsymbol{K}_{\tilde{A}a} \odot \left(\boldsymbol{B}^{\top} \boldsymbol{K}_{\bar{W}\dot{w}_i}\right)\right)$$

where we take $\{\dot{w}_i\}_{i=1}^{t_h} = \{\bar{w}_i\}_{i=1}^{n_h} \cup \{\tilde{w}_i\}_{i=1}^{m_h}$

---

#### C.1.2 Proximal maximum moment restriction algorithm

The PMMR algorithm relies on the following conditional moment restriction result:

**Lemma C.1** (Lemma (1) in [11]). *A measurable function $h$ on the domain $\mathcal{W} \times \mathcal{A}$ solves*

$$\mathbb{E}[Y|Z = z, A = a] = \int_{\mathcal{W}} h(w, a)p(w|z, a)dw$$

*if and only if it satisfies the conditional moment restriction (CMR):* $\mathbb{E}[Y - h(W, A)|Z, A] = 0$.

Lemma (C.1) implies that $\mathbb{E}[(Y - h(W, A))g(Z, A)] = 0$ for all measurable $g$ defined on the domain $\mathcal{Z} \times \mathcal{A}$. Thus, to solve the CMR, the PMMR algorithm solves the CMR by minimizing the regularized maximum moment restriction (MMR) objective to find the function $h \in \mathcal{H}_{\mathcal{W}\mathcal{A}}$:

$$\mathcal{L}_{\text{MMR}}(h) = \sup_{\substack{g \in \mathcal{H}_{\mathcal{Z}\mathcal{A}} \\ \|g\| \le 1}} \mathbb{E}[(Y - h(W, A))g(Z, A)]^2 + \lambda_{\text{MMR}}\|h\|_{\mathcal{H}_{\mathcal{W}\mathcal{A}}}. \tag{22}$$

Under the integrability condition $\mathbb{E}\left[(Y - h(W, A))^2 \langle \phi_{\mathcal{Z}\mathcal{A}}(Z, A), \phi_{\mathcal{Z}\mathcal{A}}(Z, A)\rangle_{\mathcal{H}_{\mathcal{Z}\mathcal{A}}}\right] < \infty$, Mastouri et al. [11, Lemma (2)] shows that the regularized MMR objective admits the equivalent form:

$$\mathcal{L}_{\text{MMR}}(h) = \mathbb{E}\left[(Y - h(W, A))(Y' - h(W', A'))\langle \phi_{\mathcal{Z}\mathcal{A}}(Z, A), \phi_{\mathcal{Z}\mathcal{A}}(Z', A')\rangle_{\mathcal{H}_{\mathcal{Z}\mathcal{A}}}\right] + \lambda_{\text{MMR}}\|h\|_{\mathcal{H}_{\mathcal{W}\mathcal{A}}}, \tag{23}$$

where $V'$ denotes an independent copy of $V \in A, Z, W$, and $\phi_{\mathcal{Z}\mathcal{A}}$ is the canonical feature map of $\mathcal{H}_{\mathcal{Z}\mathcal{A}}$. In this paper, we adopt the tensor product structure for the feature map and RKHS by defining

$\phi_{\mathcal{Z}\mathcal{A}}(z,a) = \phi_{\mathcal{Z}}(z) \otimes \phi_{\mathcal{A}}(a)$ and $\mathcal{H}_{\mathcal{Z}\mathcal{A}} = \mathcal{H}_{\mathcal{Z}} \otimes \mathcal{H}_{\mathcal{A}}$. This choice allows us to construct $\mathcal{H}_{\mathcal{Z}\mathcal{A}}$ from separate kernels over $\mathcal{Z}$ and $\mathcal{A}$. However, it is also possible to define a kernel directly on the joint domain $\mathcal{Z} \times \mathcal{A}$ for PMMR algorithm, in which case the feature map and RKHS do not need to take the form of a tensor product.

As a result, given training samples $\{y_i, w_i, z_i, a_i\}_{i=1}^t$, the empirical objective for PMMR is defined as

$$\hat{\mathcal{L}}_{\text{MMR}}(h) = \frac{1}{t^2} \sum_{i=1}^t \sum_{j=1}^t (y_i - h_i)(y_j - h_j) k_{ij} + \lambda_{\text{MMR}} \|h\|_{\mathcal{H}_{\mathcal{W}\mathcal{A}}}, \tag{24}$$

where $h_i = h(w_i, a_i)$ and $k_{ij} = \langle \phi_{\mathcal{Z}\mathcal{A}}(z_i, a_i), \phi_{\mathcal{Z}\mathcal{A}}(z_j, a_j) \rangle_{\mathcal{H}_{\mathcal{Z}\mathcal{A}}}$. Recall that we assume $h \in \mathcal{H}_{\mathcal{W}} \otimes \mathcal{H}_{\mathcal{A}}$. By the representer theorem [44], the solution to Equation (24) admits the following representation:

$$h = \sum_{i=1}^t \alpha_i \phi_{\mathcal{W}}(w_i) \otimes \phi_{\mathcal{A}}(a_i).$$

Substituting this representation into the sample loss in Equation (24), the empirical objective becomes

$$\hat{\boldsymbol{\alpha}} = \arg\min_{\boldsymbol{\alpha}} \frac{1}{t^2} (\mathbf{Y} - \boldsymbol{L}\boldsymbol{\alpha})^\top \boldsymbol{W} (\mathbf{Y} - \boldsymbol{L}\boldsymbol{\alpha}) + \lambda_{\text{MMR}} \boldsymbol{\alpha}^\top \boldsymbol{L}\boldsymbol{\alpha},$$

where $\mathbf{Y} = [y_1 \quad y_2 \quad \dots \quad y_t]^\top$, $\boldsymbol{\alpha} = [\alpha_1 \quad \alpha_2 \quad \dots \quad \alpha_t]^\top$, $[\boldsymbol{L}]_{ij} = k_{\mathcal{W}}(w_i, w_j) k_{\mathcal{A}}(a_i, a_j)$, and $[\boldsymbol{W}]_{ij} = k_{\mathcal{Z}}(z_i, z_j) k_{\mathcal{Z}}(a_i, a_j)$. The PMMR algorithm solves for the coefficient vector $\boldsymbol{\alpha}$. We summarize the procedure in Algorithm (3), following the variant presented by Xu and Gretton [62].

---

**Algorithm 3** Proximal Maximum Moment Restriction Algorithm [11, 62]

---

**Input**: Training samples $\{w_i, z_i, a_i, y_i\}_{i=1}^t$,
**Parameters**: Regularization parameter $\lambda_{\text{MMR}}$, and kernel functions $k_{\mathcal{F}}(\cdot, \cdot)$ for each $\mathcal{F} \in \{\mathcal{A}, \mathcal{Z}, \mathcal{W}\}$.
**Output**: Bridge function estimation $\hat{h}$.
1: For variables $F \in \{W, Z, A\}$ with domain $\mathcal{F}$, define the (required) kernel matrices/vectors as follows:

$$\boldsymbol{K}_{FF} = [k_{\mathcal{F}}(f_i, f_j)]_{ij} \in \mathbb{R}^{t \times t}, \quad \boldsymbol{K}_{Ff} = [k_{\mathcal{F}}(f_i, f)]_i \in \mathbb{R}^t.$$

2: Construct the following kernel matrices and the coefficient vector:

$$\boldsymbol{L} = \boldsymbol{K}_{AA} \odot \boldsymbol{K}_{WW}, \quad \boldsymbol{G} = \boldsymbol{K}_{AA} \odot \boldsymbol{K}_{ZZ}, \quad \boldsymbol{\alpha} = \sqrt{\boldsymbol{G}} \left( \sqrt{\boldsymbol{G}} \boldsymbol{L} \sqrt{\boldsymbol{G}} + t\lambda_{\text{MMR}} \boldsymbol{I} \right)^{-1} \sqrt{\boldsymbol{G}} \mathbf{Y},$$

where $\sqrt{\boldsymbol{G}}$ is the square root of the matrix $\boldsymbol{G}$.
3: The bridge function estimation is given by $\hat{h}(w, a) = \boldsymbol{\alpha}^\top (\boldsymbol{K}_{Aa} \odot \boldsymbol{K}_{Ww})$.
4: The dose-response estimation is given by $\hat{\theta}_{ATE}(a) \approx \sum_{i=1}^t \hat{h}(w_i, a) = \sum_{i=1}^t \boldsymbol{\alpha}^\top (\boldsymbol{K}_{Aa} \odot \boldsymbol{K}_{Ww_i})$.

---

### C.2 Treatment bridge function-based algorithm

An alternative identification result in the proxy causal learning setting, based on density ratios, is presented in [23]. Unlike previous works [21, 22], this approach bypasses the explicit density ratio estimation step by leveraging a novel bridge function definition. Specifically, [23] proposes Kernel Alternative Proxy (KAP) algorithm, a two-stage regression method to estimate the treatment bridge function $\varphi_0$ satisfying

$$\mathbb{E}\left[\varphi_0(Z, a) \mid W, A = a\right] = \frac{p(W)p(a)}{p(W, a)}.$$

This new approach aims to minimize the following regularized population loss function, assuming that $\varphi \in \mathcal{H}_{\mathcal{Z}\mathcal{A}}$

$$\mathcal{L}_{\text{KAP}}(\varphi) = \mathbb{E}\left[(r(W, A) - \mathbb{E}\left[\varphi(Z, A) \mid W, A\right])^2\right] + \lambda_{\varphi, 2} \|\varphi\|_{\mathcal{H}_{\mathcal{Z}\mathcal{A}}}$$

$$= \mathbb{E}\left[\mathbb{E}\left[\varphi(Z, A) \mid W, A\right]^2\right] - 2\mathbb{E}_W\mathbb{E}_A\left[\mathbb{E}\left[\varphi(Z, A) \mid W, A\right]\right] + \lambda_{\varphi,2}\|\varphi\|_{\mathcal{H}_{\mathcal{Z}\mathcal{A}}} + \text{const.,} \tag{25}$$

where $\lambda_{\varphi,2}$ is the regularization parameter for the second stage regression, the notation $\mathbb{E}_W\mathbb{E}_A[.]$ denotes the decoupled expectation taken over $W$ and $A$, i.e., for any function $\ell$, $\mathbb{E}_W\mathbb{E}_A[\ell(W, A)] = \int \ell(w, a)p(w)p(a)dwda$, and $'\text{const.}'$ represents the terms independent of $\varphi$. As in the outcome bridge function estimation, this loss function cannot be directly minimized due to the presence of the conditional expectation term $\mathbb{E}\left[\varphi(Z, A) \mid W, A\right]$. Notably, appealing to Assumption (3.1), this expectation can be rewritten as

$$\mathbb{E}\left[\varphi(Z, A) \mid W, A\right] = \mathbb{E}\left[\langle\varphi, \phi_{\mathcal{Z}}(Z) \otimes \phi_{\mathcal{A}}(A)\rangle_{\mathcal{H}_{\mathcal{Z}\mathcal{A}}} \mid W, A\right]$$
$$= \langle\varphi, \mathbb{E}[\phi_{\mathcal{Z}}(Z) \mid W, A] \otimes \phi_{\mathcal{A}}(A)\rangle_{\mathcal{H}_{\mathcal{Z}\mathcal{A}}}$$

Thus, the first step in KAP method is to estimate the conditional mean embedding $\mu_{Z|W,A}(w, a) = \mathbb{E}[\phi_{\mathcal{Z}}(Z) \mid W = w, A = a]$ using vector valued kernel ridge regression. Specifically, under the regularity condition that $\mathbb{E}[g(Z)|W = \cdot, A = \cdot]$ lies in $\mathcal{H}_{\mathcal{W}\mathcal{A}}$ for all $g \in \mathcal{H}_{\mathcal{Z}}$, there exists an operator $C_{Z|W,A} \in \mathcal{S}_2(\mathcal{H}_{\mathcal{W}\mathcal{A}}, \mathcal{Z})$ such that $\mu_{Z|W,A}(w, a) = C_{Z|W,A}(\phi_{\mathcal{W}}(w) \otimes \phi_{\mathcal{A}}(a))$. Let $\{\bar{w}_i, \bar{z}_i, \bar{a}_i\}_{i=1}^{n_\varphi}$ be i.i.d. samples from the distribution $p(W, Z, A)$, denoting the first-stage samples, where $n_\varphi$ is the number of first-stage samples for KAP algorithm. The conditional expectation operator is learned by minimizing the regularized least-squares function:

$$\hat{\mathcal{L}}^c(C)_{\text{KAP}} = \frac{1}{n}\sum_{i=1}^{n}\|\phi_{\mathcal{Z}}(\bar{z}_i) - C\left(\phi_{\mathcal{W}}(\bar{w}_i) \otimes \phi_{\mathcal{A}}(\bar{a}_i)\right)\| + \lambda_{\varphi,1}\|C\|^2_{\mathcal{S}_2(\mathcal{H}_{\mathcal{W}\mathcal{A}},\mathcal{H}_{\mathcal{Z}})},$$

where $\lambda_{\varphi,1}$ is the regularization parameter for first-stage regression of KAP method. The minimizer $\hat{C}_{Z|W,A}$ admits a closed form solution and is given by [23]:

$$\hat{\mu}_{Z|W,A}(w, a) = \hat{C}_{Z|W,A}\left(\phi_{\mathcal{Z}}(z) \otimes \phi_{\mathcal{A}}(a)\right) = \sum_{i=1}^{n_\varphi}\boldsymbol{\beta}_i(w, a)\phi_{\mathcal{Z}}(\bar{z}_i),$$

where

$$\boldsymbol{\beta}(w, a) = \left(\boldsymbol{K}_{\bar{W}\bar{W}} \odot \boldsymbol{K}_{\bar{A}\bar{A}} + n_\varphi\lambda_{\varphi,1}\boldsymbol{I}\right)^{-1}\left(\boldsymbol{K}_{\bar{W}w} \odot \boldsymbol{K}_{\bar{A}a}\right).$$

By substituting this approximation into the sample-based version of the loss function $\mathcal{L}_{\text{KAP}}(\varphi)$, Bozkurt et al. [23] derives an algorithm to estimate the bridge function $\varphi_0$. Let $\{\tilde{w}_i, \tilde{a}_i\}_{i=1}^{m_\varphi}$ be i.i.d. samples from the distribution $p(W, A)$, denoting the second-stage samples, where $m_\varphi$ is the number of samples for the second-stage regression of KAP algorithm. The sample-based version of the loss function in Equation (25) can be written as

$$\hat{\mathcal{L}}(\varphi)_{\text{KAP}} = \frac{1}{m_\varphi}\sum_{i=1}^{m_\varphi}\langle\varphi, \hat{\mu}_{Z|W,A}(\tilde{w}_i, \tilde{a}_i) \otimes \phi_{\mathcal{A}}(\tilde{a}_i)\rangle^2_{\mathcal{H}_{\mathcal{Z}\mathcal{A}}}$$

$$- 2\frac{1}{m_\varphi(m_\varphi - 1)}\sum_{i=1}^{m_\varphi}\sum_{\substack{j=1 \\ j \neq i}}^{m_\varphi}\left\langle\varphi, \hat{\mu}_{Z|W,A}(\tilde{w}_j, \tilde{a}_i) \otimes \phi_{\mathcal{A}}(\tilde{a}_i)\right\rangle_{\mathcal{H}_{\mathcal{Z}\mathcal{A}}} + \lambda_{\varphi,2}\|\varphi\|^2_{\mathcal{H}_{\mathcal{Z}\mathcal{A}}} \tag{26}$$

The representer theorem [44] ensures that $\varphi$ admits the representation

$$\varphi = \sum_{i=1}^{m_\varphi}\gamma_i\hat{\mu}_{Z|W,A}(\tilde{w}_i, \tilde{a}_i) \otimes \phi_{\mathcal{A}}(\tilde{a}_i) + \frac{\gamma_{m_\varphi+1}}{m_\varphi(m_\varphi - 1)}\sum_{j=1}^{m_\varphi}\sum_{\substack{l=1 \\ l \neq j}}^{m_\varphi}\hat{\mu}_{Z|W,A}(\tilde{w}_l, \tilde{a}_j) \otimes \phi_{\mathcal{A}}(\tilde{a}_j)$$

for some coefficients $\{\gamma_i\}_{i=1}^{m_\varphi+1}$. By substituting this representation into the sample loss in Equation (26), the minimizer $\hat{\varphi}$ admits a closed-form solution, as given in Bozkurt et al. [23], which we outline in Algorithm (4). Furthermore, the dose-response curve can be estimated via $\theta_{ATE}(a) \approx \mathbb{E}[Y\hat{\varphi}(Z, a)|A = a]$. Noting that

$$\mathbb{E}[Y\hat{\varphi}(Z, a)|A = a] = \mathbb{E}\left[Y\langle\hat{\varphi}, \phi_{\mathcal{Z}}(Z) \otimes \phi_{\mathcal{A}}(a)\rangle_{\mathcal{H}_{\mathcal{Z}\mathcal{A}}} \mid A = a\right]$$
$$= \langle\hat{\varphi}, \mathbb{E}[Y\phi_{\mathcal{Z}}(Z) \mid A = a] \otimes \phi_{\mathcal{A}}(a)\rangle_{\mathcal{H}_{\mathcal{Z}\mathcal{A}}}. \tag{27}$$

evaluating the above inner product requires estimating the conditional mean embedding $\mu_{YZ|A}(a) = \mathbb{E}[Y\phi_{\mathcal{Z}}(Z) \mid A = a]$, which can be obtained via vector-valued kernel ridge regression, similar to the first-stage regression. In particular, let $\{\dot{y}_i, \dot{z}_i, \dot{a}_i\}_{i=1}^{t_\varphi}$ denote i.i.d. samples from the distribution $p(Y, Z, A)$, denoting the third-stage samples for KAP algorithm which we take to be $\{\dot{y}_i, \dot{z}_i, \dot{a}_i\}_{i=1}^{t_\varphi} = \{\bar{y}_i, \bar{z}_i, \bar{a}_i\}_{i=1}^{n_\varphi} \cup \{\tilde{y}_i, \tilde{z}_i, \tilde{a}_i\}_{i=1}^{m_\varphi}$. Then, the conditional mean embedding $\mu_{YZ|A}(a) = C_{YZ|A}\phi_{\mathcal{A}}(a)$ is learned by minimizing the regularized least squares objective:

$$\mathcal{L}_{\text{KAP}}^{c_{yz|a}}(C) = \frac{1}{t}\sum_{i=1}^{t}\|\dot{y}_i\phi_{\mathcal{Z}}(\dot{z}_i) - C\phi_{\mathcal{A}}(\dot{a}_i)\|_{\mathcal{H}_{\mathcal{Z}}}^2 + \lambda_{\varphi,3}\|C\|_{S_2(\mathcal{H}_{\mathcal{A}},\mathcal{H}_{\mathcal{Z}})}^2,$$

where $\lambda_{\varphi,3}$ denotes the regularization parameter for the third-stage regression of KAP method. The minimizer is given by

$$\hat{\mu}_{YZ|A}(a) = \hat{C}_{YZ|A}\phi_{\mathcal{A}}(a) = \dot{\Phi}_{\mathcal{Z}}\text{diag}(\dot{\boldsymbol{Y}})[\boldsymbol{K}_{\dot{A}\dot{A}} + n\lambda_{\varphi,3}\boldsymbol{I}]^{-1}\boldsymbol{K}_{\dot{A}a},$$

where $\dot{\Phi}_{\mathcal{Z}} = [\phi_{\mathcal{Z}}(\dot{z}_1) \quad \ldots \quad \phi_{\mathcal{Z}}(\dot{z}_t)]$, $\dot{\boldsymbol{Y}} = [\dot{y}_1 \quad \ldots \quad \dot{y}_t]^T$, $\boldsymbol{K}_{\dot{A}\dot{A}} \in \mathbb{R}^{t_\varphi \times t_\varphi}$ is the kernel matrix over $\{\dot{a}_i\}_{i=1}^{t_\varphi}$, and $\boldsymbol{K}_{\dot{A}a}$ 's the kernel vector between the training points $\{\dot{a}_i\}_{i=1}^{t_\varphi}$ and the target treatment $a$. As a result, by substituting the estimation $\hat{\mu}_{YZ|A}(a)$ in Equation (27), the dose-response curve can be estimated with a closed-form solution that is efficiently implemented via matrix operations [23], as summarized below in Algorithm (4). For further details of this algorithm, we refer the reader to [23].

---

**Algorithm 4** Alternative Kernel Proxy Variable Algorithm [23]

---

**Input**: First-stage, second-stage, and third-stage: $\{\bar{w}_i, \bar{z}_i, \bar{a}_i\}_{i=1}^{n_\varphi}$, $\{\tilde{y}_i, \tilde{z}_i, \tilde{a}_i\}_{i=1}^{m_\varphi}$, $\{\dot{y}_i, \dot{z}_i, \dot{a}_i\}_{i=1}^{t_\varphi}$,

**Parameters:** Regularization parameters $\lambda_{\varphi,1}, \lambda_{\varphi,2}$, and $\lambda_{\varphi,3}$, and kernel functions $k_\mathcal{F}(\cdot, \cdot)$ for each $\mathcal{F} \in \{\mathcal{A}, \mathcal{Z}, \mathcal{W}\}$.

**Output**: Bridge function and dose response estimations: $\hat{\varphi}(\cdot, \cdot), \hat{\theta}_{\text{ATE}}(\cdot)$.

1: For variables $F \in \{W, Z, A\}$ with domain $\mathcal{F}$, define the (required) first- and second- and third-stage kernel matrices/vectors as follows:

$$\boldsymbol{K}_{\bar{F}\bar{F}} = [k_\mathcal{F}(\bar{f}_i, \bar{f}_j)]_{ij} \in \mathbb{R}^{n_\varphi \times n_\varphi}, \quad \boldsymbol{K}_{\bar{F}\tilde{F}} = [k_\mathcal{F}(\bar{f}_i, \tilde{f}_j)]_{ij} \in \mathbb{R}^{n_\varphi \times m_\varphi},$$

$$\boldsymbol{K}_{\tilde{F}\tilde{F}} = [k_\mathcal{F}(\tilde{f}_i, \tilde{f}_j)]_{ij} \in \mathbb{R}^{m_\varphi \times m_\varphi}, \quad \boldsymbol{K}_{\bar{F}f} = [k_\mathcal{F}(\bar{f}_i, f)]_i \in \mathbb{R}^{n_\varphi},$$

$$\boldsymbol{K}_{\tilde{F}f} = [k_\mathcal{F}(\tilde{f}_j, f)]_j \in \mathbb{R}^{m_\varphi}, \quad \boldsymbol{K}_{\dot{F}\dot{F}} = [k_\mathcal{F}(\dot{f}_i, \dot{f}_j)]_{ij} \in \mathbb{R}^{t_\varphi \times t_\varphi},$$

$$\boldsymbol{K}_{\bar{F}\dot{F}} = [k_\mathcal{F}(\bar{f}_i, \dot{f}_j)]_{ij} \in \mathbb{R}^{n_\varphi \times t_\varphi}, \quad \boldsymbol{K}_{\dot{F}f} = [k_\mathcal{F}(\dot{f}_i, f)] \in \mathbb{R}^{t_\varphi}$$

2: Define the following matrices:

$$\boldsymbol{B} = (\boldsymbol{K}_{\bar{W}\bar{W}} \odot \boldsymbol{K}_{\bar{A}\bar{A}} + n_\varphi \lambda_{\varphi,1} \boldsymbol{I})^{-1} (\boldsymbol{K}_{\bar{W}\tilde{W}} \odot \boldsymbol{K}_{\bar{A}\tilde{A}}),$$

$$\tilde{\boldsymbol{B}} \in \mathbb{R}^{m_\varphi \times n_\varphi} \text{ where } \tilde{\boldsymbol{B}}_{:,j} = \frac{1}{m-1} \sum_{\substack{l=1 \\ l \neq j}}^{m} (\boldsymbol{K}_{\bar{W}\bar{W}} \odot \boldsymbol{K}_{\bar{A}\bar{A}} + n_\varphi \lambda_{\varphi,1} \boldsymbol{I})^{-1} (\boldsymbol{K}_{\bar{W}\tilde{w}_l} \odot \boldsymbol{K}_{\bar{A}\tilde{a}_j})$$

$$\boldsymbol{L} = \begin{bmatrix} \boldsymbol{B}^T \boldsymbol{K}_{\bar{Z}\bar{Z}} \boldsymbol{B} \odot \boldsymbol{K}_{\tilde{A}\tilde{A}} \\ (\frac{1}{m_\varphi})^T [\boldsymbol{B}^T \boldsymbol{K}_{\bar{Z}\bar{Z}} \tilde{\boldsymbol{B}} \odot \boldsymbol{K}_{\tilde{A}\tilde{A}}]^T \end{bmatrix}^T \in \mathbb{R}^{m_\varphi \times (m_\varphi+1)},$$

$$\boldsymbol{M} = \begin{bmatrix} [\boldsymbol{B}^T \boldsymbol{K}_{\bar{Z}\bar{Z}} \tilde{\boldsymbol{B}} \odot \boldsymbol{K}_{\tilde{A}\tilde{A}}] \frac{1}{m_\varphi} \\ (\frac{1}{m_\varphi})^T [\tilde{\boldsymbol{B}}^T \boldsymbol{K}_{\bar{Z}\bar{Z}} \tilde{\boldsymbol{B}} \odot \boldsymbol{K}_{\tilde{A}\tilde{A}}] \frac{1}{m_\varphi} \end{bmatrix} \in \mathbb{R}^{(m_\varphi+1)},$$

$$\boldsymbol{N} = \begin{bmatrix} \boldsymbol{L} & \boldsymbol{M} \end{bmatrix} \in \mathbb{R}^{(m_\varphi+1) \times (m_\varphi+1)},$$

$$\boldsymbol{\gamma} = \left(\frac{1}{m_\varphi} \boldsymbol{L}^\top \boldsymbol{L} + \lambda_{\varphi,2} \boldsymbol{N}\right)^{-1} \boldsymbol{M} \in \mathbb{R}^{m_\varphi+1}$$

3: The bridge function estimation is given by $\hat{\varphi}(z, a) = \boldsymbol{\gamma}_{1:m_\varphi}^T [(\boldsymbol{B}^T \boldsymbol{K}_{\bar{Z}z}) \odot \boldsymbol{K}_{\tilde{A}a}] + \boldsymbol{\gamma}_{m_\varphi+1} \left(\frac{1}{m_\varphi}\right)^T [(\tilde{\boldsymbol{B}}^T \boldsymbol{K}_{\bar{Z}z}) \odot \boldsymbol{K}_{\tilde{A}a}]$

4: The dose response estimation is given by

$$\hat{\theta}_{ATE}(a) = \boldsymbol{\gamma}_{1:m}^T \left(\boldsymbol{B}^T \left(\boldsymbol{K}_{\bar{Z}\dot{Z}} \text{diag}(\dot{\boldsymbol{Y}}) [\boldsymbol{K}_{\dot{A}\dot{A}} + t_\varphi \lambda_{\varphi,3} \boldsymbol{I}]^{-1} \boldsymbol{K}_{\dot{A}a}\right) \odot \boldsymbol{K}_{\tilde{A}a}\right)$$

$$+ \boldsymbol{\gamma}_{m+1} \left(\tilde{\boldsymbol{B}}^T \left(\boldsymbol{K}_{\bar{Z}\dot{Z}} \text{diag}(\dot{\boldsymbol{Y}}) [\boldsymbol{K}_{\dot{A}\dot{A}} + t_\varphi \lambda_{\varphi,3} \boldsymbol{I}]^{-1} \boldsymbol{K}_{\dot{A}a}\right) \odot \boldsymbol{K}_{\tilde{A}a}\right) \frac{1}{m_\varphi}$$

---

## C.3 Doubly robust kernel proxy variable algorithm

Armed with the estimation procedures for the outcome and treatment bridge functions, we now present the doubly robust estimation algorithm for the dose-response curve. As discussed in Section (3.2), we need to derive an algorithm to estimate the term $\mathbb{E}[\varphi_0(Z, a) h_0(W, a) \mid A = a]$. Let $\{z_i, w_i, a_i\}_i^t \subset \mathcal{D}$ denote i.i.d. observations from the distribution $\mathbb{P}(Z, W, A)$. We observe the following

$$\mathbb{E}[\varphi_0(Z, a) h_0(W, a) \mid A = a] \approx \mathbb{E}[\hat{\varphi}(Z, a) \hat{h}(W, a) \mid A = a]$$

$$= \mathbb{E}\left[\left\langle \hat{\varphi}, \phi_\mathcal{Z}(Z) \otimes \phi_\mathcal{A}(a) \right\rangle_{\mathcal{H}_\mathcal{Z} \otimes \mathcal{H}_\mathcal{A}} \left\langle \hat{h}, \phi_\mathcal{W}(W) \otimes \phi_\mathcal{A}(a) \right\rangle_{\mathcal{H}_\mathcal{W} \otimes \mathcal{H}_\mathcal{A}}\right]$$

$$= \mathbb{E}\left[\left\langle \hat{\varphi} \otimes \hat{h}, (\phi_\mathcal{Z}(Z) \otimes \phi_\mathcal{A}(a)) \otimes (\phi_\mathcal{W}(W) \otimes \phi_\mathcal{A}(a)) \right\rangle_{\mathcal{H}_\mathcal{Z} \otimes \mathcal{H}_\mathcal{A} \otimes \mathcal{H}_\mathcal{W} \otimes \mathcal{H}_\mathcal{A}}\right].$$

As detailed in Section (S.M. C.1.1), the outcome bridge function in the KPV algorithm has the following representation (see Appendix B.3 in [11]):

$$\hat{h} = \sum_i^{n_h} \sum_j^{m_h} \beta_{ij} \phi_\mathcal{W}(\bar{w}_i) \otimes \phi_\mathcal{A}(\tilde{a}_j)$$

where the set of variables $\bar{w}_i$ and $\tilde{a}_j$ denote the samples from first and second stage regressions of KPV algorithm, respectively. Furthermore, as detailed in Appendix (C.2), the treatment bridge function admits the representation

$$\hat{\varphi} = \sum_{i=1}^{m_\varphi} \gamma_i \hat{\mu}_{Z|W,A}(\tilde{w}_i, \tilde{a}_i) \otimes \phi_{\mathcal{A}}(\tilde{a}_i) + \frac{\gamma_{m_\varphi+1}}{m_\varphi(m_\varphi-1)} \sum_{j=1}^{m_\varphi} \sum_{\substack{l=1 \\ l \neq j}}^{m_\varphi} \hat{\mu}_{Z|W,A}(\tilde{w}_l, \tilde{a}_j) \otimes \phi_{\mathcal{A}}(\tilde{a}_j),$$

which we can write without loss of generality as

$$\hat{\varphi} = \sum_{l=1}^{m_\varphi} \sum_{t=1}^{m_\varphi} \gamma_{lt} \hat{\mu}_{Z|W,A}(\tilde{w}_t, \tilde{a}_l) \otimes \phi_{\mathcal{A}}(\tilde{a}_l)$$

by appropriately choosing the coefficients $\gamma_{lt}$ from the set $\{\alpha_i\}_{i=1}^{m+1} \cup \{0\}$. Next, we notice the following

$$\mathbb{E}[\hat{\varphi}(Z,a)\hat{h}(W,a) \mid A = a] =$$

$$= \mathbb{E}\bigg[\bigg\langle \bigg(\sum_l \sum_t \gamma_{lt} \hat{\mu}_{Z|W,A}(\tilde{w}_t, \tilde{a}_l) \otimes \phi_{\mathcal{A}}(\tilde{a}_l)\bigg) \otimes \bigg(\sum_i \sum_j \beta_{ij} \phi_{\mathcal{W}}(\bar{w}_i) \otimes \phi_{\mathcal{A}}(\tilde{a}_j)\bigg)$$

$$, (\phi_{\mathcal{Z}}(Z) \otimes \phi_{\mathcal{A}}(a)) \otimes (\phi_{\mathcal{W}}(W) \otimes \phi_{\mathcal{A}}(a)) \bigg\rangle_{\mathcal{H}_{\mathcal{Z}\mathcal{A}} \otimes \mathcal{H}_{\mathcal{W}\mathcal{A}}} \bigg| A = a\bigg]$$

$$= \sum_{i,j} \sum_{l,t} \alpha_{ij} \gamma_{lt} \mathbb{E}\bigg[\bigg\langle \hat{\mu}_{Z|W,A}(\tilde{w}_t, \tilde{a}_l) \otimes \phi_{\mathcal{A}}(\tilde{a}_l) \otimes \phi_{\mathcal{W}}(\bar{w}_i) \otimes \phi_{\mathcal{A}}(\tilde{a}_j)$$

$$, \phi_{\mathcal{Z}}(Z) \otimes \phi_{\mathcal{A}}(a) \otimes \phi_{\mathcal{W}}(W) \otimes \phi_{\mathcal{A}}(a) \bigg\rangle_{\mathcal{H}_{\mathcal{Z}\mathcal{A}} \otimes \mathcal{H}_{\mathcal{W}\mathcal{A}}} \bigg| A = a\bigg]$$

$$= \sum_{i,j} \sum_{l,t} \alpha_{ij} \gamma_{lt} \mathbb{E}\bigg[\bigg\langle \hat{\mu}_{Z|W,A}(\tilde{w}_t, \tilde{a}_l), \phi_{\mathcal{Z}}(Z)\bigg\rangle_{\mathcal{H}_{\mathcal{Z}}} \bigg\langle \phi_{\mathcal{A}}(\tilde{a}_l), \phi_{\mathcal{A}}(a)\bigg\rangle_{\mathcal{H}_{\mathcal{A}}} \cdot$$

$$\bigg\langle \phi_{\mathcal{W}}(\bar{w}_i), \phi_{\mathcal{W}}(W)\bigg\rangle_{\mathcal{H}_{\mathcal{W}}} \bigg\langle \phi_{\mathcal{A}}(\tilde{a}_j), \phi_{\mathcal{A}}(a)\bigg\rangle_{\mathcal{H}_{\mathcal{A}}} \bigg| A = a\bigg]$$

$$= \sum_{i,j} \sum_{l,t} \alpha_{ij} \gamma_{lt} \mathbb{E}\bigg[\bigg\langle \hat{\mu}_{Z|W,A}(\tilde{w}_t, \tilde{a}_l) \otimes \phi_{\mathcal{W}}(\bar{w}_i), \phi_{\mathcal{Z}}(Z) \otimes \phi_{\mathcal{W}}(W)\bigg\rangle_{\mathcal{H}_{\mathcal{Z}\mathcal{W}}} \cdot$$

$$\bigg\langle \phi_{\mathcal{A}}(\tilde{a}_l) \otimes \phi_{\mathcal{A}}(\tilde{a}_j), \phi_{\mathcal{A}}(a) \otimes \phi_{\mathcal{A}}(a)\bigg\rangle_{\mathcal{H}_{\mathcal{A}\mathcal{A}}} \bigg| A = a\bigg]$$

$$= \sum_{i,j} \sum_{l,t} \alpha_{ij} \gamma_{lt} \bigg\langle \hat{\mu}_{Z|W,A}(\tilde{w}_t, \tilde{a}_l) \otimes \phi_{\mathcal{W}}(\bar{w}_i), \mathbb{E}\left[\phi_{\mathcal{Z}}(Z) \otimes \phi_{\mathcal{W}}(W) \mid A = a\right]\bigg\rangle_{\mathcal{H}_{\mathcal{Z}\mathcal{W}}} \cdot$$

$$\bigg\langle \phi_{\mathcal{A}}(\tilde{a}_l) \otimes \phi_{\mathcal{A}}(\tilde{a}_j), \phi_{\mathcal{A}}(a) \otimes \phi_{\mathcal{A}}(a)\bigg\rangle_{\mathcal{H}_{\mathcal{A}\mathcal{A}}}. \tag{28}$$

In order to evaluate the sum of inner products above, we need the conditional mean embedding $\mu_{Z,W|A}(a) = \mathbb{E}\left[\phi_{\mathcal{Z}}(Z) \otimes \phi_{\mathcal{W}}(W) \mid A = a\right]$. We approximate this term with vector-valued kernel ridge regression. In particular, under the regularity condition $\mathbb{E}[g(Z,W) \mid A = \cdot] \in \mathcal{H}_{\mathcal{A}}$ for all $g \in \mathcal{H}_{\mathcal{Z}\mathcal{W}}$, there exists an operator $C_{Z,W|A} \in \mathcal{S}_2(\mathcal{H}_{\mathcal{A}}, \mathcal{H}_{\mathcal{Z}\mathcal{W}})$ such that $\mathbb{E}\left[\phi_{\mathcal{Z}}(Z) \otimes \phi_{\mathcal{W}}(W) \mid A = a\right] = C_{Z,W|A}\phi_{\mathcal{A}}(a)$. This operator can be learned by minimizing the regularized least-squares function:

$$\hat{\mathcal{L}}_{DR}^c(C) = \frac{1}{t} \sum_{i=1}^{t} \|\phi_{\mathcal{Z}}(z_i) \otimes \phi_{\mathcal{W}}(w_i) - C\phi_{\mathcal{A}}(a_i)\|_{\mathcal{H}_{\mathcal{Z}\mathcal{W}}}^2 + \lambda_{DR} \|C\|_{\mathcal{S}_2(\mathcal{H}_{\mathcal{A}}, \mathcal{H}_{\mathcal{Z}\mathcal{W}})}^2.$$

The minimizer is given by

$$\hat{\mu}_{Z,W|A}(a) = \hat{C}_{Z,W|A}\phi_{\mathcal{A}}(a) = \sum_i \xi_i \phi_{\mathcal{Z}}(z_i) \otimes \phi_{\mathcal{W}}(w_i) = \Phi_{\mathcal{Z}\mathcal{W}} \left(\boldsymbol{K}_{AA} + t\lambda_{DR}\boldsymbol{I}\right)^{-1} \boldsymbol{K}_{Aa}, \tag{29}$$

where $\xi_i = \left[\left(\boldsymbol{K}_{AA} + t\lambda_{DR}\boldsymbol{I}\right)^{-1} \boldsymbol{K}_{Aa}\right]_i$ and $\Phi_{\mathcal{Z}\mathcal{W}} = [\phi_{\mathcal{Z}}(z_1) \otimes \phi_{\mathcal{W}}(w_1) \quad \ldots \quad \phi_{\mathcal{Z}}(z_t) \otimes \phi_{\mathcal{W}}(w_t)]$. By plugging this approximation to conditional

mean embedding in Equation (28), we observe that

$$\sum_{i,j}\sum_{l,t}\alpha_{ij}\gamma_{lt}\Big\langle\hat\mu_{Z|W,A}(\tilde w_t,\tilde a_l)\otimes\phi_{\mathcal W}(\bar w_i),\sum_s\xi_s\phi_{\mathcal Z}(z_s)\otimes\phi_{\mathcal W}(w_s)\Big\rangle_{\mathcal H_{\mathcal{ZW}}}\cdot$$

$$\Big\langle\phi_{\mathcal A}(\tilde a_l)\otimes\phi_{\mathcal A}(\tilde a_j),\phi_{\mathcal A}(a)\otimes\phi_{\mathcal A}(a)\Big\rangle_{\mathcal H_{\mathcal{AA}}}$$

$$=\sum_{i,j}\sum_{l,t}\sum_s\alpha_{ij}\gamma_{lt}\xi_s\Big\langle\hat\mu_{Z|W,A}(\tilde w_t,\tilde a_l)\otimes\phi_{\mathcal W}(\bar w_i),\phi_{\mathcal Z}(z_s)\otimes\phi_{\mathcal W}(w_s)\Big\rangle_{\mathcal H_{\mathcal{ZW}}}\cdot$$

$$\Big\langle\phi_{\mathcal A}(\tilde a_l)\otimes\phi_{\mathcal A}(\tilde a_j),\phi_{\mathcal A}(a)\otimes\phi_{\mathcal A}(a)\Big\rangle_{\mathcal H_{\mathcal{AA}}}$$

$$=\sum_s\xi_s\Big\langle\sum_{l,t}\gamma_{lt}\hat\mu_{Z|W,A}(\tilde w_t,\tilde a_l)\otimes\phi_{\mathcal A}(\tilde a_l),\phi_{\mathcal Z}(z_s)\otimes\phi_{\mathcal A}(a)\Big\rangle_{\mathcal H_{\mathcal{ZA}}}\cdot$$

$$\Big\langle\sum_{i,j}\alpha_{ij}\phi_{\mathcal W}(\bar w_i)\otimes\phi_{\mathcal A}(\tilde a_j),\phi_{\mathcal W}(w_s)\otimes\phi_{\mathcal A}(a)\Big\rangle_{\mathcal H_{\mathcal{WA}}}$$

$$=\sum_s\xi_s\hat\varphi(z_s,a)\hat h(w_s,a).$$

In conclusion, we estimate the term $\mathbb E[\varphi_0(Z,a)h_0(W,a)\mid A=a]$ with the following expression:

$$\mathbb E[\varphi_0(Z,a)h_0(W,a)\mid A=a]\approx\sum_s\xi_s(a)\hat\varphi(z_s,a)\hat h(w_s,a),\tag{30}$$

where $\xi_s(a)=\Big[(\boldsymbol K_{AA}+t\lambda_{DR}\boldsymbol I)^{-1}\boldsymbol K_{Aa}\Big]_s$. The same procedure can be applied with the PMMR representation of the outcome-bridge function $\hat h$ that will lead to the same form of estimation in Equation (30). For completeness of this section, we repeat the pseudo-code of doubly robust dose-response estimation algorithms that we give in Section (3.2) below. Note that DRKPV uses the KPV algorithm (summarized in Algorithm (2)), whereas DRPMMR employs the PMMR method (summarized in Algorithm (3)).

---

**Algorithm 5** Doubly Robust Kernel Proxy Variable Algorithm (Replication of Algorithm (1) in Section (3.2)).

---

**Input**: Training samples $\{y_i,w_i,z_i,a_i\}_{i=1}^t\subset\mathcal D$,
**Parameters:** Regularization parameter $\lambda_{\text{DR}}$ and the parameters of Algorithms (4) and (2) or (3).
**Output**: Doubly robust dose-response estimation $\hat\theta_{\text{ATE}}^{(\text{DR})}(a)$ for all $a\in\mathcal A$.

1: Compute the estimated outcome bridge function $\hat h$ and dose-response curve $\hat\theta_1(\cdot)=\hat E[\hat h(W,\cdot)]$ with either Algorithm (2) or (3).
2: Compute the estimated treatment bridge function $\hat\varphi$ and the dose-response curve $\hat\theta_2(\cdot)=\hat E[Y\hat\varphi(Z,\cdot)\mid A=\cdot]$ with Algorithm (4).
3: Let $\hat\theta_3(\cdot)$ be given by $\hat\theta_3(\cdot)=\sum_i\xi_i(\cdot)\hat\varphi(z_i,\cdot)\hat h(w_i,\cdot)$, where $\xi_i(a)=\big[(\boldsymbol K_{AA}+t\lambda_{\text{DR}}\boldsymbol I)^{-1}\boldsymbol K_{Aa}\big]_i$ for all $a\in\mathcal A$.
4: For each $a\in\mathcal A$, return the doubly robust dose-response estimation as

$$\hat\theta_{\text{ATE}}^{(\text{DR})}(a)=\hat\theta_1(a)+\hat\theta_2(a)-\hat\theta_3(a).$$

---

# D   Discussion on the existence of bridge functions

In this section, we present the conditions under which the outcome and treatment bridge functions exist in their respective RKHSs, following the formulations in Miao et al. [5], Deaner [10], Xu et al. [13], and Bozkurt et al. [23]. To this end, we consider the conditional expectation operators defined by

$$E_a:\mathcal L_2(\mathcal W,p_{W|A=a})\to\mathcal L_2(\mathcal Z,p_{Z|A=a}),$$
$$F_a:\mathcal L_2(\mathcal Z,p_{Z|A=a})\to\mathcal L_2(\mathcal W,p_{W|A=a}),$$

such that

$$E_a\ell(z) = \mathbb{E}[\ell(W)|Z=z, A=a], \qquad z \in \mathcal{Z}$$
$$F_a\ell(w) = \mathbb{E}[\ell(Z)|W=w, A=a], \qquad w \in \mathcal{W}.$$

Here, $p_{W|A=a}$ and $p_{Z|A=a}$ denote the conditional distributions of $W$ and $Z$ given $A=a$, respectively.

To ensure the existence of the bridge functions, we impose the following assumptions on the conditional expectation operators.

**Assumption D.1** (Assumption (4) in [13]). *For each $a \in \mathcal{A}$, the operator $E_a$ is compact operator with singular value decomposition $\{\eta_{E_a,i}, \phi_{E_a,i}, \psi_{E_a,i}\}_{i=1}^\infty$.*

**Assumption D.2** (Assumption (10.3) in [23]). *For each $a \in \mathcal{A}$, the operator $F_a$ is compact operator with singular value decomposition $\{\eta_{F_a,i}, \phi_{F_a,i}, \psi_{F_a,i}\}_{i=1}^\infty$.*

To apply Picard's Theorem [63, Theorem 15.8]—as used in Lemma (2) of Xu et al. [13] and Theorem (10.6) of Bozkurt et al. [23]—to establish the existence of bridge functions, we make the following additional assumptions.

**Assumption D.3** (Assumption (5) in [13]). *For each $a \in \mathcal{A}$, the conditional expectation $\ell_{Y|a} := \mathbb{E}[Y \mid Z = \cdot, A = a]$ satisfies*

$$\sum_{i=1}^\infty \frac{1}{\eta_{E_a,i}^2} \left| \langle \ell_{Y|a}, \psi_{E_a,i} \rangle_{\mathcal{L}_2(\mathcal{Z}, p_{Z|A=a})} \right|^2 < \infty,$$

*where the singular system $\{\eta_{E_a,i}, \phi_{E_a,i}, \psi_{E_a,i}\}_{i=1}^\infty$ is given in Assumption (D.1).*

**Assumption D.4** (Assumption (10.4) in [23]). *For each $a \in \mathcal{A}$, the density ratio $r_a(W) := \frac{p(W)p(a)}{p(W,a)}$ satisfies*

$$\sum_{i=1}^\infty \frac{1}{\eta_{F_a,i}^2} \left| \langle r_a, \psi_{F_a,i} \rangle_{\mathcal{L}_2(\mathcal{W}, p_{W|A=a})} \right|^2 < \infty,$$

*where the singular system $\{\eta_{F_a,i}, \phi_{F_a,i}, \psi_{F_a,i}\}_{i=1}^\infty$. is given in Assumption (D.2).*

We are now ready to state the results on the existence of bridge functions. The following lemma establishes the existence of the outcome bridge function.

**Lemma D.5** (Lemma (2) in [13]). *Suppose that Assumptions (2.2), (2.5), (D.1), and (D.3) hold. Then, for each $a \in \mathcal{A}$, there exists a solution to the functional equation*

$$\mathbb{E}[Y \mid Z, A] = \int h_0(w, A)p(w \mid Z, A)dw.$$

The next lemma establishes the existence of the treatment bridge function.

**Lemma D.6** (Theorem (10.6) in [23]). *Suppose that Assumptions (2.2), (2.3), (D.2), and (D.4) hold. Then, for each $a \in \mathcal{A}$, there exists a solution to the functional equation*

$$\mathbb{E}[\varphi_0(Z, a)|W, A = a] = \frac{p(W)p(a)}{p(W, a)}, \qquad a \in \mathcal{A}.$$

Note that the existence of the **outcome bridge function** hinges on Assumption (2.5), which also ensures the identifiability of the dose-response curve when using the **treatment bridge function** (see Theorem (2.6)). Conversely, the existence of the **treatment bridge function** hinges on Assumption (2.3), which guarantees identifiability of the dose-response curve through the **outcome bridge function** (see Theorem (2.4)). Therefore, these completeness assumptions play distinct yet complementary roles in establishing both existence and identifiability for the respective bridge functions.

## E    Consistency results

In this section, we provide the convergence results of the algorithms that we proposed. First, the following two sections respectively review the consistency results for outcome bridge function- and treatment bridge function-based methods from [11, 12, 23].

### E.1 Consistency results for treatment bridge function-based method

Here, we overview the consistency result of the treatment bridge function based on the results in [23]. For expositional clarity, we first review the results from [23], as we leverage similar developments in presenting the consistency results for the outcome bridge function in the next section. We begin by introducing the assumptions regarding the noise between the treatment and the outcome.

**Assumption E.1** (Assumption (12.1-iv) in [23]). *We assume that there exists $R, \sigma > 0$ such that for all $q \geq 2$, $P_A-$almost surely,*

$$\mathbb{E}[(Y - \mathbb{E}[Y \mid A])^q \mid A] \leq \frac{1}{2}q!\sigma^2 R^{q-2}. \tag{31}$$

Next, we assume that the problem is well-specified, as stated in the following assumption:

**Assumption E.2** (Assumption (5.1) in [23]). *(1) There exists $C_{Z|W,A} \in S_2(\mathcal{H}_{\mathcal{W}\mathcal{A}}, \mathcal{H}_{\mathcal{Z}})$ such that $\mu_{Z|W,A}(W, A) = C_{Z|W,A}\phi_{\mathcal{W}\mathcal{A}}(W, A)$; (2) There exists a solution $\varphi_0 \in \mathcal{H}_{\mathcal{Z}\mathcal{A}}$ of Equation (2); (3) There exists $C_{YZ|A} \in S_2(\mathcal{H}_{\mathcal{A}}, \mathcal{H}_{\mathcal{Z}})$ such that $\mathbb{E}[Y\phi_{\mathcal{Z}}(Z)|A] = C_{YZ|A}\phi_{\mathcal{A}}(A)$.*

Following proof techniques from [64], Bozkurt et al. [23] show that convergence to the minimum RKHS norm solution of the treatment bridge equation—characterized in the following definition—is sufficient for consistency in estimating the dose-response curve using the Kernel Alternative Proxy (KAP) method. We follow the same setting here.

**Definition E.3** (Treatment bridge function solution with minimum RKHS norm, Definition (12.2) in [23]). *Under Assumption (E.1-(2)), the minimum RKHS solution to Equation (2) is defined as*

$$\bar{\varphi}_0 = \arg\min_{\varphi \in \mathcal{H}_{\mathcal{Z}\mathcal{A}}} \|\varphi\|_{\mathcal{H}_{\mathcal{Z}\mathcal{A}}} \quad s.t. \quad \mathbb{E}[\varphi(Z, A)|W, A] = r(W, A),$$

*with $r(W, A) = \frac{p(W)p(A)}{p(W,A)}$.*

Next, we define the (uncentered) covariance operators corresponding to the three-stages of the KAP method:

**Definition E.4** (Definition (12.4) in [23]). *The covariance operators are defined as*

- *(Stage 1)* $\Sigma_{\varphi,1} = \mathbb{E}[\phi_{\mathcal{W}\mathcal{A}}(W, A) \otimes \phi_{\mathcal{W}\mathcal{A}}(W, A)], \qquad \phi_{\mathcal{W}\mathcal{A}}(W, A) = \phi_{\mathcal{W}}(W) \otimes \phi_{\mathcal{A}}(A);$
- *(Stage 2)* $\Sigma_{\varphi,2} = \mathbb{E}\left[\left(\left(\mu_{Z|W,A}(W, A) \otimes \phi_{\mathcal{A}}(A)\right) \otimes \left(\mu_{Z|W,A}(W, A) \otimes \phi_{\mathcal{A}}(A)\right)\right)\right];$
- *(Stage 3)* $\Sigma_{\varphi,3} = \mathbb{E}[\phi_{\mathcal{A}}(A) \otimes \phi_{\mathcal{A}}(A)].$

The operators $\Sigma_{\varphi,1}$, $\Sigma_{\varphi,2}$, and $\Sigma_{\varphi,3}$ are self-adjoint and positive semi-definite. With Assumption (3.1), they belong to the trace class, which ensures their compactness and implies that they have a countable spectrum [65].

Furthermore, we state the source condition (SRC) and eigenvalue decay (EVD) assumptions [66, 67]:

**Assumption E.5** (Assumption (12.6) in [23]). *We assume that the following conditions hold:*

- *There exists a constant $B_{\varphi,1} < \infty$ such that for a given $\beta_{\varphi,1} \in (1, 3]$,*

$$\|C_{Z|W,A}\Sigma_{\varphi,1}^{-\frac{\beta_{\varphi,1}-1}{2}}\|_{S_2(\mathcal{H}_{\mathcal{W}\mathcal{A}}, \mathcal{H}_{\mathcal{Z}})} \leq B_{\varphi,1}$$

- *There exists a constant $B_{\varphi,2} < \infty$ such that for a given $\beta_{\varphi,2} \in (1, 3]$,*

$$\|\Sigma_{\varphi,2}^{-\frac{\beta_{\varphi,2}-1}{2}}\bar{\varphi}_0\|_{\mathcal{H}_{\mathcal{Z}\mathcal{A}}} \leq B_{\varphi,2}.$$

- *There exists a constant $B_{\varphi,3} < \infty$ such that for a given $\beta_{\varphi,3} \in (1, 3]$,*

$$\|C_{YZ|A}\Sigma_{\varphi,3}^{-\frac{\beta_{\varphi,3}-1}{2}}\|_{S_2(\mathcal{H}_{\mathcal{A}}, \mathcal{H}_{\mathcal{Z}})} \leq B_{\varphi,3}.$$

**Assumption E.6** (Assumption (12.9) in [23]). *We assume the following conditions hold*

- *Let $(\lambda_{\varphi,1,i})_{i\geq 1}$ be the eigenvalues of $\Sigma_{\varphi,1}$. For some constant $c_{\varphi,1} > 0$ and parameter $p_{\varphi,1} \in (0, 1]$ and for all $i \geq 1$,*

$$\lambda_{\varphi,1,i} \leq c_{\varphi,1}i^{-1/p_{\varphi,1}}.$$

- *Let $(\lambda_{\varphi,2,i})_{i\geq 1}$ be the eigenvalues of $\Sigma_{\varphi,2}$. For some constant $c_{\varphi,2} > 0$ and parameter $p_{\varphi,2} \in (0,1]$ and for all $i \geq 1$,*

$$\lambda_{\varphi,2,i} \leq c_{\varphi,2} i^{-1/p_{\varphi,2}}.$$

- *Let $(\lambda_{\varphi,3,i})_{i\geq 1}$ be the eigenvalues of $\Sigma_{\varphi,3}$. For some constant $c_{\varphi,3} > 0$ and parameter $p_{\varphi,3} \in (0,1]$ and for all $i \geq 1$,*

$$\lambda_{\varphi,3,i} \leq c_{\varphi,3} i^{-1/p_{\varphi,3}}.$$

Now, we are ready to state the convergence result of the treatment bridge function. We note that Assumptions (E.2-3), (E.5-3) and (E.6-3) are not required for the bridge function consistency. However, the consistency of our proposed double robust estimators rely on these assumptions as we show in Section (S.M. E.3).

**Theorem E.7** (Theorem 12.21 in [23])**.** *Suppose Assumptions (3.1), (E.1), (E.2-1 & 2), (E.5-1 & 2) and (E.6-1 & 2) hold and set $\lambda_{\varphi,1} = \Theta\left(n_\varphi^{-\frac{1}{\beta_{\varphi,1}+p_{\varphi,1}}}\right)$ and $n_\varphi = m_\varphi^{\iota_\varphi \frac{\beta_{\varphi,1}+p_{\varphi,1}}{\beta_{\varphi,1}-1}}$ where $\iota_\varphi > 0$. Then,*

i. *If $\iota_\varphi \leq \frac{\beta_{\varphi,2}+1}{\beta_{\varphi,2}+p_{\varphi,2}}$ then $\|\hat{\varphi} - \bar{\varphi}_0\|_{\mathcal{H}_{\mathcal{ZA}}} = O_p\left(m_\varphi^{-\frac{\iota_\varphi}{2}\frac{\beta_{\varphi,2}-1}{\beta_{\varphi,2}+1}}\right)$ with $\lambda_{\varphi,2} = \Theta\left(m_\varphi^{-\frac{\iota_\varphi}{\beta_{\varphi,2}+1}}\right)$;*

ii. *If $\iota_\varphi \geq \frac{\beta_{\varphi,2}+1}{\beta_{\varphi,2}+p_{\varphi,2}}$ then $\|\hat{\varphi} - \bar{\varphi}_0\|_{\mathcal{H}_{\mathcal{ZA}}} = O_p\left(m_\varphi^{-\frac{1}{2}\frac{\beta_{\varphi,2}-1}{\beta_{\varphi,2}+p_{\varphi,2}}}\right)$ with $\lambda_{\varphi,2} = \Theta\left(m_\varphi^{-\frac{1}{\beta_{\varphi,2}+p_{\varphi,2}}}\right)$.*

$\iota_\varphi$ controls the ratio of stage 1 to stage 2 samples. In Theorem E.7-(ii), when $\iota_\varphi \geq \frac{\beta_{\varphi,2}+1}{\beta_{\varphi,2}+p_{\varphi,2}}$, i.e. when we have enough stage 1 samples relatively to stage 2 samples, the convergence rate is optimal in $m_\varphi$ under the assumptions of the theorem (see Section 5 [23]).

## E.2 Consistency results for outcome bridge function-based methods

In the following subsections, we review the consistency results for the outcome bridge functions of both the KPV and PMMR methods.

### E.2.1 Consistency result for kernel proxy variable

We first review the consistency result of the outcome bridge function with the Kernel Proxy Variable (KPV) algorithm that have been obtained concurrently by [11, 12]. We assume that the problem is well-specified as stated in the following assumption:

**Assumption E.8** (Assumption (2.1) and (8.1) in [12])**.** *(1) There exists $C_{W|Z,A} \in S_2(\mathcal{H}_{\mathcal{ZA}}, \mathcal{H}_{\mathcal{W}})$ such that $\mu_{W|Z,A}(Z, A) = C_{W|Z,A}\phi_{\mathcal{ZA}}(Z, A)$; (2) There exists a solution $h_0 \in \mathcal{H}_{\mathcal{WA}}$ of Equation (1).*

Following a similar convention to [23], we define the minimum RKHS norm solution to the outcome bridge function equation:

**Definition E.9** (Outcome bridge function solution with minimum RKHS norm)**.** *Under Assumption (E.8-(2)), the minimum RKHS solution to Equation (1) is defined as*

$$\bar{h}_0 = \underset{h \in \mathcal{H}_{\mathcal{WA}}}{\arg\min} \|h\|_{\mathcal{H}_{\mathcal{WA}}} \quad s.t. \quad \mathbb{E}[h(W, A)|Z, A] = \int y p(y \mid z, a) dy,$$

Similar to [23], we establish the convergence to the minimum norm RKHS solution for the outcome bridge function. Now, we define the covariance operators corresponding to different stages of regression of KPV.

**Definition E.10** (Covariance Operators)**.** *The covariance operators are defined as*

- *(Stage 1) $\Sigma_{h,1} = \mathbb{E}\left[\phi_{\mathcal{ZA}}(Z, A) \otimes \phi_{\mathcal{ZA}}(Z, A)\right], \qquad \phi_{\mathcal{ZA}}(Z, A) = \phi_{\mathcal{Z}}(Z) \otimes \phi_{\mathcal{A}}(A)$;*
- *(Stage 2) $\Sigma_{h,2} = \mathbb{E}\left[\left(\mu_{W|Z,A}(Z, A) \otimes \phi_{\mathcal{A}}(A)\right) \otimes \left(\mu_{W|Z,A}(Z, A) \otimes \phi_{\mathcal{A}}(A)\right)\right]$;*

Similarly to the previous section $\Sigma_{h,1}$ and $\Sigma_{h,2}$ are self-adjoint, positive semi-definite and trace class under Assumption (3.1). We state the source condition (SRC) and eigenvalue decay (EVD) assumptions [66, 67] :

**Assumption E.11** (SRC for KPV). *We assume that the following conditions hold:*

- *There exists a constant $B_{h,1} < \infty$ such that for a given $\beta_{h,1} \in (1,3]$,*

$$\|C_{W|Z,A}\Sigma_{h,1}^{-\frac{\beta_{h,1}-1}{2}}\|_{S_2(\mathcal{H}_{\mathcal{Z}\mathcal{A}},\mathcal{H}_{\mathcal{W}})} \leq B_{h,1}$$

- *There exists a constant $B_{h,2} < \infty$ such that for a given $\beta_{h,2} \in (1,3]$,*

$$\|\Sigma_{h,2}^{-\frac{\beta_{h,2}-1}{2}}\bar{h}_0\|_{\mathcal{H}_{\mathcal{W}\mathcal{A}}} \leq B_{h,2}.$$

**Assumption E.12** (EVD for KPV). *We assume the following conditions hold*

- *Let $(\lambda_{h,1,i})_{i\geq1}$ be the eigenvalues of $\Sigma_{h,1}$. For some constant $c_{h,1} > 0$ and parameter $p_{h,1} \in (0,1]$ and for all $i \geq 1$,*

$$\lambda_{h,1,i} \leq c_{h,1}i^{-1/p_{h,1}}.$$

- *Let $(\lambda_{h,2,i})_{i\geq1}$ be the eigenvalues of $\Sigma_{h,2}$. For some constant $c_{h,2} > 0$ and parameter $p_{h,2} \in (0,1]$ and for all $i \geq 1$,*

$$\lambda_{h,2,i} \leq c_{h,2}i^{-1/p_{h,2}}.$$

**Theorem E.13** (Theorem (7) [11]). *Suppose Assumptions (3.1), (E.1), (E.8), (E.11), and (E.12) hold and set $\lambda_{h,1} = \Theta\left(n_h^{-\frac{1}{\beta_{h,1}+p_{h,1}}}\right)$ and $n_h = m_h^{\iota_h\frac{\beta_{h,1}+p_{h,1}}{\beta_{h,1}-1}}$ where $\iota_h > 0$. Then,*

i. *If $\iota_h \leq \frac{\beta_{h,2}+1}{\beta_{h,2}+p_{h,2}}$, then $\|\hat{h} - \bar{h}_0\|_{\mathcal{H}_{\mathcal{W}\mathcal{A}}} = O_p\left(m_h^{-\frac{\iota_h}{2}\frac{\beta_{h,2}-1}{\beta_{h,2}+1}}\right)$ with $\lambda_{h,2} = \Theta\left(m_h^{-\frac{\iota_h}{\beta_{h,2}+1}}\right)$,*

ii. *If $\iota_h \geq \frac{\beta_{h,2}+1}{\beta_{h,2}+p_{h,2}}$, then $\|\hat{h} - \bar{h}_0\|_{\mathcal{H}_{\mathcal{W}\mathcal{A}}} = O_p\left(m_h^{-\frac{1}{2}\frac{\beta_{h,2}-1}{\beta_{h,2}+p_{h,2}}}\right)$ with $\lambda_{h,2} = \Theta\left(m_h^{-\frac{1}{\beta_{h,2}+p_{h,2}}}\right)$.*

Similarly to Theorem (E.7), $\iota_h$ controls the ratio of stage 1 to stage 2 samples. In Theorem (E.13-(ii)), when we have enough stage 1 samples relatively to stage 2 samples, the convergence rate is optimal in $m_h$ under the assumptions of the theorem. A similar result is obtained by Singh [12, Theorem (3)] with a slightly worse requirement on stage 1 samples in order to achieve the optimal rate in $m_h$.

Technically, Mastouri et al. [11] require $Y$ to be almost surely bounded. However, we can resort to the weaker assumption of sub-exponential noise, Assumption (E.1), by using the concentration inequality of [67, Theorem (26)] as done in [64, 23].

### E.2.2 Consistency result for proximal maximum moment restriction

Here, we state the convergence result of the outcome bridge function for the PMMR algorithm, originally established by Mastouri et al. [11]. This result plays a key role in our consistency analysis of the DRPMMR algorithm. Similarly to KPV, we assume that the problem is well-specified but we only require Assumption (E.8-(2)).

**Definition E.14** (Outcome bridge function solution with minimum RKHS norm - PMMR). *Under Assumption (E.8-(2)), we define*

$$\check{h}_0 = \underset{h \in \mathcal{H}_{\mathcal{W}\mathcal{A}}}{\arg\min}\|h\|_{\mathcal{H}_{\mathcal{W}\mathcal{A}}} \quad s.t. \quad \mathbb{E}[h(W,A)\phi(Z,A)] = \mathbb{E}[Y\phi(Z,A)].$$

Note that under Assumption (E.8-(2)) there is always a solution to $\mathbb{E}[h(W,A)\phi(Z,A)] = \mathbb{E}[Y\phi(Z,A)]$. Next, we introduce the cross-covariance operator associated to PMMR.

**Definition E.15** (Cross-covariance Operator). $T = \mathbb{E}[\phi_{\mathcal{Z}\mathcal{A}}(Z,A) \otimes \phi_{\mathcal{W}\mathcal{A}}(W,A)].$

To ensure the consistency of the PMMR outcome bridge function estimator, we require the following assumption.

**Assumption E.16** (Assumption 6 in [11]). *We assume that there exist $C_Y$ such that $|Y| < C_Y$ almost surely and $\mathbb{E}[Y] < C_Y$.*

**Assumption E.17** (SRC for PMMR - Assumption 16 in [11]). *There exists a constant $B'_{h,2} < \infty$ such that for a given $\gamma \in (1,3]$,*

$$\|(T^*T)^{-\frac{\gamma-1}{2}} \check{h}_0\|_{\mathcal{H}_{\mathcal{W}\mathcal{A}}} \le B'_{h,2}.$$

**Theorem E.18** (Theorem 3 in [11]). *Suppose Assumptions (3.1), (E.16), (E.8-(2), and (E.17) hold and set $\lambda_{PMMR} = \Theta\left(t^{-\frac{1}{\gamma+1}}\right)$, then*

$$\|\hat{h} - \check{h}_0\|_{\mathcal{H}_{\mathcal{A}\mathcal{W}}} = O_p\left(t^{-\frac{1}{2}\frac{\gamma-1}{\gamma+1}}\right)$$

Using a refined analysis as done in [64, 23] it should be possible to improve their result with a rate depending on the eigenvalue decay of the operator $T$ as well as relaxing Assumption (E.16) to Assumption (E.1).

**Remark E.19.** *Notice that the convergence rate given in Mastouri et al. [11, Theorem 3] is $\|\hat{h} - \check{h}_0\|_{\mathcal{H}_{\mathcal{A}\mathcal{W}}} = O_p\left(t^{-\frac{1}{2}\min(\frac{1}{2},\frac{\gamma-1}{\gamma+1})}\right)$. However, since $\gamma \in (1,3]$, we have $\frac{\gamma-1}{\gamma+1} \le \frac{1}{2}$. Therefore, the $\min(\cdot)$ operation in this consistency result is not necessary.*

### E.3 Consistency results for doubly robust kernel methods

In this section, we provide the consistency results of our proposed methods DRKPV and DRPMMR. Our results will rely on the consistency results of the outcome bridge function- and treatment bridge function-based methods that we discussed in Section (S.M. E.2) and (S.M. E.1). Furthermore, we have seen that our doubly robust estimation procedures require learning the conditional mean embedding $\mu_{Z,W|A}(a) = C_{Z,W|A}\phi_{\mathcal{A}}(a) = \mathbb{E}[\phi_{\mathcal{Z}}(Z) \otimes \phi_{\mathcal{W}}(W) \mid A = a]$. Recall that we approximate this CME via kernel ridge regression. We assume the following conditions:

**Assumption E.20.** *There exists $C_{Z,W|A} \in S_2(\mathcal{H}_{\mathcal{A}}, \mathcal{H}_{\mathcal{Z}\mathcal{W}})$ such that $\mu_{Z,W|A}(a) = C_{Z,W|A}\phi_{\mathcal{A}}(a)$.*

**Assumption E.21.** *There exists a constant $B_{DR} < \infty$ such that for a given $\beta_{DR} \in (1,3]$,*

$$\|C_{Z,W|A}\Sigma_{\varphi,3}^{-\frac{\beta_{DR}-1}{2}}\|_{S_2(\mathcal{H}_{\mathcal{A}},\mathcal{H}_{\mathcal{Z}\mathcal{W}})} \le B_{DR}.$$

First, recall that we showed in Theorem (2.7) that the dose-response can be identified with

$$\theta_{ATE}(a) = \mathbb{E}[\varphi_0(Z,a)(Y - h_0(W,a)) \mid A = a] + \mathbb{E}[h_0(W,a)].$$

We define an intermediate quantity $\bar{\theta}_{ATE}(a)$ that is given by

$$\bar{\theta}_{ATE}(a) = \mathbb{E}\left[\hat{\varphi}(Z,a)\left(Y - \hat{h}(W,a)\right)\Big|A = a\right] + \mathbb{E}[\hat{h}(W,a)].$$

Noticing, by triangle inequality, that

$$|\theta_{ATE}(a) - \hat{\theta}_{ATE}(a)| \le |\theta_{ATE}(a) - \bar{\theta}_{ATE}(a)| + |\bar{\theta}_{ATE}(a) - \hat{\theta}_{ATE}(a)|,$$

we will subsequently derive bounds for the two terms above in order to achieve the consistency result of our proposed methods.

**Lemma E.22.** *Let $h_0$ and $\varphi_0$ be outcome and treatment bridge functions that satisfy Equations (1) and (2), respectively. Then,*

$$\theta_{ATE}(a) - \bar{\theta}_{ATE}(a) = \mathbb{E}\left[(\varphi_0(Z,a) - \hat{\varphi}(Z,a))\left(h_0(W,a) - \hat{h}(W,a)\right)|A = a\right].$$

*Proof.*

$$\theta_{ATE}(a) - \bar{\theta}_{ATE}(a) = \mathbb{E}[\varphi_0(Z,a)(Y - h_0(W,a)) \mid A = a] + \mathbb{E}[h_0(W,a)]$$

$$-\mathbb{E}\left[\hat{\varphi}(Z,a)\left(Y-\hat{h}(W,a)\right)\Big|A=a\right]-\mathbb{E}[\hat{h}(W,a)]$$

$$=\mathbb{E}\left[\varphi_0(Z,a)Y\mid A=a\right]-\mathbb{E}\left[\hat{\varphi}(Z,a)Y\mid A=a\right]$$

$$-\mathbb{E}\left[\varphi_0(Z,a)h_0(W,a)\mid A=a\right]+\mathbb{E}\left[\hat{\varphi}(Z,a)\hat{h}(W,a)\mid A=a\right]+\mathbb{E}[h_0(W,a)-\hat{h}(W,a)]$$

$$-\mathbb{E}\left[\hat{\varphi}(Z,a)h_0(W,a)\mid A=a\right]+\mathbb{E}\left[\hat{\varphi}(Z,a)h_0(W,a)\mid A=a\right]$$

$$=\mathbb{E}\left[(\varphi_0(Z,a)-\hat{\varphi}(Z,a))Y\mid A=a\right]-\mathbb{E}\left[(\varphi_0(Z,a)-\hat{\varphi}(Z,a))h_0(W,a)\mid A=a\right]$$

$$-\mathbb{E}\left[\hat{\varphi}(Z,a)(h_0(W,a)-\hat{h}(W,a))\mid A=a\right]+\mathbb{E}[h_0(W,a)-\hat{h}(W,a)]$$

$$+\mathbb{E}\left[\varphi_0(Z,a)(h_0(W,a)-\hat{h}(W,a))\mid A=a\right]-\mathbb{E}\left[\varphi_0(Z,a)(h_0(W,a)-\hat{h}(W,a))\mid A=a\right]$$

$$=\mathbb{E}\left[(\varphi_0(Z,a)-\hat{\varphi}(Z,a))(Y-h_0(W,a))\mid A=a\right]$$

$$+\mathbb{E}\left[(\varphi_0(Z,a)-\hat{\varphi}(Z,a))(h_0(W,a)-\hat{h}(W,a))\mid A=a\right]+\mathbb{E}[h_0(W,a)-\hat{h}(W,a)]$$

$$-\mathbb{E}\left[\varphi_0(Z,a)(h_0(W,a)-\hat{h}(W,a))\mid A=a\right]$$

$$=\mathbb{E}\left[\mathbb{E}\left[Y-h_0(W,a)\big|Z,A=a\right](\varphi_0(Z,a)-\hat{\varphi}(Z,a))\big|A=a\right]$$

$$+\mathbb{E}\left[(\varphi_0(Z,a)-\hat{\varphi}(Z,a))(h_0(W,a)-\hat{h}(W,a))\mid A=a\right]+\mathbb{E}[h_0(W,a)-\hat{h}(W,a)]$$

$$-\mathbb{E}\left[\varphi_0(Z,a)(h_0(W,a)-\hat{h}(W,a))\mid A=a\right]$$

$$=\mathbb{E}\left[(\varphi_0(Z,a)-\hat{\varphi}(Z,a))\left(h_0(W,a)-\hat{h}(W,a)\right)\big|A=a\right]$$

$$+\mathbb{E}\left[\varphi_0(Z,a)\left(h_0(W,a)-\hat{h}(W,a)\right)\big|A=a\right]+\mathbb{E}[h_0(W,a)-\hat{h}(W,a)]$$

$$=\mathbb{E}\left[(\varphi_0(Z,a)-\hat{\varphi}(Z,a))\left(h_0(W,a)-\hat{h}(W,a)\right)\big|A=a\right]$$

$$+\int\underbrace{\left(\int\varphi_0(z,a)p(z|w,a)dz\right)}_{p(w)/p(w|a)}\left(h_0(w,a)-\hat{h}(w,a)\right)p(w|a)dw$$

$$+\mathbb{E}[h_0(W,a)-\hat{h}(W,a)]$$

$$=\mathbb{E}\left[(\varphi_0(Z,a)-\hat{\varphi}(Z,a))\left(h_0(W,a)-\hat{h}(W,a)\right)\big|A=a\right]$$

$$+\int\left(h_0(w,a)-\hat{h}(w,a)\right)p(w)dw+\mathbb{E}[h_0(W,a)-\hat{h}(W,a)]$$

$$=\mathbb{E}\left[(\varphi_0(Z,a)-\hat{\varphi}(Z,a))\left(h_0(W,a)-\hat{h}(W,a)\right)\big|A=a\right]$$

$\square$

**Lemma E.23.** *Suppose Assumption (3.1) hold and let $h_0$, $\varphi_0$ be as in Lemma (E.22), with $h_0\in\mathcal{H}_{\mathcal{WA}}$ and $\varphi_0\in\mathcal{H}_{\mathcal{ZA}}$. Then,*

$$|\theta_{ATE}(a)-\bar{\theta}_{ATE}(a)|\le\kappa^4\|\varphi_0-\hat{\varphi}\|\|h_0-\hat{h}\|.$$

*Proof.* By Lemma (E.22),

$$|\theta_{ATE}(a)-\bar{\theta}_{ATE}(a)|=\left|\mathbb{E}\left[(\varphi_0(Z,a)-\hat{\varphi}(Z,a))\left(h_0(W,a)-\hat{h}(W,a)\right)\big|A=a\right]\right|$$

$$\le\mathbb{E}\left[|\varphi_0(Z,a)-\hat{\varphi}(Z,a)||h_0(W,a)-\hat{h}(W,a)|\big|A=a\right]$$

$$=\mathbb{E}\left[|\langle\varphi_0-\hat{\varphi},\phi_{\mathcal{Z}}(Z)\otimes\phi_{\mathcal{A}}(a)\rangle|\left|\langle h_0-\hat{h},\phi_{\mathcal{W}}(W)\otimes\phi_{\mathcal{A}}(a)\rangle\right|\big|A=a\right]$$

$$\le\mathbb{E}\left[\|\varphi_0-\hat{\varphi}\|\|\phi_{\mathcal{Z}}(Z)\otimes\phi_{\mathcal{A}}(a)\|\|h_0-\hat{h}\|\|\phi_{\mathcal{W}}(W)\otimes\phi_{\mathcal{A}}(a)\|\big|A=a\right]$$

$$\le\kappa^4\|\varphi_0-\hat{\varphi}\|\|h_0-\hat{h}\|.$$

$\square$

Next, we need to derive a bound for $|\bar{\theta}_{ATE}(a) - \hat{\theta}_{ATE}(a)|$.

**Lemma E.24.** *Suppose Assumptions (3.1), and (E.20) hold and let $h_0, \varphi_0$ be as in Lemma E.23. Then,*

$$|\bar{\theta}_{ATE}(a) - \hat{\theta}_{ATE}(a)|$$

$$\leq \kappa^2 \left( \|\hat{\varphi} - \varphi_0\| + \|\varphi_0\| \right) \|C_{YZ|A} - \hat{C}_{YZ|A}\| + \kappa \left( \|\hat{h} - h_0\| + \|h_0\| \right) \|\hat{\mu}_{\mathcal{W}} - \mu_{\mathcal{W}}\|$$

$$+ \kappa^3 \left( \|\varphi_0\|\|h_0\| + \|\hat{\varphi} - \varphi_0\|\|h_0\| + \|\hat{h} - h_0\|\|\varphi_0\| + \|\hat{\varphi} - \varphi_0\|\|\hat{h} - h_0\| \right) \|C_{ZW|A} - \hat{C}_{ZW|A}\|.$$

*Proof.* We start by noting that we can write

$$\bar{\theta}_{ATE}(a) = \mathbb{E}\left[\hat{\varphi}(Z, a)(Y - \hat{h}(W, a)) \mid A = a\right] + \mathbb{E}[\hat{h}(W, a)]$$

$$= \mathbb{E}[Y\hat{\varphi}(Z, a) \mid A = a] + \mathbb{E}[\hat{h}(W, a)] - \mathbb{E}[\hat{\varphi}(Z, a)\hat{h}(W, a) \mid A = a]$$

$$= \mathbb{E}[\langle \hat{\varphi}, Y\phi_{\mathcal{Z}}(Z) \otimes \phi_{\mathcal{A}}(a)\rangle \mid A = a] + \mathbb{E}[\langle \hat{h}, \phi_{\mathcal{W}}(W) \otimes \phi_{\mathcal{A}}(a)\rangle]$$

$$- \mathbb{E}[\langle \hat{\varphi} \otimes \hat{h}, (\phi_{\mathcal{Z}}(Z) \otimes \phi_{\mathcal{A}}(a)) \otimes (\phi_{\mathcal{W}}(W) \otimes \phi_{\mathcal{A}}(a))\rangle \mid A = a]$$

$$= \langle \hat{\varphi}, C_{YZ|A}\phi_{\mathcal{A}}(a) \otimes \phi_{\mathcal{A}}(a)\rangle + \langle \hat{h}, \mu_{\mathcal{W}} \otimes \phi_{\mathcal{A}}(a)\rangle - \left\langle \hat{\varphi} \otimes \hat{h}, C_{ZW|A}^{(a)}\phi_{\mathcal{A}}(a)\right\rangle,$$

where we define

$$C_{ZW|A}^{(a)}\phi_{\mathcal{A}}(a) = \mathbb{E}\left[(\phi_{\mathcal{Z}}(Z) \otimes \phi_{\mathcal{A}}(a)) \otimes (\phi_{\mathcal{W}}(W) \otimes \phi_{\mathcal{A}}(a))\rangle \mid A = a\right]. \tag{32}$$

Similarly, we observe that we can write our doubly robust estimate as

$$\hat{\theta}_{ATE}(a) = \langle \hat{\varphi}, \hat{C}_{YZ|A}\phi_{\mathcal{A}}(a) \otimes \phi_{\mathcal{A}}(a)\rangle + \langle \hat{h}, \hat{\mu}_{\mathcal{W}} \otimes \phi_{\mathcal{A}}(a)\rangle - \left\langle \hat{\varphi} \otimes \hat{h}, \hat{C}_{ZW|A}^{(a)}\phi_{\mathcal{A}}(a)\right\rangle,$$

and define $\hat{C}_{ZW|A}^{(a)}$ to be the augmented version of the conditional mean operator estimate in Equation (29), that is given by

$$\hat{C}_{Z,W|A}^{(a)}\phi_{\mathcal{A}}(a) = \sum_i \xi_i \phi_{\mathcal{Z}}(z_i) \otimes \phi_{\mathcal{A}}(a) \otimes \phi_{\mathcal{W}}(w_i) \otimes \phi_{\mathcal{A}}(a)$$

where $\xi_i(a) = \left[ (\boldsymbol{K}_{AA} + t\lambda_{DR}\boldsymbol{I})^{-1} \boldsymbol{K}_{Aa} \right]_i$. Therefore, we have that

$$\bar{\theta}_{ATE}(a) - \hat{\theta}_{ATE}(a) = \langle \hat{\varphi}, C_{YZ|A}\phi_{\mathcal{A}}(a) \otimes \phi_{\mathcal{A}}(a)\rangle + \langle \hat{h}, \mu_{\mathcal{W}} \otimes \phi_{\mathcal{A}}(a)\rangle - \left\langle \hat{\varphi} \otimes \hat{h}, C_{ZW|A}^{(a)}\phi_{\mathcal{A}}(a)\right\rangle$$

$$- \langle \hat{\varphi}, \hat{C}_{YZ|A}\phi_{\mathcal{A}}(a) \otimes \phi_{\mathcal{A}}(a)\rangle - \langle \hat{h}, \hat{\mu}_{\mathcal{W}} \otimes \phi_{\mathcal{A}}(a)\rangle + \left\langle \hat{\varphi} \otimes \hat{h}, \hat{C}_{ZW|A}^{(a)}\phi_{\mathcal{A}}(a)\right\rangle$$

$$= \left\langle \hat{\varphi}, \left(C_{YZ|A} - \hat{C}_{YZ|A}\right) \phi_{\mathcal{A}}(a) \otimes \phi_{\mathcal{A}}(a)\right\rangle$$

$$+ \left\langle \hat{h}, (\mu_{\mathcal{W}} - \hat{\mu}_{\mathcal{W}}) \otimes \phi_{\mathcal{A}}(a)\right\rangle - \left\langle \hat{\varphi} \otimes \hat{h}, \left(C_{ZW|A}^{(a)} - \hat{C}_{ZW|A}^{(a)}\right) \phi_{\mathcal{A}}(a)\right\rangle$$

$$= \left\langle \hat{\varphi} - \varphi_0, \left(C_{YZ|A} - \hat{C}_{YZ|A}\right) \phi_{\mathcal{A}}(a) \otimes \phi_{\mathcal{A}}(a)\right\rangle$$

$$+ \left\langle \varphi_0, \left(C_{YZ|A} - \hat{C}_{YZ|A}\right) \phi_{\mathcal{A}}(a) \otimes \phi_{\mathcal{A}}(a)\right\rangle$$

$$+ \left\langle \hat{h} - h_0, (\mu_{\mathcal{W}} - \hat{\mu}_{\mathcal{W}}) \otimes \phi_{\mathcal{A}}(a)\right\rangle + \langle h_0, (\mu_{\mathcal{W}} - \hat{\mu}_{\mathcal{W}}) \otimes \phi_{\mathcal{A}}(a)\rangle$$

$$- \left\langle \hat{\varphi} \otimes \hat{h}, \left(C_{ZW|A}^{(a)} - \hat{C}_{ZW|A}^{(a)}\right) \phi_{\mathcal{A}}(a)\right\rangle$$

As a result,

$$|\bar{\theta}_{ATE}(a) - \hat{\theta}_{ATE}(a)|$$

$$\leq \kappa^2 \left(\|\hat{\varphi} - \varphi_0\| + \|\varphi_0\|\right) \|C_{YZ|A} - \hat{C}_{YZ|A}\| + \kappa \left(\|\hat{h} - h_0\| + \|h_0\|\right) \|\hat{\mu}_{\mathcal{W}} - \mu_{\mathcal{W}}\|$$

$$+ \left(\|\varphi_0\|\|h_0\| + \|\hat{\varphi} - \varphi_0\|\|h_0\| + \|\hat{h} - h_0\|\|\varphi_0\| + \|\hat{\varphi} - \varphi_0\|\|\hat{h} - h_0\|\right) \left\|\left(C_{ZW|A}^{(a)} - \hat{C}_{ZW|A}^{(a)}\right)\phi_{\mathcal{A}}(a)\right\|$$

Note that $\mathcal{H}_{\mathcal{ZAWA}}$ and $\mathcal{H}_{\mathcal{ZWAA}}$ are isometric Hilbert spaces. Let $\Psi$ be the canonical isomorphism,

$$\Psi : \mathcal{H}_{\mathcal{Z}} \otimes \mathcal{H}_{\mathcal{W}} \otimes \mathcal{H}_{\mathcal{A}} \otimes \mathcal{H}_{\mathcal{A}} \to \mathcal{H}_{\mathcal{Z}} \otimes \mathcal{H}_{\mathcal{A}} \otimes \mathcal{H}_{\mathcal{W}} \otimes \mathcal{H}_{\mathcal{A}}$$

such that it acts on the elementary tensors as $\Psi(u \otimes v \otimes a_1 \otimes a_2) = u \otimes a_1 \otimes v \otimes a_2$. $\Psi$ is a unitary isomorphism, hence,

$$\left\|C_{ZW|A}^{(a)}\phi_{\mathcal{A}}(a) - \hat{C}_{ZW|A}^{(a)}\phi_{\mathcal{A}}(a)\right\|_{\mathcal{H}_{\mathcal{ZAWA}}}$$

$$= \left\|\Psi\left(\left(C_{ZW|A}\phi_{\mathcal{A}}(a)\right) \otimes \phi_{\mathcal{A}}(a) \otimes \phi_{\mathcal{A}}(a)\right) - P\left(\left(\hat{C}_{ZW|A}\phi_{\mathcal{A}}(a)\right) \otimes \phi_{\mathcal{A}}(a) \otimes \phi_{\mathcal{A}}(a)\right)\right\|_{\mathcal{H}_{\mathcal{ZAWA}}}$$

$$= \left\|\left(C_{ZW|A}\phi_{\mathcal{A}}(a)\right) \otimes \phi_{\mathcal{A}}(a) \otimes \phi_{\mathcal{A}}(a) - \left(\hat{C}_{ZW|A}\phi_{\mathcal{A}}(a)\right) \otimes \phi_{\mathcal{A}}(a) \otimes \phi_{\mathcal{A}}(a)\right\|_{\mathcal{H}_{\mathcal{ZWAA}}}$$

$$\leq \kappa^2 \left\|C_{ZW|A}\phi_{\mathcal{A}}(a) - \hat{C}_{ZW|A}\phi_{\mathcal{A}}(a)\right\|_{\mathcal{H}_{\mathcal{ZW}}} \leq \kappa^3 \left\|C_{ZW|A} - \hat{C}_{ZW|A}\right\|_{S_2(\mathcal{H}_{\mathcal{A}}, \mathcal{H}_{\mathcal{ZW}})}$$

Using this inequality, we obtain the desired bound as

$$|\bar{\theta}_{ATE}(a) - \hat{\theta}_{ATE}(a)|$$

$$\leq \kappa^2 \left(\|\hat{\varphi} - \varphi_0\| + \|\varphi_0\|\right) \|C_{YZ|A} - \hat{C}_{YZ|A}\| + \kappa \left(\|\hat{h} - h_0\| + \|h_0\|\right) \|\hat{\mu}_{\mathcal{W}} - \mu_{\mathcal{W}}\|$$

$$+ \kappa^3 \left(\|\varphi_0\|\|h_0\| + \|\hat{\varphi} - \varphi_0\|\|h_0\| + \|\hat{h} - h_0\|\|\varphi_0\| + \|\hat{\varphi} - \varphi_0\|\|\hat{h} - h_0\|\right) \|C_{ZW|A} - \hat{C}_{ZW|A}\|$$

$$\square$$

Combining the bounds from Lemma (E.23) and (E.24), we observe that

$$|\theta_{\text{ATE}}(a) - \hat{\theta}_{\text{ATE}}(a)| \leq \kappa^4 \|\varphi_0 - \hat{\varphi}\|\|h_0 - \hat{h}\| \tag{33}$$

$$+ \kappa^2 \left(\|\hat{\varphi} - \varphi_0\| + \|\varphi_0\|\right) \|C_{YZ|A} - \hat{C}_{YZ|A}\| + \kappa \left(\|\hat{h} - h_0\| + \|h_0\|\right) \|\hat{\mu}_{\mathcal{W}} - \mu_{\mathcal{W}}\|$$

$$+ \kappa^3 \left(\|\varphi_0\|\|h_0\| + \|\hat{\varphi} - \varphi_0\|\|h_0\| + \|\hat{h} - h_0\|\|\varphi_0\| + \|\hat{\varphi} - \varphi_0\|\|\hat{h} - h_0\|\right) \|C_{ZW|A} - \hat{C}_{ZW|A}\|$$

Hence, the convergence of our methods will depend on the factors $\|\hat{h} - h_0\|$, $\|\hat{\varphi} - \varphi_0\|$, $\|\hat{\mu}_{\mathcal{W}} - \mu_{\mathcal{W}}\|$, $\|C_{YZ|A} - \hat{C}_{YZ|A}\|$, and $\|C_{ZW|A} - \hat{C}_{ZW|A}\|$. We have already established the consistency results for $\|\hat{h} - h_0\|$ and $\|\hat{\varphi} - \varphi_0\|$, and $\|\hat{\mu}_{\mathcal{W}} - \mu_{\mathcal{W}}\|$ converges with $O_p(t^{-1/2})$ [11]. Hence, we need to establish the converge results for $\|C_{YZ|A} - \hat{C}_{YZ|A}\|$, and $\|C_{ZW|A} - \hat{C}_{ZW|A}\|$.

**Theorem E.25** (Theorem (12.22) [23], Theorem (3) [45])**.** *Suppose Assumptions (E.1), (E.2-3), (3.1) (E.5-3) and (E.6-3) hold and take $\lambda_{\varphi,3} = \Theta\left(t_{\varphi}^{-\frac{1}{\beta_{\varphi,3} + p_{\varphi,3}}}\right)$. There is a constant $J_3 > 0$ independent of $t_{\varphi} \geq 1$ and $\delta \in (0, 1)$ such that*

$$\left\|\hat{C}_{YZ|A} - C_{YZ|A}\right\|_{S_2(\mathcal{H}_{\mathcal{A}}, \mathcal{H}_{\mathcal{Z}})} \leq J_3 \log(5/\delta) \left(\frac{1}{\sqrt{t_{\varphi}}}\right)^{\frac{\beta_{\varphi,3} - 1}{\beta_{\varphi,3} + p_{\varphi,3}}} =: r_{\varphi,3}(\delta, t_{\varphi}, \beta_{\varphi,3}, p_{\varphi,3}),$$

*is satisfied for sufficiently large $t_{\varphi} \geq 1$ with probability at least $1 - \delta$.*

**Theorem E.26** (Theorem (3) [45])**.** *Suppose Assumptions (E.20), (3.1), (E.21) and (E.6-3), hold and take $\lambda_{DR} = \Theta\left(t^{-\frac{1}{\beta_{DR} + p_{\varphi,3}}}\right)$. There is a constant $J_{DR} > 0$ independent of $t \geq 1$ and $\delta \in (0, 1)$ such that*

$$\left\|\hat{C}_{ZW|A} - C_{ZW|A}\right\|_{S_2(\mathcal{H}_{\mathcal{A}}, \mathcal{H}_{\mathcal{ZW}})} \leq J_{DR} \log(5/\delta) \left(\frac{1}{\sqrt{t}}\right)^{\frac{\beta_{DR} - 1}{\beta_{DR} + p_{\varphi,3}}} =: r_{DR}(\delta, t, \beta_{DR}, p_{\varphi,3}),$$

*is satisfied for sufficiently large $t \geq 1$ with probability at least $1 - \delta$.*

Recall that KPV and KAP use the number of samples $\{n_h, m_h, t_h\}$ and $\{n_\varphi, m_\varphi, t_\varphi\}$ in their first, second, and third stage of regression, respectively. Following the original implementations from Mastouri et al. [11] and Bozkurt et al. [23], we will reuse the data in their third-stage, i.e., $t_h = t_\varphi = t$, as pointed out in Algorithms (2) and (4).

**Theorem E.27.** *Suppose the assumptions in Theorems (E.7), (E.13), (E.25), and (E.26) hold. For a given training dataset $\mathcal{D} = \{y_i, w_i, z_i, a_i\}_{i=1}^{t}$, let $\{n_h, m_h, t_h\}$ and $\{n_\varphi, m_\varphi, t_\varphi\}$ denote the number of samples used in first-, second-, and third-stage regressions for KPV and KAP, respectively, with $t_h = t_\varphi = t$. Then, for DRKPV algorithm with high probability,*

i. *If* $\iota_h \leq \frac{\beta_{h,2}+1}{\beta_{h,2}+p_{h,2}}$ *and* $\iota_\varphi \leq \frac{\beta_{\varphi,2}+1}{\beta_{\varphi,2}+p_{\varphi,2}}$. *Set* $\lambda_{h,2} = \Theta\left(m_h^{-\frac{\iota_h}{\beta_{h,2}+1}}\right)$ *and* $\lambda_{\varphi,2} = \Theta\left(m_\varphi^{-\frac{\iota_\varphi}{\beta_{\varphi,2}+1}}\right)$. *Then,*

$$|\theta_{ATE}(a) - \hat{\theta}_{ATE}(a)| = O_p\left(t^{-\frac{1}{2}\frac{\beta_{\varphi,3}-1}{\beta_{\varphi,3}+p_{\varphi,3}}} + m_h^{-\frac{\iota_h}{2}\frac{\beta_{h,2}-1}{\beta_{h,2}+1}} m_\varphi^{-\frac{\iota_\varphi}{2}\frac{\beta_{\varphi,2}-1}{\beta_{\varphi,2}+1}}\right)$$

ii. *If* $\iota_h \leq \frac{\beta_{h,2}+1}{\beta_{h,2}+p_{h,2}}$ *and* $\iota_\varphi \geq \frac{\beta_{\varphi,2}+1}{\beta_{\varphi,2}+p_{\varphi,2}}$. *Set* $\lambda_{h,2} = \Theta\left(m_h^{-\frac{\iota_h}{\beta_{h,2}+1}}\right)$ *and* $\lambda_{\varphi,2} = \Theta\left(m_\varphi^{-\frac{1}{\beta_{\varphi,2}+p_{\varphi,2}}}\right)$. *Then,*

$$|\theta_{ATE}(a) - \hat{\theta}_{ATE}(a)| = O_p\left(t^{-\frac{1}{2}\frac{\beta_{\varphi,3}-1}{\beta_{\varphi,3}+p_{\varphi,3}}} + m_h^{-\frac{\iota_h}{2}\frac{\beta_{h,2}-1}{\beta_{h,2}+1}} m_\varphi^{-\frac{1}{2}\frac{\beta_{\varphi,2}-1}{\beta_{\varphi,2}+p_{\varphi,2}}}\right)$$

iii. *If* $\iota_h \geq \frac{\beta_{h,2}+1}{\beta_{h,2}+p_{h,2}}$ *and* $\iota_\varphi \leq \frac{\beta_{\varphi,2}+1}{\beta_{\varphi,2}+p_{\varphi,2}}$. *Set* $\lambda_{h,2} = \Theta\left(m_h^{-\frac{1}{\beta_{h,2}+p_{h,2}}}\right)$ *and* $\lambda_{\varphi,2} = \Theta\left(m_\varphi^{-\frac{\iota_\varphi}{\beta_{\varphi,2}+1}}\right)$. *Then,*

$$|\theta_{ATE}(a) - \hat{\theta}_{ATE}(a)| = O_p\left(t^{-\frac{1}{2}\frac{\beta_{\varphi,3}-1}{\beta_{\varphi,3}+p_{\varphi,3}}} + m_h^{-\frac{1}{2}\frac{\beta_{h,2}-1}{\beta_{h,2}+p_{h,2}}} m_\varphi^{-\frac{\iota_\varphi}{2}\frac{\beta_{\varphi,2}-1}{\beta_{\varphi,2}+1}}\right)$$

iv. *If* $\iota_h \geq \frac{\beta_{h,2}+1}{\beta_{h,2}+p_{h,2}}$ *and* $\iota_\varphi \geq \frac{\beta_{\varphi,2}+1}{\beta_{\varphi,2}+p_{\varphi,2}}$. *Set* $\lambda_{h,2} = \Theta\left(m_h^{-\frac{1}{\beta_{h,2}+p_{h,2}}}\right)$ *and* $\lambda_{\varphi,2} = \Theta\left(m_\varphi^{-\frac{1}{\beta_{\varphi,2}+p_{\varphi,2}}}\right)$. *Then,*

$$|\theta_{ATE}(a) - \hat{\theta}_{ATE}(a)| = O_p\left(t^{-\frac{1}{2}\frac{\beta_{\varphi,3}-1}{\beta_{\varphi,3}+p_{\varphi,3}}} + m_h^{-\frac{1}{2}\frac{\beta_{h,2}-1}{\beta_{h,2}+p_{h,2}}} m_\varphi^{-\frac{1}{2}\frac{\beta_{\varphi,2}-1}{\beta_{\varphi,2}+p_{\varphi,2}}}\right)$$

*Proof.* We combine the bounds in Lemmas (E.23) and (E.24), by using the convergence to the minimum norm RKHS norm bridge function solutions and discarding the faster terms, to obtain

$$|\theta_{\text{ATE}}(a) - \hat{\theta}_{\text{ATE}}(a)| \lesssim \|\bar{\varphi}_0 - \hat{\varphi}\|\|\bar{h}_0 - \hat{h}\| + \|C_{YZ|A} - \hat{C}_{YZ|A}\|$$
$$+ \|\hat{\mu}_{\mathcal{W}} - \mu_{\mathcal{W}}\| + \|C_{ZW|A} - \hat{C}_{ZW|A}\|.$$

Note that the term $\|\hat{\mu}_{\mathcal{W}} - \mu_{\mathcal{W}}\|$ converges with $t^{-1/2}$ rate [11] hence can be discarded as well. The terms $\|C_{YZ|A} - \hat{C}_{YZ|A}\|$ and $\|C_{ZW|A} - \hat{C}_{ZW|A}\|$ converges with $O_p\left(t^{-\frac{1}{2}\frac{\beta_{\varphi,3}-1}{\beta_{\varphi,3}+p_{\varphi,3}}}\right)$ with the given regularizer parameters $\lambda_{\varphi,3}$ and $\lambda_{\text{DR}}$ in Theorems (E.25) and (E.26). Hence, the convergence will be governed by

$$|\theta_{\text{ATE}}(a) - \hat{\theta}_{\text{ATE}}(a)| \lesssim \|\bar{\varphi}_0 - \hat{\varphi}\|\|\bar{h}_0 - \hat{h}\| + t^{-\frac{1}{2}\frac{\beta_{\varphi,3}-1}{\beta_{\varphi,3}+p_{\varphi,3}}}.$$

Appealing to the conditions in Theorems (E.7) and (E.13), we will have four conditions depending on the first and second stage data splitting conditions of KPV and KAP algorithms:

i. Suppose $\iota_h \leq \frac{\beta_{h,2}+1}{\beta_{h,2}+p_{h,2}}$ and $\iota_\varphi \leq \frac{\beta_{\varphi,2}+1}{\beta_{\varphi,2}+p_{\varphi,2}}$. Then, condition [i] in Theorem (E.13) and condition [i] in Theorem (E.7) apply. Combining these bound gives

$$|\theta_{\text{ATE}}(a) - \hat{\theta}_{\text{ATE}}(a)| = O_p\left( t^{-\frac{1}{2}\frac{\beta_{\varphi,3}-1}{\beta_{\varphi,3}+p_{\varphi,3}}} + m_h^{-\frac{\iota_h}{2}\frac{\beta_{h,2}-1}{\beta_{h,2}+1}} m_\varphi^{-\frac{\iota_\varphi}{2}\frac{\beta_{\varphi,2}-1}{\beta_{\varphi,2}+1}} \right),$$

with $\lambda_{h,2} = \Theta\left(m^{-\frac{\iota_h}{\beta_{h,2}+1}}\right)$ and $\lambda_{\varphi,2} = \Theta\left(m_\varphi^{-\frac{\iota_\varphi}{\beta_{\varphi,2}+1}}\right)$.

ii. Suppose $\iota_h \leq \frac{\beta_{h,2}+1}{\beta_{h,2}+p_{h,2}}$ and $\iota_\varphi \geq \frac{\beta_{\varphi,2}+1}{\beta_{\varphi,2}+p_{\varphi,2}}$. Then, condition [i] in Theorem (E.13) and condition [ii] in Theorem (E.7) apply. Combining these bound gives

$$|\theta_{\text{ATE}}(a) - \hat{\theta}_{\text{ATE}}(a)| = O_p\left( t^{-\frac{1}{2}\frac{\beta_{\varphi,3}-1}{\beta_{\varphi,3}+p_{\varphi,3}}} + m_h^{-\frac{\iota_h}{2}\frac{\beta_{h,2}-1}{\beta_{h,2}+1}} m_\varphi^{-\frac{1}{2}\frac{\beta_{\varphi,2}-1}{\beta_{\varphi,2}+p_{\varphi,2}}} \right).$$

with $\lambda_{h,2} = \Theta\left(m^{-\frac{\iota_h}{\beta_{h,2}+1}}\right)$ and $\lambda_{\varphi,2} = \Theta\left(m_\varphi^{-\frac{1}{\beta_{\varphi,2}+p_{\varphi,2}}}\right)$.

iii. Suppose $\iota_h \geq \frac{\beta_{h,2}+1}{\beta_{h,2}+p_{h,2}}$ and $\iota_\varphi \leq \frac{\beta_{\varphi,2}+1}{\beta_{\varphi,2}+p_{\varphi,2}}$. Then, condition [ii] in Theorem (E.13) and condition [i] in Theorem (E.7) apply. Combining these bound gives

$$|\theta_{\text{ATE}}(a) - \hat{\theta}_{\text{ATE}}(a)| = O_p\left( t^{-\frac{1}{2}\frac{\beta_{\varphi,3}-1}{\beta_{\varphi,3}+p_{\varphi,3}}} + m_h^{-\frac{1}{2}\frac{\beta_{h,2}-1}{\beta_{h,2}+p_{h,2}}} m_\varphi^{-\frac{\iota_\varphi}{2}\frac{\beta_{\varphi,2}-1}{\beta_{\varphi,2}+1}} \right),$$

with $\lambda_{h,2} = \Theta\left(m_h^{-\frac{1}{\beta_{h,2}+p_{h,2}}}\right)$ and $\lambda_{\varphi,2} = \Theta\left(m_\varphi^{-\frac{\iota_\varphi}{\beta_{\varphi,2}+1}}\right)$.

iv. Suppose $\iota_h \geq \frac{\beta_{h,2}+1}{\beta_{h,2}+p_{h,2}}$ and $\iota_\varphi \geq \frac{\beta_{\varphi,2}+1}{\beta_{\varphi,2}+p_{\varphi,2}}$. Then, condition [ii] in Theorem (E.13) and condition [ii] in Theorem (E.7) apply. Combining these bound gives

$$|\theta_{\text{ATE}}(a) - \hat{\theta}_{\text{ATE}}(a)| = O_p\left( t^{-\frac{1}{2}\frac{\beta_{\varphi,3}-1}{\beta_{\varphi,3}+p_{\varphi,3}}} + m_h^{-\frac{1}{2}\frac{\beta_{h,2}-1}{\beta_{h,2}+p_{h,2}}} m_\varphi^{-\frac{1}{2}\frac{\beta_{\varphi,2}-1}{\beta_{\varphi,2}+p_{\varphi,2}}} \right).$$

with $\lambda_{h,2} = \Theta\left(m_h^{-\frac{1}{\beta_{h,2}+p_{h,2}}}\right)$ and $\lambda_{\varphi,2} = \Theta\left(m_\varphi^{-\frac{1}{\beta_{\varphi,2}+p_{\varphi,2}}}\right)$.

$\square$

**Theorem E.28.** *Suppose the assumptions in Theorems (E.7), (E.17), (E.25), and (E.26) hold. For a given training dataset $\mathcal{D} = \{y_i, w_i, z_i, a_i\}_{i=1}^t$, let $\{n_\varphi, m_\varphi, t_\varphi\}$ denote the number of samples used in first-, second-, and third-stage regressions of KAP algorithm with $t_\varphi = t$. Then, for DRPMMR algorithm with high probability*

i. *If $\iota_\varphi \leq \frac{\beta_{\varphi,2}+1}{\beta_{\varphi,2}+p_{\varphi,2}}$, set $\lambda_{\varphi,2} = \Theta\left(m_\varphi^{-\frac{\iota_\varphi}{\beta_{\varphi,2}+1}}\right)$. Then,*

$$|\theta_{ATE}(a) - \hat{\theta}_{ATE}(a)| = O_p\left( t^{-\frac{1}{2}\frac{\beta_{\varphi,3}-1}{\beta_{\varphi,3}+p_{\varphi,3}}} + t^{-\frac{1}{2}\frac{\gamma-1}{\gamma+1}} m_\varphi^{-\frac{\iota_\varphi}{2}\frac{\beta_{\varphi,2}-1}{\beta_{\varphi,2}+p_{\varphi,2}}} \right),$$

ii. If $\iota_\varphi \geq \frac{\beta_{\varphi,2}+1}{\beta_{\varphi,2}+p_{\varphi,2}}$, set $\lambda_{\varphi,2} = \Theta\left(m_\varphi^{-\frac{1}{\beta_{\varphi,2}+p_{\varphi,2}}}\right)$. Then,

$$|\theta_{ATE}(a) - \hat\theta_{ATE}(a)| = O_p\left(t^{-\frac{1}{2}\frac{\beta_{\varphi,3}-1}{\beta_{\varphi,3}+p_{\varphi,3}}} + t^{-\frac{1}{2}\frac{\gamma-1}{\gamma+1}}m_\varphi^{-\frac{1}{2}\frac{\beta_{\varphi,2}-1}{\beta_{\varphi,2}+p_{\varphi,2}}}\right),$$

*Proof.* Combining the bounds from Lemmas (E.23) and (E.24), and using the convergence of the estimators to the minimum-norm RKHS bridge function solutions, we obtain the following by discarding higher-order terms:

Similar to the proof of Theorem (E.27), the bound will be governed by

$$|\theta_{\text{ATE}}(a) - \hat\theta_{\text{ATE}}(a)| \lesssim \|\bar\varphi_0 - \hat\varphi\|\|\check h_0 - \hat h\| + t^{-\frac{1}{2}\frac{\beta_{\varphi,3}-1}{\beta_{\varphi,3}+p_{\varphi,3}}}.$$

Now, using the conditions in Theorem (E.7) and the rate in Theorem (E.17), we will have two conditions:

i. Suppose $\iota_\varphi \leq \frac{\beta_{\varphi,2}+1}{\beta_{\varphi,2}+p_{\varphi,2}}$, then condition [i] in Theorem (E.13) applies. Hence,

$$|\theta_{\text{ATE}}(a) - \hat\theta_{\text{ATE}}(a)| = O_p\left(t^{-\frac{1}{2}\frac{\beta_{\varphi,3}-1}{\beta_{\varphi,3}+p_{\varphi,3}}} + t^{-\frac{1}{2}\frac{\gamma-1}{\gamma+1}}m_\varphi^{-\frac{\iota_\varphi}{2}\frac{\beta_{\varphi,2}-1}{\beta_{\varphi,2}+p_{\varphi,2}}}\right),$$

with $\lambda_{\varphi,2} = \Theta\left(m_\varphi^{-\frac{\iota_\varphi}{\beta_{\varphi,2}+1}}\right)$.

ii. Suppose $\iota_\varphi \geq \frac{\beta_{\varphi,2}+1}{\beta_{\varphi,2}+p_{\varphi,2}}$, then condition [ii] in Theorem (E.13) applies. Hence,

$$|\theta_{\text{ATE}}(a) - \hat\theta_{\text{ATE}}(a)| = O_p\left(t^{-\frac{1}{2}\frac{\beta_{\varphi,3}-1}{\beta_{\varphi,3}+p_{\varphi,3}}} + t^{-\frac{1}{2}\frac{\gamma-1}{\gamma+1}}m_\varphi^{-\frac{1}{2}\frac{\beta_{\varphi,2}-1}{\beta_{\varphi,2}+p_{\varphi,2}}}\right),$$

with $\lambda_{\varphi,2} = \Theta\left(m_\varphi^{-\frac{1}{\beta_{\varphi,2}+p_{\varphi,2}}}\right)$.

$\square$

**Remark E.29** (Curse of Dimensionality). *Our proposed algorithms (DRKPV and DRPMMR), along with KPV and KAP methods, rely on multi-stage kernel ridge regressions. Kernel Ridge Regression is known to achieve minimax-optimal rates in moderate dimension, but in very high-dimensional regimes, particularly when the input dimension $d$ grows with the sample size $t$ (i.e., $d/t^\beta \to c$ for some $\beta \in (0,1)$), the situation becomes more subtle [46, 45].*

*Specifically, the effectiveness of KRR suffers from two primary issues in high dimensions:*

- *When functions lie in Sobolev classes, the achievable rate explicitly depends on both smoothness and dimension. Intuitively, the regression function must be smooth enough relative to the ambient dimension $d$ for KRR to remain consistent. More specifically, Fischer and Steinwart [67, Corollary 5 & 6] (for standard scalar KRR) and Li et al. [45, Corollary 1 & 2] (for vector-valued KRR) show that if the target function has smoothness parameter $s$ (in Sobolev sense), the optimal rate of convergence in $L_2$ norm is $O(t^{-\frac{s}{s+d/2}})$. This rate clearly demonstrates the curse of dimensionality, as for a fixed $s$, the bound becomes vacuous as $d \to \infty$.*

- *Donhauser et al. [46] show that for rotationally invariant kernels (such as RBF or Matérn), a polynomial approximation barrier arises: the learned function is effectively restricted to low-degree polynomials as $d$ grows, regardless of eigenvalue decay. This implies that consistency in high dimensions is limited unless additional structural assumptions are imposed.*

*In our work, we do not address the curse of dimensionality. Instead, our contribution is to show that doubly robust PCL estimators can be constructed without density ratio estimation and kernel-smoothing, thereby extending practical applicability to continuous and high-dimensional treatments*

*where prior DR methods [21, 22] fail. Our theorems remain valid in high-dimensional settings provided the smoothness and effective RKHS dimension assumptions hold. However, we acknowledge that when input dimension grows with sample size, methods based on standard RBF or Matérn kernels may indeed fail. Addressing this deeper theoretical limitation is left as a future work.*

**Remark E.30** (Asymptotic Efficiency and Normality). *While our identification formula in Theorem (2.7) is derived from the Efficient Influence Function (EIF), our resulting estimator is not a classical one-step estimator [47, 48] and does not automatically inherit local efficiency or asymptotic normality [28]. Unlike standard EIF-based estimators that leverage first-order orthogonality [30] between two coupled nuisance components (e.g., outcome regression and propensity score), our method involves three components—the outcome bridge, treatment bridge, and a correction term—each estimated separately. As a result, the bias terms accumulate additively, rather than canceling multiplicatively, as in traditional doubly robust estimators. Establishing local efficiency would thus require further analysis, including proving asymptotic linearity, implementing cross-fitting, and verifying attainment of the semiparametric variance bound—steps we leave for future work. Similarly, asymptotic normality is not guaranteed; although the estimator is consistent and derived from an EIF, a central limit theorem would require controlling higher-order remainder terms across all three nuisance components—particularly challenging in our framework. We therefore refrain from claiming asymptotic normality in the present version. Given the nonregularity typical of continuous treatment settings [59, 60, 61], a slower-than-$\sqrt{n}$ convergence rate is generally expected for the parameter, a point supported by both theory and our convergence results.*

# F  Supplementary on numerical experiments

Here, we provide additional details on the numerical experiments presented in Section (5), including hyperparameter optimization procedures and supplementary experimental results.

## F.1  Kernel

We utilize the Gaussian (RBF) kernel for our experiments, defined as

$$k_{\mathcal{F}}(f_i, f_j) = \exp\left(\frac{-\|f_i - f_j\|_2^2}{2l^2}\right) \tag{34}$$

for $f_i, f_j \in \mathbb{R}^{d_{\mathcal{F}}}$. This kernel is widely used due to its boundedness, continuity, and characteristic property [68]. The parameter $l > 0$, known as the length scale, controls the smoothness of the kernel. We set $l$ using the commonly adopted *median heuristic*, which sets $l^2$ to half the median of the pairwise squared Euclidean distances in the dataset $\{f_i\}_{i=1}^n$, that is:

$$l^2 = \frac{1}{2}\text{median}(\{\|f_i - f_j\|_2^2 : 1 \le i < j \le n\}).$$

This approach has been frequently used in causal inference applications, including Singh et al. [69], Mastouri et al. [11], Singh [12], Xu and Gretton [62], Bozkurt et al. [23].

In addition, we consider a dimension-wise variant of the Gaussian kernel, defined as the product of one-dimensional Gaussian kernels applied to each coordinate:

$$k_{\mathcal{F}}(f_i, f_j) = \prod_{k=1}^{d_{\mathcal{F}}} \exp\left(\frac{-\|f_i^{(k)} - f_j^{(k)}\|_2^2}{2l^{(k)^2}}\right) \tag{35}$$

where $f_i^{(k)}$ denotes the $k$-th coordinate of vector $f_i$. Each length scale $l^{(k)}$ can be set independently using the median heuristic applied to that specific dimension. We refer to the kernel in Equation (35) as the *columnwise Gaussian kernel*.

Following the setup in Bozkurt et al. [23], we apply the columnwise Gaussian kernel to the outcome proxy variable $W$ within the KAP algorithm for the synthetic low-dimensional experiment in Section (5). For all other experiments and for all variables used in both outcome and treatment bridge-based methods, we use the standard Gaussian kernel defined in Equation (34).

We also consider the Matérn kernel which provides a flexible alternative to the Gaussian kernel by introducing a smoothness parameter. The Matérn kernel between two points $f_i$ and $f_j$ is given by [70]

$$k_{\mathcal{F},\nu}(f_i, f_j) = \frac{2^{1-\nu}}{\Gamma(\nu)} \left( \frac{\sqrt{2\nu}}{l} \|f_i - f_j\|_2 \right)^\nu \mathcal{K}_\nu \left( \frac{\sqrt{2\nu}}{l} \|f_i - f_j\|_2 \right). \tag{36}$$

Here, the parameters $\nu$ and $l$ are positive variables, $\Gamma(\cdot)$ is the gamma function, and $\mathcal{K}_\nu(\cdot)$ is modified Bessel function of the second kind [71]. The Matérn family of kernels satisfies the assumptions on kernels, including those in Assumption (3.1) [72]. The parameter $\nu$ directly controls the differentiability of the kernel function; as $\nu \to \infty$, the kernel converges to the Gaussian kernel in Equation (34).

For practical implementation and to explore varying levels of smoothness, we use cases where $\nu$ is a half integer in the form $\nu = p + 1/2$, which yield simplified, closed-form polynomial expressions [70]:

$$k_{\mathcal{F},\nu=p+1/2}(f_i, f_j) = \exp \left( -\frac{\sqrt{2\nu}}{l} \|f_i - f_j\| \right) \frac{\Gamma(p+1)}{\Gamma(2p+1)} \sum_{k=0}^{p} \frac{(p+k)!}{k!(p-k)!} \left( \frac{\sqrt{8\nu}}{l} \|f_i - f_j\| \right)^{p-k}. \tag{37}$$

In our ablation studies in Section (S.M. (F.7)), we test the performance of our method across Matérn kernels corresponding to different integer values of $p$.

## F.2 Discussion on the data splitting of KPV and KAP

We detail the data splitting strategy employed for the multi-stage KPV and KAP algorithms, which derive their samples from the full training observations $\{y_i, w_i, z_i, a_i\}_{i=1}^{t}$.

| Algorithm Stage | Samples Used | Implementation Details |
|---|---|---|
| **KPV (Stage 1)** | $\{\bar{w}_i, \bar{z}_i, \bar{a}_i\}_{i=1}^{n_h}$ | Uses $n_h = \lfloor t/2 \rfloor$ random samples. |
| **KPV (Stage 2)** | $\{\tilde{y}_i, \tilde{z}_i, \tilde{a}_i\}_{i=1}^{m_h}$ | Uses $m_h = t - n_h$. Samples are **disjoint** from Stage 1. |
| **KPV (Stage 3)** | $\{\dot{w}_i\}_{i=1}^{t_h}$ | Uses the full set: $t_h = t$. |
| **KAP (Stage 1)** | $\{\bar{w}_i, \bar{z}_i, \bar{a}_i\}_{i=1}^{n_\varphi}$ | Uses $n_\varphi = \lfloor t/2 \rfloor$ random samples. |
| **KAP (Stage 2)** | $\{\tilde{w}_i, \tilde{a}_i\}_{i=1}^{m_\varphi}$ | Uses $m_\varphi = t - n_\varphi$. Samples are **disjoint** from Stage 1. |
| **KAP (Stage 3)** | $\{\dot{y}_i, \dot{z}_i, \dot{a}_i\}_{i=1}^{t_\varphi}$ | Uses the full set: $t_\varphi = t$. |

While we use this structured splitting in our implementation, our consistency proofs in Section (E) do not assume disjoint splits, similar to [11, 12, 23]. Indeed, using the full dataset per stage still retains theoretical consistency (e.g., Corollary 1 in [12] for outcome bridge-based methods).

However, structured data splitting is useful in practice for two main reasons:

1. **Hyperparameter Tuning:** For regression stages lacking a closed-form LOOCV solution (such as the second stage of both KPV and KAP), using the first-stage data as a held-out set provides a clean and non-overlapping validation loss for regularizer tuning. See Section (S.M. F.3) for the corresponding discussion.

2. **Observation Types:** Splitting naturally accommodates the structure of the different marginal distributions required across stages. For instance, the KAP first-stage utilizes the marginal distribution of $(W, Z, A)$, whereas the third-stage requires data from the distribution of $(Y, Z, A)$, facilitating estimation when only partial marginals are available.

## F.3 Hyperparameter optimization procedures

We describe the procedures used to tune the regularization parameters $\lambda_{h,1}$, $\lambda_{h,2}$, $\lambda_{\varphi,1}$, $\lambda_{\varphi,2}$, $\lambda_{\varphi,3}$, $\lambda_{\text{MMR}}$, and $\lambda_{\text{DR}}$. Specifically, $\lambda_{h,1}$, $\lambda_{\varphi,1}$, $\lambda_{\varphi,3}$, and $\lambda_{\text{DR}}$ are selected using leave-one-out cross-validation (LOOCV), which admits a closed-form solution in the case of kernel ridge regression. In

contrast, $\lambda_{h,2}$, $\lambda_{\varphi,2}$ and $\lambda_{\text{MMR}}$ are tuned using validation loss on a held-out set. To avoid repetition, we first review the LOOCV procedure in the general kernel ridge regression setting, followed by the closed-form expressions for the relevant regression stages of the KPV and KAP algorithms. We then present the validation loss formulas used to tune $\lambda_{\varphi,2}$ and $\lambda_{\text{MMR}}$.

**Leave one out cross validation in kernel ridge regression**: We consider the problem of estimating the conditional mean function $f_0(x) = \mathbb{E}[Y \mid X = x]$ from an observational data $\{x_i, y_i\}_{i=1}^{t} \subset \mathbb{R}^{d_{\mathcal{X}}} \times \mathbb{R}^{d_{\mathcal{Y}}}$, denote the dimensions of inputs and outputs, respectively. The kernel ridge regression (KRR) estimator for $f_0$ is given by

$$\hat{f} = \underset{f \in \mathcal{H}_{\mathcal{X}}}{\arg\min} \frac{1}{t} \sum_{i=1}^{t} \|y_i - \langle f, \phi_{\mathcal{X}}(x_i) \rangle_{\mathcal{H}_{\mathcal{X}}}\|_{\mathcal{Y}}^2 + \lambda \|f\|_{\mathcal{H}_{\mathcal{X}}}, \tag{38}$$

where $\mathcal{H}_{\mathcal{X}}$ is a reproducing kernel Hilbert space (RKHS) on domain $\mathcal{X}$ with the associated canonical feature map $\phi_{\mathcal{X}}(\cdot) : \mathcal{X} \to \mathcal{H}_{\mathcal{X}}$, and $\lambda > 0$ is the regularization parameter. The closed-form solution to Equation (38) is

$$\hat{f} = \boldsymbol{Y}^{\top} (\boldsymbol{K}_{XX} + t\lambda \boldsymbol{I})^{-1} \Phi_{\mathcal{X}}, \tag{39}$$

where $\boldsymbol{Y} = [y_1 \ \ldots \ y_t]^{\top}$, $\Phi_{\mathcal{X}} = [\phi_{\mathcal{X}}(x_1) \ \ldots \ \phi_{\mathcal{X}}(x_t)]$, and $\boldsymbol{K}_{XX}$ is the kernel matrix computed from the inputs $\{x_i\}_{i=1}^{t}$. To select an appropriate value for $\lambda$, we employ leave-one-out cross-validation (LOOCV), which assesses generalization by sequentially excluding each data point and evaluating prediction error. The LOOCV objective is defined as

$$\text{LOOCV}(\lambda) = \frac{1}{t} \sum_{j=1}^{t} \left\| y_j - \hat{f}_{-j}(x_i) \right\|_{\mathcal{Y}}^2, \tag{40}$$

where $\hat{f}_{-j}$ KRR estimator trained on all data except the $j$-th observation. In the KRR setting, the LOOCV loss admits a closed-form expression, as given by the following result:

**Theorem F.1** (Algorithm (F.1) in [69]). *Consider the kernel ridge regression setup introduced in Equation (38) where $\{x_i\}_{i=1}^{t}$ denotes the input data and $\{y_i\}_{i=1}^{t}$ denotes the corresponding outputs. Then, the LOOCV loss is given by*

$$LOOCV(\lambda) = \frac{1}{t} \|\tilde{\boldsymbol{H}}_{\lambda}^{-1} \boldsymbol{H}_{\lambda} \boldsymbol{Y}\|_{\mathcal{Y}}^2 = \frac{1}{t} \text{Tr}\left( \tilde{\boldsymbol{H}}_{\lambda}^{-1} \boldsymbol{H}_{\lambda} \boldsymbol{Y} \boldsymbol{Y}^{\top} \boldsymbol{H}_{\lambda}^{\top} \tilde{\boldsymbol{H}}_{\lambda}^{-\top} \right), \tag{41}$$

*where*

$$\boldsymbol{H}_{\lambda} = \boldsymbol{I} - \boldsymbol{K}_{XX}(\boldsymbol{K}_{XX} + n\lambda \boldsymbol{I})^{-1} \in \mathbb{R}^{t \times t}, \quad \tilde{\boldsymbol{H}}_{\lambda} = diag(\boldsymbol{H}_{\lambda}) \in \mathbb{R}^{t \times t}.$$

The regularization parameter $\lambda$ can then be selected by minimizing the LOOCV loss over a predefined grid $\Lambda \subset \mathbb{R}$:

$$\lambda^* = \underset{\Lambda \subset \mathbb{R}}{\arg\min} \frac{1}{t} \|\tilde{\boldsymbol{H}}_{\lambda}^{-1} \boldsymbol{H}_{\lambda} \boldsymbol{Y}\|_{\mathcal{Y}}^2.$$

The proof of Theorem (F.1) can be found in Singh et al. [69] (see Algorithm (F.1)). We apply Theorem (F.1) to tune the regularization parameters $\lambda_{h,1}$, $\lambda_{\varphi,1}$, $\lambda_{\varphi,3}$, and $\lambda_{\text{DR}}$.

### F.3.1 Hyperparameter selection for $\lambda_{h,1}$, $\lambda_{\varphi,1}$, $\lambda_{\varphi,3}$, and $\lambda_{\text{DR}}$

Below, we present the application of the LOOCV tuning procedure to the regularization parameters $\lambda_{h,1}$, $\lambda_{\varphi,1}$, $\lambda_{\varphi,3}$, and $\lambda_{\text{DR}}$.

- **KPV first-stage regression:** The first-stage regression in the KPV algorithm, given samples $\{\bar{w}_i, \bar{z}_i, \bar{a}_i\}_{i=1}^{n_h}$, is a kernel ridge regression from inputs $\{\phi_{\mathcal{Z}}(\bar{z}_i) \otimes \phi_{\mathcal{A}}(\bar{a}_i)\}_{i=1}^{n_h}$ to the outcomes $\{\phi_{\mathcal{W}}(\bar{w}_i)\}_{i=1}^{n_h}$. Therefore, the LOOCV loss for $\lambda_{h,1}$ is given by

$$\text{LOOCV}(\lambda_{h,1}) = \frac{1}{n_h} \text{Tr}\left( \tilde{\boldsymbol{H}}_{\lambda_{h,1}}^{-1} \boldsymbol{H}_{\lambda_{h,1}} \boldsymbol{K}_{\bar{W}\bar{W}} \boldsymbol{H}_{\lambda_{h,1}}^{\top} \tilde{\boldsymbol{H}}_{\lambda_{h,1}}^{-\top} \right),$$

where

$$\boldsymbol{H}_{\lambda_{h,1}} = \boldsymbol{I} - (\boldsymbol{K}_{\bar{Z}\bar{Z}} \odot \boldsymbol{K}_{\bar{A}\bar{A}})(\boldsymbol{K}_{\bar{Z}\bar{Z}} \odot \boldsymbol{K}_{\bar{A}\bar{A}} + n_h \lambda_{h,1} \boldsymbol{I})^{-1} \in \mathbb{R}^{n_h \times n_h},$$

$$\tilde{\boldsymbol{H}}_{\lambda_{h,1}} = \mathrm{diag}(\boldsymbol{H}_{\lambda_{h,1}}) \in \mathbb{R}^{n_h \times n_h}.$$

We use a logarithmically spaced grid of 25 values in the range $[5 \times 10^{-5}, 1]$ and select the value of $\lambda_{h,1}$ that minimizes the LOOCV loss.

- **KAP first-stage regression:** The first-stage regression in the KAP algorithm, using samples $\{\bar{w}_i, \bar{z}_i, \bar{a}_i\}_{i=1}^{n_\varphi}$, is a kernel ridge regression from inputs $\{\phi_{\mathcal{W}}(\bar{w}_i) \otimes \phi_{\mathcal{A}}(\bar{a}_i)\}_{i=1}^{n_\varphi}$ to the outcomes $\{\phi_{\mathcal{Z}}(\bar{z}_i)\}_{i=1}^{n_\varphi}$. Hence, the LOOCV loss for $\lambda_{\varphi,1}$ is given by

$$\mathrm{LOOCV}(\lambda_{\varphi,1}) = \frac{1}{n_\varphi} \mathrm{Tr}\left(\tilde{\boldsymbol{H}}_{\lambda_{\varphi,1}}^{-1} \boldsymbol{H}_{\lambda_{\varphi,1}} \boldsymbol{K}_{\bar{Z}\bar{Z}} \boldsymbol{H}_{\lambda_{\varphi,1}}^{\top} \tilde{\boldsymbol{H}}_{\lambda_{\varphi,1}}^{-\top}\right),$$

where

$$\boldsymbol{H}_{\lambda_{\varphi,1}} = \boldsymbol{I} - (\boldsymbol{K}_{\bar{W}\bar{W}} \odot \boldsymbol{K}_{\bar{A}\bar{A}})(\boldsymbol{K}_{\bar{W}\bar{W}} \odot \boldsymbol{K}_{\bar{A}\bar{A}} + n_\varphi \lambda_{\varphi,1} \boldsymbol{I})^{-1} \in \mathbb{R}^{n_\varphi \times n_\varphi},$$

$$\tilde{\boldsymbol{H}}_{\lambda_{\varphi,1}} = \mathrm{diag}(\boldsymbol{H}_{\lambda_{\varphi,1}}) \in \mathbb{R}^{n_\varphi \times n_\varphi}.$$

We perform a grid search over 25 logarithmically spaced values in the range $[5 \times 10^{-5}, 1]$ and select the value of $\lambda_{\varphi,1}$ that minimizes the LOOCV loss.

- **KAP third-stage regression:** With the third-stage data $\{\dot{y}_i, \dot{z}_i, \dot{a}_i\}_{i=1}^{t_\varphi}$, the KAP algorithm performs kernel ridge regression from the inputs $\{\phi_{\mathcal{A}}(\dot{a}_i)\}_{i=1}^{t_\varphi}$ to the outcomes $\{\dot{y}_i \phi_{\mathcal{Z}}(\dot{z}_i)\}_{i=1}^{t_\varphi}$. Thus, the LOOCV loss for $\lambda_{\varphi,3}$ is given by

$$\mathrm{LOOCV}(\lambda_{\varphi,3}) = \frac{1}{t_\varphi} \mathrm{Tr}\left(\tilde{\boldsymbol{H}}_{\lambda_{\varphi,3}}^{-1} \boldsymbol{H}_{\lambda_{\varphi,3}} \left(\boldsymbol{K}_{\dot{Z}\dot{Z}} \odot \dot{\boldsymbol{Y}}\dot{\boldsymbol{Y}}^{\top}\right) \boldsymbol{H}_{\lambda_{\varphi,3}}^{\top} \tilde{\boldsymbol{H}}_{\lambda_{\varphi,3}}^{-\top}\right),$$

where

$$\boldsymbol{H}_{\lambda_{\varphi,3}} = \boldsymbol{I} - \boldsymbol{K}_{\dot{A}\dot{A}}(\boldsymbol{K}_{\dot{A}\dot{A}} + t_\varphi \lambda_{\varphi,3} \boldsymbol{I})^{-1} \in \mathbb{R}^{t_\varphi \times t_\varphi},$$

$$\tilde{\boldsymbol{H}}_{\lambda_{\varphi,3}} = \mathrm{diag}(\boldsymbol{H}_{\lambda_{\varphi,3}}) \in \mathbb{R}^{t_\varphi \times t_\varphi}.$$

We perform a grid search over 25 logarithmically spaced values in the range $[5 \times 10^{-5}, 1]$ and select the value of $\lambda_{\varphi,3}$ that minimizes the LOOCV loss.

- **Slack term estimation:** Our doubly robust estimator includes a term of the form $\mathbb{E}[\varphi_0(Z,a)h_0(W,a) \mid A = a]$, which requires estimating the conditional mean embedding $E[\phi_{\mathcal{Z}}(Z) \otimes \phi_{\mathcal{W}}(W) \mid A = a]$ as we derive in Section (S.M. C.3). Using training data $\{z_i, w_i, a_i\}_i^t$, we fit a kernel ridge regression from the inputs $\{\phi_{\mathcal{A}}(a_i)\}_{i=1}^t$ to the outputs $\{\phi_{\mathcal{Z}}(z_i) \otimes \phi_{\mathcal{W}}(w_i)\}_{i=1}^t$ with the regularization parameter $\lambda_{\mathrm{DR}}$. As a result, the LOOCV for $\lambda_{\mathrm{DR}}$ is given by

$$\mathrm{LOOCV}(\lambda_{\mathrm{DR}}) = \frac{1}{t} \mathrm{Tr}\left(\tilde{\boldsymbol{H}}_{\lambda_{\mathrm{DR}}}^{-1} \boldsymbol{H}_{\lambda_{\mathrm{DR}}} \left(\boldsymbol{K}_{ZZ} \odot \boldsymbol{K}_{WW}\right) \boldsymbol{H}_{\lambda_{\mathrm{DR}}}^{\top} \tilde{\boldsymbol{H}}_{\lambda_{\mathrm{DR}}}^{-\top}\right),$$

where

$$\boldsymbol{H}_{\lambda_{\mathrm{DR}}} = \boldsymbol{I} - \boldsymbol{K}_{AA}(\boldsymbol{K}_{AA} + t\lambda_{\mathrm{DR}} \boldsymbol{I})^{-1} \in \mathbb{R}^{t \times t},$$

$$\tilde{\boldsymbol{H}}_{\lambda_{\mathrm{DR}}} = \mathrm{diag}(\boldsymbol{H}_{\lambda_{\mathrm{DR}}}) \in \mathbb{R}^{t \times t}.$$

For $\lambda_{\mathrm{DR}}$, we use a grid search over 25 logarithmically spaced values in the range $[5 \times 10^{-5}, 1]$ to minimize the LOOCV loss.

### F.3.2 Hyperparameter selection for $\lambda_{h,2}$, $\lambda_{\varphi,2}$, and $\lambda_{\mathrm{MMR}}$

In this section, we provide details of the tuning procedures of the other regularization parameters $\lambda_{h,2}$, $\lambda_{\varphi,2}$, and $\lambda_{\mathrm{MMR}}$. Specifically, we leverage the validation loss on a held-out set to tune these parameters.

- **KPV second-stage regression:** The second-stage regression in KPV approximates the outcome bridge function via optimizing the objective given in Equation (20) To tune the regularization

parameter $\lambda_{h,2}$, we treat the first-stage samples $\{\bar{w}_i, \bar{z}_i, \bar{a}_i\}_{i=1}^{n_h}$ as a held-out validation set and minimize the validation loss:

$$\lambda_{h,2}^* = \arg\min \frac{1}{n_h} \sum_{i=1}^{n_h} \left( \bar{y}_i - \langle \hat{h}, \hat{\mu}_{W|Z,A}(\bar{z}_i, \bar{a}_i) \otimes \phi_{\mathcal{A}}(\bar{a}_i) \rangle \right)^2,$$

where $\hat{h}$ is the solution to the optimization problem in Equation (20), as described in Algorithm (2), and $\hat{\mu}_{W|Z,A}$ denotes the estimated conditional mean embedding from the first stage.

In our experiments, we used a grid of 25 logarithmically spaced values in the range $[5 \times 10^{-5}, 1]$.

- **KAP second-stage regression:** The second-stage regression of KAP estimates the treatment bridge function via optimizing the objective given in Equation (26). To tune the regularization parameter $\lambda_{\varphi,2}$, we treat the first-stage samples $\{\bar{w}_i, \bar{z}_i, \bar{a}_i\}_{i=1}^{n_\varphi}$ as a held-out validation set and minimize the validation loss:

$$\hat{\mathcal{L}}(\varphi)_{\text{KAP}}^{\text{Val}} = \frac{1}{n_\varphi} \sum_{i=1}^{n_\varphi} \langle \varphi, \hat{\mu}_{Z|W,A}(\bar{w}_i, \bar{a}_i) \otimes \phi_{\mathcal{A}}(\bar{a}_i) \rangle_{\mathcal{H}_{\mathcal{Z}\mathcal{A}}}^2$$

$$- 2\frac{1}{m_\varphi(n_\varphi - 1)} \sum_{i=1}^{n_\varphi} \sum_{\substack{j=1 \\ j\neq i}}^{m_\varphi} \left\langle \varphi, \hat{\mu}_{Z|W,A}(\bar{w}_j, \bar{a}_i) \otimes \phi_{\mathcal{A}}(\bar{a}_i) \right\rangle_{\mathcal{H}_{\mathcal{Z}\mathcal{A}}}.$$

This objective admits a closed-form expression:

$$\hat{\mathcal{L}}_{\text{KAP}}^{\text{Val}}(\varphi) = \frac{1}{n} \begin{bmatrix} \alpha_{1:m_\varphi} \\ \alpha_{m_\varphi+1} \end{bmatrix} \begin{bmatrix} \boldsymbol{B}^T \boldsymbol{K}_{ZZ} \boldsymbol{C} \odot \boldsymbol{K}_{\tilde{A}A} \\ (\frac{1}{m_\varphi})^T (\tilde{\boldsymbol{B}}^T \boldsymbol{K}_{ZZ} \boldsymbol{C} \odot \boldsymbol{K}_{\tilde{A}A}) \end{bmatrix} \begin{bmatrix} \boldsymbol{C}^T \boldsymbol{K}_{ZZ} \boldsymbol{B} \odot \boldsymbol{K}_{A\tilde{A}} \\ (\boldsymbol{C}^T \boldsymbol{K}_{ZZ} \tilde{\boldsymbol{B}} \odot \boldsymbol{K}_{A\tilde{A}}) \frac{1}{m_\varphi} \end{bmatrix}^\top \begin{bmatrix} \alpha_{1:m_\varphi} \\ \alpha_{m_\varphi+1} \end{bmatrix}$$

$$- 2 \begin{bmatrix} \alpha_{1:m_\varphi} \\ \alpha_{m_\varphi+1} \end{bmatrix}^\top \begin{bmatrix} (\boldsymbol{B}^\top \boldsymbol{K}_{ZZ} \bar{\boldsymbol{C}} \odot \boldsymbol{K}_{\tilde{A}A}) \frac{1}{n_\varphi} \\ (\frac{1}{m_\varphi})^\top (\tilde{\boldsymbol{B}}^\top \boldsymbol{K}_{ZZ} \bar{\boldsymbol{C}} \odot \boldsymbol{K}_{\tilde{A}A}) \frac{1}{n_\varphi} \end{bmatrix}.$$

Here, the matrices $\boldsymbol{C}$ and $\bar{\boldsymbol{C}}$ are defined as:

$$\boldsymbol{C} = \left( \boldsymbol{K}_{\bar{W}\bar{W}} \odot \boldsymbol{K}_{\bar{A}\bar{A}} + n_\varphi \lambda_{\varphi,1} \boldsymbol{I} \right)^{-1} (\boldsymbol{K}_{\bar{W}\bar{W}} \odot \boldsymbol{K}_{\bar{A}\bar{A}}),$$

$$\bar{\boldsymbol{C}}_{:,j} = \frac{1}{n_\varphi} \sum_{\substack{l=1 \\ l\neq j}}^{n_\varphi} \left( \boldsymbol{K}_{\bar{W}\bar{W}} \odot \boldsymbol{K}_{\bar{A}\bar{A}} + n_\varphi \lambda_{\varphi,1} \boldsymbol{I} \right)^{-1} (\boldsymbol{K}_{\bar{W}\bar{w}_l} \odot \boldsymbol{K}_{\bar{A}\bar{a}_j}), \quad \forall j.$$

For full derivation, see Bozkurt et al. [23, Section (13.2.2)]. To avoid overfitting, Bozkurt et al. [23] additionally propose minimizing the validation loss augmented by a model complexity penalty:

$$\lambda_{\varphi,2}^* = \arg\min \hat{\mathcal{L}}_{\text{KAP}}^{\text{Val}}(\varphi) + \frac{2\sigma_\varphi^2}{m_\varphi} \text{Tr}\left( \left( \boldsymbol{L}^T \boldsymbol{L} + m\lambda_{\varphi,2} \boldsymbol{I} \right)^{-1} \boldsymbol{L}^T \boldsymbol{L} \right).$$

for some fixed $\sigma_\varphi > 0$.

In our experiments, we used a grid of 25 logarithmically spaced values in the range $[5 \times 10^{-5}, 1]$ to tune $\lambda_{\varphi,2}$. Following the complexity regularization parameters in Bozkurt et al. [23], we set $\sigma_\varphi = 1$ or the synthetic low-dimensional setting as well as the legalized abortion and crime dataset, and $\sigma_\varphi = 3$ for dSprite and grade retention datasets.

- **PMMR regularization parameter tuning:** To tune the regularization parameter $\lambda_{\text{MMR}}$, we follow a procedure similar to that used in the second-stage regression of the KPV algorithm. Specifically, we set aside a small validation subset from the training data $\{y_i, w_i, z_i, a_i\}_{i=1}^t$ and evaluate the validation loss based on the bridge function predictions. We use a grid of 25 logarithmically spaced values in the range $[5 \times 10^{-5}, 10^{-3}]$ to tune $\lambda_{MMR}$, with $10\%$ of the training set held out as a validation set.

### F.4 Additional numerical experiments with misspecified bridge functions

A higher-resolution version of the misspecification analysis presented in Section (5) (Figure (3)) is provided below in Figure (4) for enhanced legibility.

Here, we conduct an additional ablation study to evaluate the robustness of our methods under misspecification of bridge functions, building on the experimental setup from Section (5). In this experiment, we introduce higher levels of noise into the bridge function coefficients. Recall that, by the representer theorem, the bridge functions admit the following forms:

- KPV: $\hat{h} = \sum_{i=1}^{n_h} \sum_{j=1}^{m_h} \alpha_{ij} \phi_{\mathcal{W}}(w_i) \otimes \phi_{\mathcal{A}}(\tilde{a}_j)$,
- PMMR: $\hat{h} = \sum_{i=1}^{t} \alpha_i \phi_{\mathcal{W}}(w_i) \otimes \phi_{\mathcal{A}}(a_i)$,
- KAP: $\hat{\varphi} = \sum_{l=1}^{m_\varphi} \sum_{t=1}^{m_\varphi} \gamma_{lt} \hat{\mu}_{Z|W,A}(\tilde{w}_t, \tilde{a}_l) \otimes \phi_{\mathcal{A}}(\tilde{a}_l)$

To simulate misspecification, we first train DRKPV (or DRPMMR) in the synthetic low-dimensional setting and then perturb either the outcome (KPV/PMMR) or treatment (KAP) bridge coefficients by injecting Gaussian noise.

Figures (5a) and (5b) display DRKPV results, averaged over five independent runs with standard deviation bands. In Figure (5a), outcome bridge coefficients are perturbed via $\alpha_{ij} \leftarrow \alpha_{ij} + \varepsilon_{ij}$, where $\varepsilon_{ij} \sim \mathcal{N}(0, 0.5)$. Similarly, in Figure (5b), treatment bridge coefficients are jittered as $\gamma_{ij} \leftarrow \gamma_{ij} + \varepsilon_{ij}$ with $\varepsilon_{ij} \sim \mathcal{N}(0, 0.5)$. Despite the misspecification, DRKPV successfully recovers the true causal function, with the slack term compensating for the introduced error.

Figures (5c) and (5d) present analogous results for DRPMMR, where Gaussian noise $\mathcal{N}(0, 0.5)$ is added to one of the learned bridge functions. As with DRKPV, DRPMMR remains robust, accurately recovering the true causal effect even under significant misspecification.

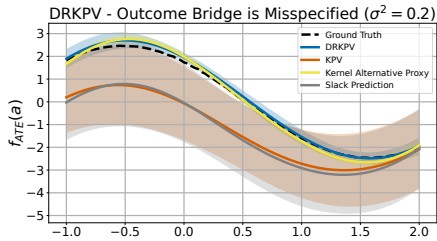

(a) DRKPV, Outcome Bridge Misspecified

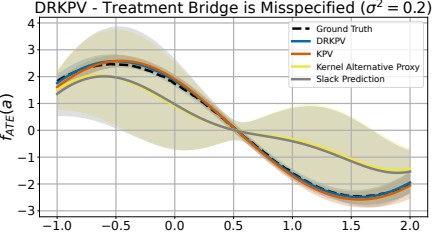

(b) DRKPV, Treatment Bridge Misspecified

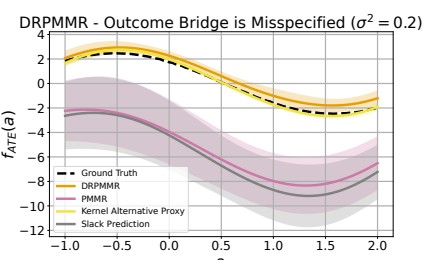

(c) DRPMMR, Outcome Bridge Misspecified

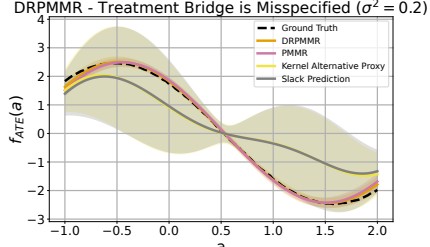

(d) DRPMMR, Treatment Bridge Misspecified

Figure 4: (Duplicate of Figure (3)) Experimental results in bridge function misspecifications with the synthetic low-dimensional data with jittering noise sampled from $\mathcal{N}(0, 0.2)$: (a) DRKPV estimates under outcome bridge misspecification; (b) DRKPV estimates under treatment bridge misspecification; (c) DRPMMR estimates under outcome bridge misspecification; and (d) DRPMMR estimates under treatment bridge misspecification.

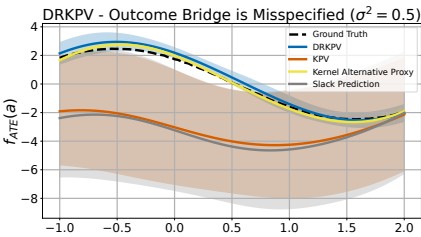

(a) DRKPV, Outcome Bridge Misspecified

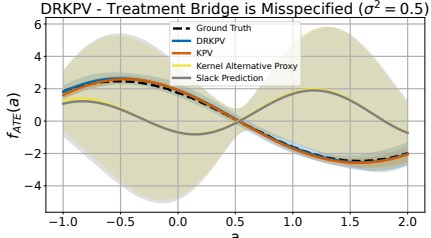

(b) DRKPV, Treatment Bridge Misspecified

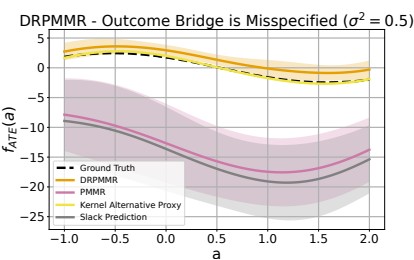

(c) DRPMMR, Outcome Bridge Misspecified

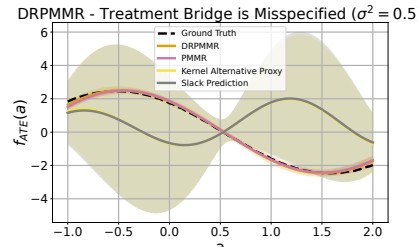

(d) DRPMMR, Treatment Bridge Misspecified

Figure 5: Experimental results under bridge function misspecifications with the synthetic low dimensional data with jittering noise sampled from $\mathcal{N}(0, 0.5)$: (a) DRKPV estimation when the outcome bridge is misspecified, (b) DRKPV estimation when the treatment bridge function is misspecified, (c) DRPMMR estimation when the outcome bridge function is misspecified, (d) DRPMMR estimation when the treatment bridge is misspecified.

## F.5 Numerical experiments with Job Corps dataset

In this section, we present numerical experiments with the Job Corps dataset [54, 55], which we accessed through the public repository provided by Singh et al. [69] (`https://github.com/liyuan9988/KernelCausalFunction/tree/master`). The U.S. Job Corps Program is an educational initiative designed to support disadvantaged youth. In the corresponding observational dataset, the treatment variable $A$ captures the total number of hours participants spent in academic or vocational training, whereas the outcome variable $Y$ measures the proportion of weeks the participant was employed during the program's second year. The covariate vector $U \in \mathbb{R}^{65}$ includes demographic and socioeconomic features such as gender, ethnicity, age, language proficiency, education level, marital status, household size, among others.

To adapt this dataset to the proximal causal learning (PCL) framework, Bozkurt et al. [23] proposed synthetic proxy generation schemes specifically crafted to test the limits of the completeness conditions stated in Assumptions (2.3) and (2.5). We adopt the six experimental settings introduced in their work to evaluate the performance of our proposed doubly robust estimators.

**Setting 1:** Let $W = U + \epsilon_w$ and $Z = g\left(U^{(1:20)}/\max\left(U^{(1:20)}\right)\right) + \epsilon_z$ where $g(x) = 0.8\frac{\exp(x)}{1+\exp(x)} + 0.1$ is applied elementwise, and both the division and maximum operations are performed elementwise. Here, the notation $U^{(1:20)}$ refers to the first 20 components of the vector $U$. Noise terms are sampled as $\epsilon_w^{(i)} \sim \mathcal{N}(0, 1)\ \forall i = 1, \ldots, 65, \epsilon_z^{(i)} \sim \mathcal{U}[-1, 1]\ \forall i = 1, \ldots, 20$.

**Setting 2:** Let $Z = U + \epsilon_z$ and $W = g\left(U^{(1:20)}/\max\left(U^{(1:20)}\right)\right) + \epsilon_w$ with $\epsilon_z^{(i)} \sim \mathcal{N}(0, 1)\ \forall i$, $\epsilon_w^{(i)} \sim \mathcal{U}[-1, 1]\ \forall i$.

**Setting 3:** Let $W = U + \epsilon_w$ and $Z = g\left(U^{(20:40)}/\max\left(U^{(20:40)}\right)\right) + \epsilon_z$ with $\epsilon_w^{(i)} \sim \mathcal{N}(0, 1)\ \forall i$, $\epsilon_z^{(i)} \sim \mathcal{U}[-1, 1]\ \forall i$.

**Setting 4:** Let $Z = U + \epsilon_z$ and $W = g\left(U^{(20:40)}/\max\left(U^{(20:40)}\right)\right) + \epsilon_w$ with $\epsilon_z^{(i)} \sim \mathcal{N}(0,1)\ \forall i$, $\epsilon_w^{(i)} \sim \mathcal{U}[-1,1]\ \forall i$.

**Setting 5:** Let $W = U + \epsilon_w$ and $Z = g\left(U^{(40:60)}/\max\left(U^{(40:60)}\right)\right) + \epsilon_z$ with $\epsilon_w^{(i)} \sim \mathcal{N}(0,1)\ \forall i$, $\epsilon_z^{(i)} \sim \mathcal{U}[-1,1]\ \forall i$.

**Setting 6:** Let $Z = U + \epsilon_z$ and $W = g\left(U^{(40:60)}/\max\left(U^{(40:60)}\right)\right) + \epsilon_w$ with $\epsilon_z^{(i)} \sim \mathcal{N}(0,1)\ \forall i$, $\epsilon_w^{(i)} \sim \mathcal{U}[-1,1]\ \forall i$.

Notice that the odd numbered settings are constructed to introduce an incomplete link between the treatment proxy $Z$ and the confounder $U$, combined with a nonlinear transformation. These configurations are prone to violating Assumption (2.3) that hinges the identifiability of dose-response with outcome bridge function (see Theorem (2.4)). On the other hand, even numbered settings are constructed to introduce an incomplete link between the outcome proxy $W$ and the confounder $U$, combined with a nonlinear transformation. These setups are likely to violate Assumption (2.5) , which underpins the identifiability guarantees of dose response with treatment bridge function (see Theorem (2.6)).

Bozkurt et al. [23] showed that their treatment bridge-based method, KAP, produces estimates that are more closely aligned with those of the oracle method *Kernel-ATE* [69], which assumes access to the true confounder $U$, compared to outcome bridge-based approaches such as KPV and KNC. They attribute this to KAP's robustness when the existence of the bridge function is violated, rather than when the assumption underlying the identifiability of the causal function is compromised. Conversely, their results suggest that KPV and KNC tend to yield estimates closer to the oracle method in scenarios where the existence of the bridge function is violated but the assumption ensuring the causal function's identifiability remains potentially valid—highlighting their strength under different failure modes. While these findings emphasize the complementary robustness of the two classes of methods under varying conditions, determining which condition holds in practice is often difficult. In this work, we demonstrate that our proposed doubly robust estimators effectively unify the strengths of both approaches, yielding consistently strong performance across diverse scenarios.

Figure (6) presents the estimated dose-response curves produced by our proposed methods across all six experimental settings averaged over 5 different realizations with standard deviation envelopes. For comparison, we include estimates from methods based solely on outcome or treatment bridge functions, along with the oracle method *Kernel-ATE* from Singh et al. [69], which has access to the true confounder $U$. For numerical comparison, Table (1) reports the mean squared distance between each algorithm's estimate and that of the oracle method. Across all settings, DRKPV consistently outperforms its non-doubly robust counterpart KPV, as well as the baselines KNC and KAP. In Settings 1, 3, 4, and 5, DRPMMR also outperforms its non-doubly robust variant PMMR and the other baselines. The only exceptions are Settings 2 and 6, where PMMR performs better than DRPMMR, based on the oracle method's dose-response estimates as the ground truth.

Table 1: Mean squared distance between each algorithm's estimated dose-response curve and that of the oracle method across the six experimental settings for the Job Corps dataset described in Section (S.M. F.5).

|  | DRKPV | DRPMMR | KPV | PMMR | KNC | KAP |
|---|---|---|---|---|---|---|
| Set. 1 | $\mathbf{0.87 \pm 0.30}$ | $1.17 \pm 0.49$ | $15.58 \pm 2.38$ | $1.55 \pm 0.44$ | $4.45 \pm 1.37$ | $1.95 \pm 0.21$ |
| Set. 2 | $1.78 \pm 0.83$ | $2.02 \pm 0.50$ | $4.30 \pm 2.05$ | $\mathbf{0.84 \pm 0.21}$ | $2.74 \pm 1.12$ | $9.67 \pm 2.95$ |
| Set. 3 | $2.58 \pm 0.65$ | $\mathbf{1.17 \pm 0.20}$ | $8.92 \pm 3.33$ | $1.63 \pm 0.35$ | $6.49 \pm 1.59$ | $3.15 \pm 0.96$ |
| Set. 4 | $2.29 \pm 0.43$ | $\mathbf{1.87 \pm 0.31}$ | $5.25 \pm 4.01$ | $1.96 \pm 0.14$ | $2.86 \pm 2.11$ | $4.12 \pm 1.21$ |
| Set. 5 | $1.07 \pm 0.43$ | $\mathbf{0.89 \pm 0.34}$ | $10.20 \pm 5.32$ | $1.26 \pm 0.30$ | $4.45 \pm 2.29$ | $1.83 \pm 0.71$ |
| Set. 6 | $0.99 \pm 0.42$ | $1.03 \pm 0.13$ | $4.99 \pm 2.59$ | $\mathbf{0.62 \pm 0.19}$ | $2.87 \pm 1.02$ | $2.70 \pm 1.48$ |

### F.6 Ablation study: hyperparameter sensitivity on the length scale

Here, we investigate the sensitivity of our methods to the selection of the kernel length-scale hyperparameter, $l$, for the Gaussian kernel used throughout our experiments. In the main paper (and detailed in Section (S.M. F.1)), we employed the median heuristic to set the length scale. This process

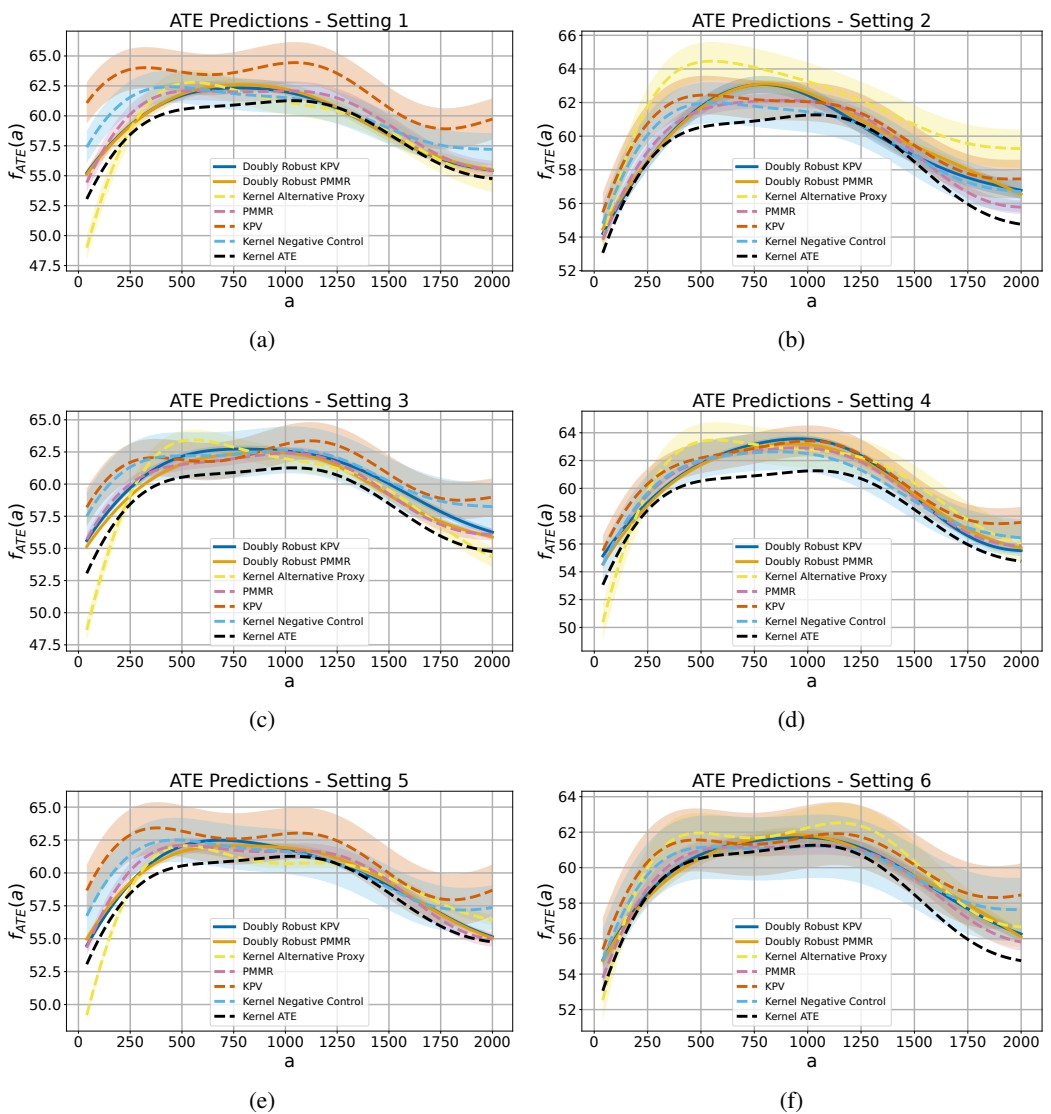

Figure 6: Dose-response estimation curves for the Job Corps experimental settings described in Section (S.M. F.5). Figures (6a)-(6f) display the dose-response estimates from our proposed methods, DRKPV and DRPMMR, compared against KPV, PMMR, KNC, KAP, and the oracle method, Kernel-ATE.

involves setting $l$ equal to the $0.5$-quantile of the computed pairwise distances across all training data. To demonstrate the robustness of our proposed DRKPV and DRPMMR methods, we vary the quantile value used for setting $l$ for each of the involved kernels. Specifically, we test the algorithms on the synthetic low-dimensional setting (with $t = 2000$) and the Legalized Abortion & Crime dataset by varying the quantile within the set $\{0.25, 0.4, 0.5, 0.6, 0.75, 0.9\}$. The corresponding results are presented in Table (2). The table demonstrates that the performance of both DRKPV and DRPMMR is stable across a broad range of length-scale quantile selections, and confirms that the $0.50$-quantile (median heuristic) is a robust choice. Degradation in performance is only significant at the extreme high end of the scale (e.g., the $0.90$-quantile in the synthetic dataset), indicating that our methods are generally robust to the specific selection of $l$.

Table 2: Ablation study on hyperparameter sensitivity: Mean Squared Error (MSE) across different length-scale ($\ell$) selections for the RBF kernel. The length scale is set to the specified quantile of pairwise distances.

| Algorithm | Quantile | MSE (mean $\pm$ std) | |
| --- | --- | --- | --- |
| | | Synthetic Low-Dim | Abortion & Crime |
| DRKPV | 0.25 | $0.055 \pm 0.035$ | $0.024 \pm 0.007$ |
| | 0.40 | $0.028 \pm 0.016$ | $0.020 \pm 0.014$ |
| | **0.50 (Median)** | $\mathbf{0.026 \pm 0.015}$ | $\mathbf{0.018 \pm 0.010}$ |
| | 0.60 | $0.027 \pm 0.018$ | $0.019 \pm 0.012$ |
| | 0.75 | $0.034 \pm 0.019$ | $0.023 \pm 0.016$ |
| | 0.90 | $0.22 \pm 0.096$ | $0.021 \pm 0.018$ |
| DRPMMR | 0.25 | $0.026 \pm 0.011$ | $0.016 \pm 0.005$ |
| | 0.40 | $0.020 \pm 0.011$ | $\mathbf{0.015 \pm 0.008}$ |
| | **0.50 (Median)** | $\mathbf{0.019 \pm 0.012}$ | $0.016 \pm 0.009$ |
| | 0.60 | $0.024 \pm 0.018$ | $0.018 \pm 0.009$ |
| | 0.75 | $0.034 \pm 0.033$ | $0.025 \pm 0.013$ |
| | 0.90 | $0.22 \pm 0.130$ | $0.022 \pm 0.015$ |

## F.7 Ablation study: performance with the Matérn kernel class

We now demonstrate the performance of our proposed methods across different kernel choices by conducting an experiment with the Matérn kernel class in the synthetic low-dimensional setting (with $t = 2000$) and the Legalized Abortion & Crime dataset.

As detailed in Section (S.M. F.1), the Matérn kernel is controlled by the smoothness parameter $\nu$. In this study, we focus on cases where $\nu$ is a half-integer, i.e., $\nu = p + 1/2$ where $p$ is an integer. The closed form polynomial expression is given in Equation (37).

We report the performance of our proposed methods, DRKPV and DRPMMR, for varying values of the integer $p$ within the set $\{0, 1, 2, 3, 10\}$. This range allows us to explore kernel smoothness from the least smooth Exponential kernel ($\nu = 1/2$, $p = 0$) up to an approximation of the highly smooth Gaussian kernel ($\nu = 10.5$, $p = 10$).

Table 3: Ablation Study: Mean Squared Error (MSE) for various Matérn kernel selections, controlled by the smoothness parameter $\alpha = p + 1/2$.

| Algorithm | Smoothness Parameter ($p$) | MSE (mean $\pm$ std) | |
| --- | --- | --- | --- |
| | | Synthetic Low-Dim | Abortion & Crime |
| DRKPV | 0 ($\nu = 0.5$) | $0.032 \pm 0.016$ | $\mathbf{0.019 \pm 0.007}$ |
| | 1 ($\nu = 1.5$) | $\mathbf{0.022 \pm 0.013}$ | $0.022 \pm 0.015$ |
| | 2 ($\nu = 2.5$) | $\mathbf{0.022 \pm 0.013}$ | $0.020 \pm 0.012$ |
| | 3 ($\nu = 3.5$) | $\mathbf{0.022 \pm 0.013}$ | $0.023 \pm 0.017$ |
| | 10 ($\nu = 10.5$) | $0.024 \pm 0.015$ | $0.023 \pm 0.014$ |
| DRPMMR | 0 ($\nu = 0.5$) | $0.029 \pm 0.011$ | $\mathbf{0.016 \pm 0.003}$ |
| | 1 ($\nu = 1.5$) | $0.019 \pm 0.013$ | $0.019 \pm 0.009$ |
| | 2 ($\nu = 2.5$) | $0.019 \pm 0.013$ | $0.018 \pm 0.010$ |
| | 3 ($\nu = 3.5$) | $\mathbf{0.018 \pm 0.013}$ | $0.018 \pm 0.010$ |
| | 10 ($\nu = 10.5$) | $0.022 \pm 0.016$ | $0.021 \pm 0.011$ |

## F.8 Scalability analysis: impact of Nyström approximation on performance

We now analyze the computational complexity of our proposed DRKPV method and introduce the Nyström approximation as a technique to enhance its scalability for large datasets. While we focus on DRKPV for simplicity, this discussion extends directly to DRPMMR.

The DRKPV framework comprises three primary computational components: 1) the KPV algorithm (Algorithm (2)), 2) the KAP algorithm (Algorithm (4)), and 3) the estimation of the slack correction term (step 3 in Algorithm (1)). We break down the complexity analysis as follows:

- **KPV Algorithm (Stage 1):** Utilizes $n_h$ data samples and involves the inversion of an $n_h \times n_h$ Gram matrix, yielding a complexity dominated by the matrix inversion, $O(n_h^3)$.
- **KPV Algorithm (Stage 2):** Uses $m_h$ data samples and requires the inversion of $m_h \times m_h$ Gram matrix. Its complexity is therefore $O(m_h^3)$.
- **KAP Algorithm (Stage 1):** Uses $n_\varphi$ data samples and involves the inverting an $n_\varphi \times n_\varphi$ Gram matrix, with a complexity of $O(n_\varphi^3)$.
- **KAP Algorithm (Stage 2):** The complexity is dominated by the inversion of $(m_\varphi + 1) \times (m_\varphi + 1)$ matrix, scaling with $O(m_\varphi^3)$, where $m_\varphi$ is the number of samples in the second stage.
- **KAP Algorithm (Stage 3):** Leverages $t_\varphi$ data samples, and its complexity is governed by inversion of $t_\varphi \times t_\varphi$ Gram matrix, scaling with $O(t_\varphi^3)$.
- **DRKPV - Slack correction term:** As outlined in Equation (6), this step requires inverting a $t \times t$ Gram matrix, where $t$ is the total number of data samples. Hence, the computational complexity is $O(t^3)$.

In conclusion, the overall computational complexity of our method, DRKPV, is dominated by the inversion of the largest matrix, scaling as $O(t^3)$. This is similar to the complexity of standard kernel methods like kernel ridge regression.

In general, the $O(t^3)$ computational complexity is a significant bottleneck for applications involving large datasets. To address this, we propose the Nyström approximation as a concrete step towards making our method more scalable.

**Nyström approximation:** Recall from Equations (38)-(39), the kernel ridge regression (KRR) estimator for the conditional mean function $f_0 = \mathbb{E}[Y \mid X = x]$ uses observational data $\{x_i, y_i\}_{i=1}^t$ and is given by:

$$\hat{f}(x) = \boldsymbol{Y}^\top (\boldsymbol{K}_{XX} + t\lambda\boldsymbol{I})^{-1} \boldsymbol{K}_{Xx}.$$

As detailed in the complexity breakdown above, the required matrix inversion of the $t \times t$ matrix $(\boldsymbol{K}_{XX} + t\lambda\boldsymbol{I})$ makes this solution expensive to compute, as the complexity scales cubically with the number of data points. The Nyström approximation technique [73] tackles this bottleneck by relying on a low-rank approximation to the Gram matrix $\boldsymbol{K}_{XX}$. This approximation is constructed by using a smaller subset of landmark points $\{s_i\}_{i=1}^p \subset \{x_i\}_{i=1}^t$, where $p \ll t$. These landmarks are typically selected using various sampling schemes [74]. The closed form expression with this approximation is given by (See Eqatuion (2) in [75]):

$$\hat{f}(x) = \boldsymbol{Y}^\top \boldsymbol{K}_{XS} \left( \boldsymbol{K}_{XS}^\top \boldsymbol{K}_{XS} + t\lambda\boldsymbol{K}_{SS} \right)^{-1} \boldsymbol{K}_{Sx} \tag{42}$$

where $[\boldsymbol{K}_{XS}]_{i,j} = k_\mathcal{X}(x_i, s_j)$, $[\boldsymbol{K}_{SS}]_{ij} = k_\mathcal{X}(s_i, s_j)$, and $[\boldsymbol{K}_{Sx}]_i = k_\mathcal{X}(s_i, x)$. The primary computational advantage of this Nyström form is that the matrix requiring inversion, $\left( \boldsymbol{K}_{XS}^\top \boldsymbol{K}_{XS} + t\lambda\boldsymbol{K}_{SS} \right)^{-1}$, is now only $p \times p$. Assuming that the number of landmarks $p$ is substantially smaller than the total number of samples ($p \ll t$), the computational cost of kernel ridge regression solution is reduced from $O(t^3)$ to $O(p^2 t)$.

We apply this low-rank approximation to the computationally intensive kernel regression stages in our DRKPV algorithm:

- KPV (Stage 1): Kernel ridge regression from the input features $\{\phi_\mathcal{Z}(\bar{z}_i \otimes \phi_\mathcal{A}(\bar{a}_i))\}_{i=1}^{n_h}$ to the observations $\{\phi_\mathcal{W}(\bar{w}_i)\}_{i=1}^{n_h}$.
- KPV (Stage 2): Kernel ridge regression from the input features $\{\hat{\mu}_{W|Z,A}(\tilde{z}_i, \tilde{a}_i) \otimes \phi_\mathcal{A}(\tilde{a}_i)\}_{i=1}^{m_h}$ to the observations $\{\tilde{y}_i\}_{i=1}^{m_h}$.
- KAP (Stage 1): Kernel ridge regression from the input features $\{\phi_\mathcal{W}(\bar{w}_i \otimes \phi_\mathcal{A}(\bar{a}_i))\}_{i=1}^{n_\varphi}$ to the observations $\{\phi_\mathcal{Z}(\bar{z}_i)\}_{i=1}^{n_\varphi}$.
- KAP (Stage 3): Kernel ridge regression from the input features $\{\phi_\mathcal{A}(\dot{a}_i)\}_{i=1}^{t_\varphi}$ to the observations $\{\dot{y}_i \phi_\mathcal{Z}(\dot{z}_i)\}_{i=1}^{t_\varphi}$.
- Doubly robust correction term (Equation 29): Kernel ridge regression from the input features $\{\phi_\mathcal{A}(a)_i\}_{i=1}^t$ to the observations $\{\phi_\mathcal{Z}(z_i) \otimes \phi_\mathcal{W}(w_i)\}_{i=1}^t$.

The second-stage regression of KAP involves a more specialized setup that deviates from standard KRR, but it is equally amenable to approximation techniques. Table 4 presents synthetic low-dimensional experiments comparing the full-kernel DRKPV to its Nyström-approximated version ($p = 500$ and $p = 1000$), where all regularization parameters are uniformly set to $10^{-3}$. Results clearly demonstrate that the Nyström approximation effectively reduces the algorithm's runtime while maintaining or even improving prediction accuracy. Future work will also explore stronger scalable alternatives such as scalable kernel ridge regression [75] and the neural network approaches [13, 35], including for the second-stage regression of KAP.

Table 4: Performance and scalability comparison of DRKPV with Nyström approximation in synthetic low-dimensional experiments.

| Algorithm | Data Size ($t$) | MSE (mean $\pm$ std) | Algo Run Time (s) |
|---|---|---|---|
| DRKPV (Original) | 5000 | $0.0097 \pm 0.0046$ | $61.1 \pm 0.21$ |
| DRKPV (Nyström, $p = 500$) | 5000 | $0.0082 \pm 0.0033$ | $42.5 \pm 0.19$ |
| DRKPV (Nyström, $p = 100$) | 5000 | $\mathbf{0.0071 \pm 0.0028}$ | $\mathbf{39.2 \pm 0.22}$ |
| DRKPV (Original) | 7500 | $\mathbf{0.0070 \pm 0.0033}$ | $194.5 \pm 0.35$ |
| DRKPV (Nyström, $p = 500$) | 7500 | $0.0085 \pm 0.0044$ | $132.7 \pm 0.04$ |
| DRKPV (Nyström, $p = 100$) | 7500 | $0.0091 \pm 0.0046$ | $\mathbf{127.6 \pm 0.00}$ |
| DRKPV (Original) | 10000 | $0.0065 \pm 0.0026$ | $453.9 \pm 0.51$ |
| DRKPV (Nyström, $p = 500$) | 10000 | $\mathbf{0.0059 \pm 0.0014}$ | $306.7 \pm 0.02$ |
| DRKPV (Nyström, $p = 100$) | 10000 | $0.0062 \pm 0.0022$ | $\mathbf{299.5 \pm 0.00}$ |

