# OpenReview forum: "Density Ratio-Free Doubly Robust Proxy Causal Learning"
_NeurIPS.cc/2025/Conference — NeurIPS 2025 poster_

### Official Review · Reviewer_oWY2 · 2025-06-14

**Clarity:** 4
**Significance:** 2
**Originality:** 2
**Rating:** 4
**Confidence:** 3

**Summary:**

This paper proposes two kernel-based doubly robust (DR) estimators that leverage both outcome and treatment bridge functions, enabling them to handle continuous and high-dimensional covariates. Following a framework similar to Cui et al. [17], the estimators integrate the KPV/PMMR method [15] for estimating $h$ and KAP [19] for estimating $\varphi$, thereby constructing the DR estimator. The authors establish consistency for the proposed estimator and support their approach with extensive simulation studies.

**Questions:**

1. **Figure 1**

The arrow from U to W should be solid, and the arrow from Y to W should be dashed.
Consider an example where $U \sim N(0,1)$, $W \sim N(0,1)$, $A \sim N(U,1)$, and $Y \sim N(A+W,1)$.
This example is compatible with Figure 1, but we find $E[U|W,A=0] = E[U|A=0]=0$, indicating that $\ell(U) = U$ serves a counterexample of Assumption 2.5.
Thus, if $U$ and $W$ are not associated, Assumption 2.5 cannot be satisfied, which requires a solid arrow between $U$ and $W$.
For further justification, see Table A.1 in Tchetgen Tchetgen et al. (2024, Statistical Science), which also specifies the requirement for a direct connection between $U$ and $W$.


2. **Theorem**

What notion of convergence is used in the theorems?

The paper includes high-dimensional simulations, but it’s unclear whether Theorems 4.1 and 4.2 are valid in such settings. How is the curse of dimensionality addressed? Please clarify the assumptions under which these theoretical results extend to high-dimensional cases.


3. **Reference**

In explaining Equation (2), it would be helpful to cite foundational work on stabilized weights for continuous treatments (e.g., Robins et al., 2000, Epidemiology). This would situate your methods more clearly in the literature and justify the weighting strategy.

4. **Contribution**

The fundamental concept of integrating KPV/PMMR with KAP is sound. Nevertheless, the manuscript fails to clearly delineate whether this integration yields significant theoretical or methodological innovations. Notably, [19] also asserts applicability to continuous treatment and high-dimensional settings, suggesting that these aspects are not exclusive contributions of the current paper. Consequently, the paper's contribution, as articulated, appears to be primarily confined to the development of the DR property. Consequently, the paper's contribution, as articulated, appears to be primarily confined to the development of the DR property. The paper should place clearer emphasis on why this DR development is a nontrivial task.

5. **Typo**

Line 320, Page 8: $f_{ATE} \rightarrow \theta_{ATE}$

**Ethical Concerns:**

["NO or VERY MINOR ethics concerns only"]

**Final Justification:**

I appreciate the authors’ engaging discussion. The work makes a valuable contribution to the literature, and I have accordingly raised my evaluation score.

**Limitations:**

Yes

**Quality:**

4

**Strengths And Weaknesses:**

**Quality**

The submission demonstrates technical soundness, presenting well-substantiated claims grounded in rigorous theoretical analysis and simulation-based evidence.

**Clarity**

The paper is clearly written and well-organized.

**Significance**

The work addresses a topic of considerable interest, as many practical causal inference problems involve continuous treatments. The high-dimensional extension is also promising, though the theoretical justification in this context requires further clarification.

**Originality**

While the paper builds on ideas from [15], [17], and [19], the novelty of its contributions is not clearly delineated. The main methodological development appears between Line 250 (Page 6) and Line 270 (Page 7), but this section seems to offer only incremental innovation. Furthermore, some relevant literature is omitted, reducing the clarity of the paper's positioning relative to existing work.

---

> ### Author Rebuttal · Authors · 2025-07-31
>
> We are grateful for your careful and detailed review. In what follows, we address your concerns point by point. If any issues remain open, we will gladly provide further clarification. If our responses satisfactorily resolve your concerns, we would be thankful if you would consider adjusting your score.
> > **Originality**: The main methodological development appears between Line 250 (Page 6) and Line 270 (Page 7), but this section seems to offer only incremental innovation.
> > **Questions**: 4-) The fundamental concept of integrating KPV/PMMR with KAP is sound. Nevertheless, the manuscript fails to clearly delineate whether this integration yields significant theoretical or methodological innovations. ... The paper should place clearer emphasis on why this DR development is a nontrivial task.
>
> Although our work builds on prior literature, our contributions extend substantially beyond the section you referenced, and we will revise the manuscript to make this clearer.
> *    **Novel doubly robust (DR) identification in PCL**: Theorem 2.7 provides the first DR identification result in the PCL setting that avoids indicator functions and kernel smoothing. Prior DR methods [17, 18] are restricted to discrete or univariate continuous treatments, whereas our approach naturally extends to high-dimensional and continuous treatments. We highlight this advancement in Remarks 2.8 and 2.9.
> *    **Density-ratio-free DR formulation**: While avoiding density ratio estimation was introduced in [19], our integration of treatment bridge–based methods into a DR framework introduces substantive methodological differences. For example, [17] employs correction terms of the form $\mathbb{E}[q(Z,A)h(W,A) I(A - a)]$, whereas our formulation uses $\mathbb{E}[\varphi(Z,A)h(W,A)| A = a],$ reflecting a distinct treatment bridge function $\varphi$. .This yields a structurally new identification formula and results in the **first density-ratio-free DR PCL estimators**. Importantly, our kernel-based construction provides easily implementable closed-form solutions, which are nontrivial to derive as they require careful algebra leveraging properties of reproducing kernel Hilbert spaces (Appendix C.3).
> *    **Strong uniform convergence guarantees**: Our results (Theorems 4.1 and 4.2; details in Appendix E) establish **uniform consistency**, which is **strictly stronger than the pointwise convergence** typically used in earlier works. Achieving such rates in a doubly robust setting is technically demanding, requiring precise error control across multiple regression stages, including the novel correction term estimation.
> *   **Practical impact**: These methodological contributions translate into tangible performance gains. As shown in Section 5, our DR estimators (DRKPV and DRPMMR) substantially outperform outcome-only and treatment-only bridge methods across diverse PCL benchmarks, including challenging high-dimensional setups such as dSprite and the semi-synthetic JobCorps dataset. Furthermore, our method outperforms the previous doubly robust approach PKDR [18], which relies on kernel smoothing.
>
> We will revise our manuscript to more clearly articulate our contributions, ensuring that the nontrivial nature of our DR development is evident. In particular, by utilizing the extra content page, some of the revisions we incorporate include:
> *    Abstract line 11 will be revised as: "By using kernel mean embeddings, we propose the first density ratio-free doubly robust estimators for proxy causal learning (PCL), which have closed-form solutions and strong consistency guarantees."
> *    Line 102 will be revised as: "We establish uniform consistency of our proposed estimators which is stronger than the pointwise convergence typical in doubly robust causal learning"
> *    Revision to Remark (2.9): "This structural difference enables applicability to high-dimensional treatments, where kernel-smoothing–based DR approach fails."
>
> Together, these points demonstrate that our work provides a substantial and nontrivial advancement of the PCL literature. For further revisions on consistency section, see our corresponding response below.
>
> > **Questions**: 1-) The arrow from U to W should be solid, and the arrow from Y to W should be dashed.
>
> We appreciate the opportunity to clarify our notation. While the counterexample you provided indeed violates Assumption (2.5), we kindly disagree that it aligns with the structure of Figure (1). In our notation, a dashed arrow indicates the possibility that either: (i) $U \rightarrow W$, (ii) $W \rightarrow U$, or (iii) $U$ and $W$ share a common ancestor. The example you provided implies no dependence between $U$ and $W$, which differs from our intended interpretation. We acknowledge that our explanation may not be sufficiently clear, and we will accordingly update our manuscript to address this. Specifically, we will add the clarification after line 28 in our updated manuscript.
>
> Furthermore, this convention is consistent with Table A.1 in Tchetgen Tchetgen et al. (2024, Statistical Science), which shows that both $U \rightarrow W$ and $W \rightarrow U$ are possible.
>
> > **Significance**: The high-dimensional extension is also promising, though the theoretical justification in this context requires further clarification.
> > **Questions**: 2-) What notion of convergence is used in the theorems? The paper includes high-dimensional simulations, but it’s unclear whether Theorems 4.1 and 4.2 are valid in such settings. How is the curse of dimensionality addressed? Please clarify the assumptions under which these theoretical results extend to high-dimensional cases.
>
> In Theorems 4.1 and 4.2, we establish **uniform consistency** of our estimators DRKPV and DRPMMR. In fact, we obtain high-probability finite-sample bounds on the error in the supremum norm (see Appendix E.3, Theorems E.27 and E.28). Specifically, under smoothness assumptions on the regression stages (e.g. Assumptions E.5, E.6, E.11, E.12, E.17), our estimators converge to the true causal function at the rates we derive in Appendix E.3. These results rely on conditional mean embedding and kernel ridge regression (KRR) learning rates (see, e.g., [37, 52, 53]), which provide rigorous convergence guarantees.
>
> We agree with the reviewer that the curse of dimensionality is a real concern. Kernel Ridge Regression (KRR) is known to achieve minimax-optimal rates in moderate dimension, but in very high-dimensional regimes, particularly when the input dimension $d$ grows with the sample size $n$, i.e. $d/n^\beta \rightarrow c$ for $\beta \in(0,1)$, the situation becomes more subtle [R1]. In particular:
>
> *    When functions lie in Sobolev classes, the achievable rate explicitly depends on both smoothness and dimension. Intuitively, the regression function must be “smooth enough” relative to the ambient dimension $d$ for KRR to remain consistent. More specifically, S. Fischer and I. Steinwart [51, Corollary 5 & 6] (for standard scalar KRR) and Li et al. [R2 Corollary 1 & 2] (for vector valued KRR) show that if the target function has smoothness $s$, then the optimal rate of convergence in $L_2$ norm is $O(n^{-\frac{s}{s + d/2}})$ and that this rate is achieved by KRR. This rate clearly shows the curse of dimensionality where for a fixed $s$, the bound becomes vacuous as $d \to +\infty$.
> *    Donhauser et al. [R1] show that for rotationally invariant kernels, a polynomial approximation barrier arises: the learned function is effectively restricted to low-degree polynomials as $d$ grows, regardless of eigenvalue decay. This implies that consistency in high dimensions is limited unless additional structural assumptions are imposed.
>
> In our work, we do not claim to resolve the curse of dimensionality. Instead, our contribution is to show that doubly robust PCL estimators can be constructed without density ratio estimation and kernel-smoothing, thereby extending applicability to continuous and high-dimensional treatments where prior DR methods [17, 18] fail. Our theorems remain valid in high-dimensional settings provided the smoothness and effective RKHS dimension assumptions hold, but we acknowledge that when $d$ increases with $n$, methods based on standard RBF or Matern kernels may indeed fail.
>
> We will clarify this point in the updated version by extending the discussion in our consistency section.
>
> References
>
> [R1] Konstantin Donhauser, Mingqi Wu, Fanny Yang, (2021). "How rotational invariance of common kernels prevents generalization in high dimensions". PMLR
>
> [R2] Zhu Li, Dimitri Meunier, Mattes Mollenhauer, and Arthur Gretton. (2024) "Towards optimal sobolev norm rates for the vector-valued regularized least-squares algorithm". JMLR
>
> > **Questions**: 3-) In explaining Equation (2), it would be helpful to cite foundational work on stabilized weights for continuous treatments (e.g., Robins et al., 2000, Epidemiology). This would situate your methods more clearly in the literature and justify the weighting strategy.
> > **Originality**: Furthermore, some relevant literature is omitted, reducing the clarity of the paper's positioning relative to existing work.
>
> We will incorporate the recommended literature in the updated manuscript. In particular, we will clarify the connection between the treatment bridge–based approach and inverse-probability-weighted causal learning methods. We would also be glad to consider any further references you believe would help strengthen the positioning of our work.
> > **Questions**: 5-) Typo in line 320.
>
> Thank you for pointing out this typo; we have corrected it.

---

> > ### Comment · Reviewer_oWY2 · 2025-08-03
> >
> > I appreciate the authors’ comprehensive response.
> > Their clarifications have helped me gain a better understanding of the paper and its contribution to the literature.
> >
> > However, I still have concerns regarding Figure 1.
> >
> > First, I recommend referring to Figure 1 as a causal graph rather than a DAG, since directed acyclic graphs do not permit bi-directed arrows.
> >
> > Second, the figure caption states that “dotted bidirectional arrows indicate potential bidirectional causality or the existence of a shared latent ancestor between variables,” which permits the possibility of no association. This differs from the authors’ interpretation of <-> as representing either (i) A -> B, (ii) A <- B, or (iii) A <- C -> B, for some unmeasured variable C. This distinction should be clarified to avoid potential confusion.
> >
> > Third, the edge Z <-> A implies Z -> A, Z <- A, or Z <- C -> A. However, Assumption 2.2 may still hold even when Z and A are conditionally independent given U.
> >
> > For these reasons, Figure 1 may not serve as the most compelling causal graph for the proximal causal learning framework, and parts of the caption may be misleading. In particular, the phrase “characterizing the structure assumed in the proximal causal learning” is potentially misleading. Rather than suggesting a unique or definitive structure, the figure should be presented as one illustrative example among many possible scenarios that are consistent with Assumption 2.2.
> >
> > Regarding the convergence, my original intention was to clarify whether it holds with high probability, in probability, or almost surely. It would be helpful if this distinction were explicitly stated in the theorem, ideally by incorporating the appropriate phrase into the theorem statement itself.
> >
> > Regarding the high-dimensional case, I believe an honest evaluation would be appreciated. For instance, while phrases such as “scales well to high-dimensional treatments” are not technically false, they may be misleading---potentially giving the impression that the proposed method fully achieves the theoretical guarantees in high-dimensional settings. This point warrants clarification, and it would be helpful to explicitly acknowledge this as a limitation.

---

> > > ### Author Response · Authors · 2025-08-05
> > >
> > > We thank the reviewer for these further clarifications and take this opportunity to address them.
> > >
> > > ## Figure 1 terminology and interpretation
> > >
> > > You are correct that our current description of Figure (1) may cause confusion. We will revise both the figure caption and the surrounding text:
> > >
> > > *    We will refer to Figure (1) as a **causal graph** rather than a DAG. Bidirected arrows will be clarified as shorthand for latent confounding or ambiguous directionality, not literal DAG edges.
> > > *    We will emphasize that Figure 1 is **illustrative rather than definitive**, and other causal structures may also satisfy Assumptions (2.2)–(2.5).
> > > *    The figure caption will be revised as: “An illustrative causal graph for proxy causal learning (PCL), consistent with Assumption (2.2) [5].”
> > > *    In line 28, we will add: “In this setup, a bidirectional arrow indicates that either causal direction between the two variables is plausible, or that they may share an unobserved common cause.”
> > >
> > > ## Convergence notion
> > >
> > > Thank you for pointing this out. As noted in our rebuttal, our results in Theorems (4.1) and (4.2) rely on finite-sample high-probability bounds, as detailed in Theorems (E.27) and (E.28). In the updated manuscript, we will explicitly state that convergence in these theorems holds with high probability. We will also extend the discussion in Section (4) by drawing directly from Appendix (E).
> > >
> > > ## High-dimensional setup
> > >
> > > We acknowledge that our methods may fail in very high-dimensional regimes, i.e., when the input dimension $d$ grows with the sample size $n$. We will explicitly state this limitation in Section (4) and (6), and expand on it in Appendix (E). At the same time, we note that our framework already accommodates substantially higher-dimensional treatments than prior doubly robust PCL methods [17,18], which are restricted to discrete or univariate continuous treatments.
> > >
> > > We will add the following remark in Section (4):
> > >
> > > **Remark**: We establish uniform consistency of DRKPV and DRPMMR, which ensures control of estimation error across the entire treatment domain. These results hold under smoothness and effective RKHS dimension assumptions. We note that while kernel ridge regression provides optimal rates in Sobolev–Matern classes [R2], the rates of convergence are slower for larger input dimensions [R1, R2]. Our results therefore apply in high-dimensional settings under dimension-dependent smoothness assumptions, but addressing increasing-dimension regimes is left for future work.
> > >
> > > We again thank the reviewer for these thoughtful comments and hope this response clarifies the concerns raised. If you feel your questions have been addressed, we would be grateful if you considered reflecting that in your final evaluation.
> > >
> > > References
> > >
> > > [R1] Konstantin Donhauser, Mingqi Wu, Fanny Yang, (2021). "How rotational invariance of common kernels prevents generalization in high dimensions". PMLR
> > >
> > > [R2] Zhu Li, Dimitri Meunier, Mattes Mollenhauer, and Arthur Gretton. (2024) "Towards optimal sobolev norm rates for the vector-valued regularized least-squares algorithm". JMLR

---

### Official Review · Reviewer_R2kv · 2025-06-27

**Clarity:** 3
**Significance:** 4
**Originality:** 3
**Rating:** 5
**Confidence:** 3

**Summary:**

The work  presents a way of estimating causal functions under the proxy causal learning (PCL) framework. They account for confounders by proxy of variables for treatment (Z) and outcome (W). The two estimators presented are both of the doubly  robust (DR) kernel-based type, and combine the benefits of  both Z and W bridge-based estimattors.
The density-ratio freedom implies they they circumvent the need of the indicator finction or kernel smoothing over treatment variables.The experiments show improved performance over sysnthetic high-dimensional data.

**Questions:**

Between Z/W, one may sometimes be information-poor, and the other the more informative proxy. How does that play into the calculus?

**Ethical Concerns:**

["NO or VERY MINOR ethics concerns only"]

**Final Justification:**

I cnceded to the arguments in the comments to my review.

**Limitations:**

I'm not able to determine those critically!

**Paper Formatting Concerns:**

-

**Quality:**

4

**Strengths And Weaknesses:**

Kernel mean embeddings help with the closed-form and strongly-consistent guarantees.

It appears that the work is an important step in measuring confounders.

Experiments are on high dim (quasi continuous)  data, but not on copious volumes.  Kernel computation may be a bottleneck in that.

---

> ### Author Rebuttal · Authors · 2025-07-31
>
> We sincerely appreciate your thoughtful and valuable feedback. Below, we respond to your comments point by point. If anything remains unclear, we would be happy to elaborate further. We hope our responses address your concerns and reinforce your positive evaluation of our work.
> > **Questions:** Between Z/W, one may sometimes be information-poor, and the other the more informative proxy. How does that play into the calculus?
>
> In Appendix F.4, we conduct simulations designed precisely for such scenarios.
>
> **Setup**: The JobCorps dataset (an observational study) is adapted to the PCL framework by generating synthetic proxies with varying informativeness. The treatment variable $A$ measures hours spent in training, while the outcome $Y$  is the proportion of weeks employed in the second year. Confounders $U\in \mathbb{R}^{65}$ include demographic and socioeconomic features. [19] proposed proxy-generation schemes to test the completeness assumptions (2.3 and 2.5), which we adopt to evaluate our doubly robust estimators. Based on [19],we expect the behavior:
> *    Outcome-bridge methods perform better when $W$ is less informative, since the identification via treatment-bridge is challenged (Assumption 2.5).
> *    Treatment-bridge methods perform better when $Z$ is less informative, since identification via outcome-bridge is challenged (Assumption 2.3).
>
> Across six proxy-informativeness settings, we compare DRKPV and DRPMMR with outcome-bridge and treatment-bridge baselines. Results (Table 1, Figure 5) in Appendix F.4 show that even when one class of methods underperforms, our doubly robust estimators consistently provide strong performance by effectively combining the strengths of both approaches.
>
> This robustness arises from the double robustness property: our methods remain valid as long as either the outcome bridge or the treatment bridge is well specified.
>
> > **Strengths And Weaknesses:** Experiments are on high dim (quasi continuous) data, but not on copious volumes. Kernel computation may be a bottleneck in that.
>
> **Complexity Analysis**: The main computational bottleneck is kernel matrix inversion. As detailed in our response to Reviewer tteA (see Complexity and Scalability discussion), our method scales cubically with sample size due to multi- and single-stage kernel regressions in KPV, KAP, and PMMR. Specifically, both of our methods have complexity $O(t^3)$, where $t$ is the number of observations. In Appendix F, we will expand this discussion and include complexity breakdowns.
>
> **Scalability**: To mitigate this, we conducted additional experiments using the Nyström approximation [R1]. We discuss this extensively in our response to Reviewer tteA, which provides detailed Nyström-based results. Briefly, we applied it across all DRKPV kernel regressions (KPV first/second, KAP first/third, slack prediction), showing substantial runtime reductions while maintaining accuracy. DRPMMR can be approximated similarly. We will include these ablation studies in Appendix F. For full results and explanations, please see our response to Reviewer tteA. In future work, we aim to apply more advanced techniques, including scalable kernel ridge regression [R2] and neural mean embedding framework [R3].
>
> References
>
> [R1] Williams, C. K. I. and M. Seeger (2001). “Using the Nystrom Method to Speed Up Kernel Machines”. In: NeurIPS 13.
>
> [R2] Meanti, G., L. Carratino, L. Rosasco, and A. Rudi (2020). “Kernel methods through the roof: handling billions of points efficiently”. In: NeurIPS 34.
>
> [R3] Liyuan Xu and Heishiro Kanagawa and Arthur Gretton, (2021). "Deep Proxy Causal Learning and its Application to Confounded Bandit Policy Evaluation". In NeurIPS 35.

---

### Official Review · Reviewer_j1Ba · 2025-07-02

**Clarity:** 3
**Significance:** 3
**Originality:** 3
**Rating:** 4
**Confidence:** 3

**Summary:**

This paper addresses causal function estimation in the Proxy Causal Learning (PCL) framework, where confounders are unobserved but proxy variables are available. The authors propose two novel kernel-based doubly robust estimators that integrate outcome and treatment bridge methods. These estimators avoid kernel smoothing over the treatment and density ratio estimation, making them especially effective for continuous or high-dimensional treatments. Using kernel mean embeddings, the method achieves closed-form solutions and strong consistency. Empirical results on PCL benchmarks show superior performance over existing methods, including previous doubly robust approaches.

**Questions:**

1.	In Supp lines 939-945, the author mention that the method of utilizing IF is locally effective. In Theorem 2.7, the author derives an identification formula that inherit the dr property. Will the proposed estimators also inherit local efficiency, or at least satisfy asymptotic normality? Given the nonregularity of continuous treatment estimators, should we expect slower-than-$\sqrt{n}$ asymptotic normal.
2.	Given that kernel-based methods are involved in parts of the estimation, can the authors clarify the computational efficiency of the proposed approach?

**Ethical Concerns:**

["NO or VERY MINOR ethics concerns only"]

**Final Justification:**

Most of my concerns have been addressed by the author's rebuttal. This is a nice work, and I would like to see it in the conference.

**Limitations:**

The work does not have potential negative societal issues.

**Paper Formatting Concerns:**

No.

**Quality:**

3

**Strengths And Weaknesses:**

Strengths:
1.	The proposed estimators avoid kernel smoothing over the treatment or density ratio estimation, making them particularly suitable for continuous or high-dimensional treatment settings.
2.	Both estimators are shown to possess the doubly robust (DR) property, which enhances their robustness to model misspecification.
3.	The paper provides uniform consistency guarantees and demonstrates empirical performance, offering a compelling combination of theoretical justification and practical utility.

Weaknesses:
1.	The idea lacks novelty. The core idea of combining estimators from [10] and [19] to construct a new DR estimator is similar to the classical causal inference literature. Notably, the claimed benefits—such as avoiding kernel smoothing and handling high-dimensional treatments—are inherited from [19].
2.	Both [10] and [19] derive convergence rates for their respective estimators. Does the proposed DR estimator—formed via their combination—achieve a similar rate? If so, is it the minimum of the two, or can it attain a faster rate under certain conditions?
3.	One key advantage of DR estimators is inference. However, the paper does not analyze the asymptotic distribution of the proposed estimator, and thus it is unclear whether valid confidence intervals can be constructed.
4.	In Lines 276–278, the authors mention that different stages require different samples, obtained via either data splitting or reuse. If data splitting is applied, sample inefficiency might be a concern. If reuse is employed, what are the implications for theoretical guarantees (e.g., bias or overfitting)? This trade-off requires further discussion.

---

> ### Author Rebuttal · Authors · 2025-07-31
>
> We greatly appreciate your valuable and detailed feedback. Below, we respond to your questions and concerns point by point. Should any points remain unclear, we are glad to provide additional explanation. If our clarifications have resolved your concerns, we would be grateful if you would consider increasing your score.
> > **Weaknesses**: ... the claimed benefits—such as avoiding kernel smoothing and handling high-dimensional treatments—are inherited from [19].
>
> While we acknowledge that advantages such as avoiding kernel smoothing and accommodating high-dimensional treatments are shared across kernel-based PCL frameworks [19, 10, 11], these stem from conditional mean embeddings (CMEs) in reproducing kernel Hilbert spaces (RKHSs).
>
> Our contribution goes substantially beyond this: we introduce the **first doubly robust, density ratio-free PCL framework** using kernel machines. Specifically:
> * **Novelty of our approach**: Unlike [19], which estimates the causal function solely via integration of the treatment bridge function, our framework leverages the double robustness property. This formulation avoids density ratio estimation and ensures **robustness to misspecification of either bridge function**, a point substantiated by our empirical results.
> * **Handling high-dimensional treatments**: Because our identification relies on conditional means rather than joint expectations with indicator functions or kernel smoothing, our methods naturally extend to higher-dimensional treatments. Other kernel-based frameworks [10, 11, 19] share this advantage, but unlike ours, they are not doubly robust.
> * **Empirical validation**: Our experiments show that DRKPV and DRPMMR consistently outperform existing kernel-based methods including KAP [19], KPV/PMMR [10] across diverse PCL benchmarks including high dimensional treatment settings. Furthermore, doubly robust method PKDR [18], which incorporates kernel smoothing, fails to operate in high-dimensional treatment settings since its implementation is limited to univariate treatments. This limitation reflects the instability of ratio estimation and poor approximation of indicator functions in higher dimensions.  In contrast, our methods successfully estimate causal functions in challenging settings such as the dSprite dataset, which features high-dimensional ($64 \times 64$ image) treatments.
> > **Weaknesses**: Both [10] and [19] derive convergence rates for their respective estimators. Does the proposed DR estimator—formed via their combination—achieve a similar rate? If so, is it the minimum of the two, or can it attain a faster rate under certain conditions?
>
> Our method achieves strong uniform consistency and, under some smoothness conditions, can converge faster than both KAP and KPV individually. In Appendix F we derive rates for both DRKPV and DRPMMR. For instance, under condition (iv) of Theorem E.27, DRKPV converges with $$O_p(t^{-\frac{1}{2}\frac{\beta_{\varphi, 3}-1}{\beta_{\varphi, 3} + p_{\varphi, 3}}} + m_h^{-\frac{1}{2} \frac{\beta_{h,2} - 1}{\beta_{h,2} + p_{h,2}}} m_{\varphi}^{-\frac{1}{2}\frac{\beta_{\varphi, 2}-1}{\beta_{\varphi, 2} + p_{\varphi, 2}}}).$$
> Here, $\beta_{\cdot}$ denote smoothness parameters and $p_{\cdot}$ the effective RKHS dimensions in the corresponding regression stages. By comparison, KAP achieves $O_p(t^{-\frac{1}{2}\frac{\beta_{\varphi, 3}-1}{\beta_{\varphi, 3} + p_{\varphi, 3}}} + m_{\varphi}^{-\frac{1}{2}\frac{\beta_{\varphi, 2}-1}{\beta_{\varphi, 2} + p_{\varphi, 2}}})$ (see Theorem 5.4 in [19]), while KPV achieves $O_p(m_h^{-\frac{1}{2} \frac{\beta_{h,2} - 1}{\beta_{h,2} + p_{h,2}}})$ (see Proposition (1) in [10]). This indicates that our method converges faster than the KAP algorithm. Furthermore, our method converges faster than KPV if $t^{-\frac{1}{2}\frac{\beta_{\varphi, 3}-1}{\beta_{\varphi, 3} + p_{\varphi, 3}}} < m_h^{-\frac{1}{2} \frac{\beta_{h,2} - 1}{\beta_{h,2} + p_{h,2}}}$. The best achievable rate in this setting is $O(t^{-1/4})$, a typical nonparametric rate. This arises because our identification is based on a conditional expectation involving the treatment bridge and outcome bridge functions, rather than the expectation of a non-centered influence function. While the latter may theoretically yield faster rates, in practice such approaches suffer from density ratio estimation and kernel smoothing.  The advantages of our approach are twofold:
> * **Uniform convergence**: Unlike most doubly robust frameworks, which prove only pointwise consistency, our results (DRKPV and DRPMMR) establish uniform convergence. Depending on regression smoothness, this can yield faster rates than both outcome- and treatment-bridge methods.
> *   **Empirical performance**: As shown in Section 5, DRKPV and DRPMMR consistently outperform prior methods across benchmarks.
> > **Weaknesses**: If data splitting is applied, sample inefficiency might be a concern. If reuse is employed, what are the implications for theoretical guarantees (e.g., bias or overfitting)? This trade-off requires further discussion.
>
> In the camera-ready version, we will include a detailed discussion of sample splitting in Appendix F. Our estimator uses the same splitting setup as KPV and KAP:
> *    KPV: $n_h$ and $m_h$ for its first- and second-stage, respectively,
> *    KAP: $n_\varphi$, $m_\varphi$, and $t_\varphi$ for its first-, second-, and third-stage, respectively.
>
> Using the full dataset in each stage also preserves consistency (see Corollary 1 in [11] for an outcome bridge–based method). Our consistency proofs in Appendix E never require disjoint sample splits; under Theorem E.27 andE.28 with optimal regularization, our methods converge to the true causal structural function. Splitting is nevertheless useful in practice for two reasons:
> *    Regularization tuning: In KAP, the second-stage regularizer (no closed-form LOOCV) requires held-out data. We use first-stage data for validation and tuning.
> *    Different observation types:For example, KPV’s first stage uses $\{A, W, Z\}$, while the second-stage uses $\{A, Y, Z\}$. Splitting thus enables estimation when only partial marginals are available.
> > **Weaknesses and Questions**: ... the paper does not analyze the asymptotic distribution of the proposed estimator. Will the proposed estimators also inherit local efficiency, or at least satisfy asymptotic normality? Given the nonregularity of continuous treatment estimators, should we expect slower-than-$\sqrt{n}$ asymptotic normal.
>
> While Theorem 2.7 provides an identification formula derived from the efficient influence function (EIF), and our estimator resembles a one-step correction, we emphasize that it is not a classical one-step estimator and therefore does not automatically inherit local efficiency. Unlike standard EIF-based estimators that leverage first-order orthogonality between two coupled nuisance components (e.g., outcome regression and propensity score), our method involves **three components**—the outcome bridge, treatment bridge, and a correction term—each estimated separately. As a result, the bias terms accumulate additively, rather than canceling multiplicatively, as in traditional doubly robust estimators. Establishing local efficiency would thus require further analysis, including proving asymptotic linearity, implementing cross-fitting, and verifying attainment of the semiparametric variance bound—steps we leave for future work.
>
> Similarly, asymptotic normality is not guaranteed. Although the estimator is consistent and based on an EIF-derived formula, a central limit theorem would require controlling higher-order remainder terms across all three nuisance components—particularly challenging in our kernel-based framework. We therefore refrain from claiming asymptotic normality in the present version, though we believe it could be established in future works.
>
> Despite this, our estimators offer several practical advantages: they yield closed-form solutions, support uniform consistency guarantees, exhibit double robustness to outcome or treatment bridge misspecification, and handle continuous or high-dimensional treatments more naturally than previous methods. Empirically, they consistently outperform existing baselines.
>
> Finally, we agree with the reviewer that in the continuous treatment setting, one should generally expect slower-than-$\sqrt{n}$ convergence due to the nonregularity of the parameter—a point supported by both theory and our convergence results.
>
> We thank the reviewer for these questions and we will subsequently add a discussion paragraph in the paper.
> > **Questions**: ... computational efficiency of the proposed approach?
>
> The main computational bottleneck is kernel matrix inversion, so—as with other kernel ridge regression–based methods—our estimators scale cubically in the sample size ($O(t^3)$). As detailed in our response to Reviewer tteA (see Complexity and Scalability), both DRKPV and DRPMMR fall into this regime; we will expand this discussion and include breakdowns in Appendix F. To mitigate scalability, we applied the Nyström approximation [R1] across all DRKPV kernel regressions (KPV first/second, KAP first/third, and slack prediction), achieving substantial runtime reductions while maintaining accuracy. Detailed results are provided in our response to Reviewer tteA, and we will add these ablations in Appendix F. We plan to explore stronger approximations such as scalable kernel ridge regression [R2] and neural mean embedding methods [R3].
>
> [R1] Williams, C. K. I. and M. Seeger (2001). “Using the Nyström Method to Speed Up Kernel Machines”. NeurIPS 13.
>
> [R2] Meanti, G., L. Carratino, L. Rosasco, and A. Rudi (2020). “Kernel methods through the roof: handling billions of points efficiently”. In: NeurIPS 34.
>
> [R3] Liyuan Xu and Heishiro Kanagawa and Arthur Gretton, (2021). "Deep Proxy Causal Learning and its Application to Confounded Bandit Policy Evaluation". In NeurIPS 35.

---

> > ### Comment · Reviewer_j1Ba · 2025-08-05
> > **Response**
> >
> > Thanks for your clarification. Most of my concerns are addressed. I will increase my score.

---

> > > ### Author Response · Authors · 2025-08-06
> > >
> > > Thank you for your feedback. We are glad to hear that our responses have addressed your concerns. We truly appreciate your time and engagement.

---

### Official Review · Reviewer_tteA · 2025-07-03

**Clarity:** 3
**Significance:** 3
**Originality:** 3
**Rating:** 4
**Confidence:** 3

**Summary:**

When hidden factors influence a treatment's outcome, researchers can use proxy variables to estimate the true causal effect. However, standard methods require estimating complex density ratios, which can be unstable.
The authors developed a doubly robust method that avoids this step, leading to more stable and effective causal effect estimation with proxy variables.

Specifically, they introduces a density ratio-free, doubly robust kernel-based estimator for treatment effect estimation in the presence of unmeasured confounding, leveraging proxy variables.

- Their estimator does not rely on estimating density ratios, improving robustness and practicality.
- They employ RKHS, supporting nonparametric inference.
- Through synthetic and real-world datasets, the proposed method outperforms standard and proximal baselines, particularly in misspecified or high-dimensional scenarios.

**Questions:**

See my questions in the above “Weakness” part and below:

- How does the computational cost of the proposed kernel method scale with the number of samples and proxies?
- How sensitive is the estimator’s performance to the choice of kernel function and its parameters?

**Ethical Concerns:**

["NO or VERY MINOR ethics concerns only"]

**Final Justification:**

My main concern was satisfactorily addressed. I remain my overall assessment.

**Limitations:**

See the limitations also in the above “Weakness” and “Questions” part.

**Paper Formatting Concerns:**

Figures 2, 3 may be difficult to read when printed or viewed on small screens.

**Quality:**

3

**Strengths And Weaknesses:**

Assumptions 2.3 and 2.5 require that the proxies are "sufficiently informative" relative to the unobserved confounder, which seems resasonable to me. Though in practice it might be hard to verify.

**Strengths:**
- They developed a density ratio-free approach, which is more scalable and stable for continuous or high-dimensional treatments.
- Theorem 2.7 shows that the proposed estimators are designed to be consistent if either the outcome bridge model or the treatment bridge model is correctly specified.
- They also demonstrate Theorem 2.7 empirically through misspecification experiments, where the estimators recover the true causal effect even when one of the bridge functions is intentionally perturbed.
- The paper establishes uniform consistency for both proposed estimators (Theorems 4.1 and 4.2).

**Weakness:**
- The computational cost of kernel methods may make the method scale poorly with the number of samples. Thus might be impractical for large datasets.
- RKHS methods typically rely on kernel selection and tuning. The paper does not explore how robust the method is to kernel and hyperparameter choices.
- The DRKPV estimator, which builds on the KPV and KAP algorithms, may use data splitting for its different regression stages, but seems less discussed? Or did I missed it?

---

> ### Author Rebuttal · Authors · 2025-07-31
>
> We thank you for your detailed and constructive feedback. Below, we address your concerns point by point. If any issues remain unclear, we are glad to provide additional clarification. Should our responses resolve your concerns, we would appreciate it if you would consider adjusting your score.
> > Assumptions 2.3 and 2.5 require that the proxies are "sufficiently informative" ... Though in practice it might be hard to verify.
>
> Completeness is a ubiquitous tool in statistics and has recently been employed in causal learning to establish identification. The challenge of verifying the completeness assumption is not unique to our work but is shared across proxy causal learning (PCL) and instrumental variable (IV) regression. Appendix Section 2 of [R1] discusses completeness in the context of PCL and explicitly builds on earlier foundational work such as [R2] and [R3], and many more. These references show how completeness has been formulated and justified in general statistical inference and IV problems, and [R1] uses them to argue that the completeness assumption in PCL can be similarly framed.
>
> The general takeaway, as emphasized in [R1], is that completeness holds for a wide range of semiparametric and nonparametric models, although it is typically not testable—even when all relevant variables are observed [R2]. In addition, [R1] provides concrete examples where completeness is satisfied. For instance, [R3] shows that when proxies and confounders are continuously distributed and the proxy dimension exceeds the confounder dimension, completeness is ensured under mild regularity conditions. Thus, in PCL it is standard to assume completeness for a broad class of models. Intuitively, completeness formalizes the requirement that proxies be sufficiently informative about the confounders, highlighting the importance of collecting rich sets of proxy variables in observational studies to mitigate unobserved confounding.
>
> We will incorporate an extended discussion in our updated manuscript to explicitly address this point and strengthen clarity.
>
> [R1] Miao, W., Hu, W., Ogburn, E. L., and Zhou, X.-H. (2022), “Identifying Effects of Multiple Treatments in the Presence of Unmeasured Confounding,” Journal of the American Statistical Association.
>
> [R2] Canay, I. A., Santos, A., and Shaikh, A. M. (2013) “On the testability of identification in some nonparametric models with endogeneity.” Econometrica.
>
> [R3] Andrews, D. W. (2017) “Examples of L2-complete and boundedly-complete distributions.” Journal of Econometrics.
> > **Weaknesses and Questions**: computational cost of the proposed kernel method and its scalability to larger sets"
>
> We will add a detailed discussion in Appendix F about the computational complexity and scalability of our methods. Briefly, our method builds on multi- and single-stage kernel regressions from KPV/PMMR and KAP, each of which involves kernel matrix inversions. This results in cubic complexity with respect to the sample size. Specifically, KAP stages have complexities: (i) first-stage $O(n_\varphi^3)$, (ii) second-stage $O((m_\varphi + 1)^3)$, (iii) third-stage $O(t_\varphi^3)$. Similarly, KPV stages have complexities: (i) $O(n_h^3)$, (ii) second-stage $O(m_h^3)$. Here, $n_\varphi, m_\varphi, t_\varphi, n_h, m_h$ denote the number of samples used in different stages of KAP and KPV algorithms. Our estimator also requires computing the slack prediction (Appendix C.3), which adds a $O(t^3)$ cost, where $t$ is the sample size of the whole dataset. Overall, the complexity of our proposed method is $O(t^3)$, dominated by the largest kernel inversion.
>
> To address scalability, we provide an additional set of numerical experiments where we incorporate the Nyström approximation [R4] in all kernel regressions of DRKPV (KPV first/second stage, KAP first/third stage, slack term estimation). The second-stage regression of KAP involves a more specialized setup that differs from standard kernel ridge regression, but it is equally amenable to approximation techniques. Here, we demonstrate substantial scalability gains by applying Nyström to the kernel regression stages; extending it to the KAP second stage is feasible and will be included in future versions. We present synthetic low-dimensional experiments (Section 5) comparing DRKPV with Nyström (500/100 landmarks) to the full kernel version, where each regularization parameter is set to $10^{-3}$. Results (to be added in Appendix F) show Nyström makes DRKPV far more scalable. Future work will also explore stronger approximations such as scalable kernel ridge regression [R5] and neural mean embedding [R6], including for the second-stage regression of KAP. In this paper, we focus on providing a robust and uniformly consistent kernel method, leaving scalable variants for future work.
>
> |Algorithm| Data Size|MSE (mean±std)|Algo Run Time (mean±std)|
> |-|-|-|-|
> |DRKPV (original)| 5000|0.0097±0.0046|61.1±0.21|
> |DRKPV (Nyström, 500 landmarks)|5000|0.0082±0.0033|42.5±0.19|
> |DRKPV (Nyström, 100 landmarks)|5000|0.0071±0.0028|39.2±0.22|
> |DRKPV (original)|7500|0.0070±0.0033|194.5±0.35|
> |DRKPV (Nyström, 500 landmarks)|7500|0.0085±0.0044|132.7±0.04|
> |DRKPV (Nyström, 100 landmarks)|7500|0.0091±0.0046|127.6±0.00|
> |DRKPV (original)|10000 |0.0065±0.0026|453.9±0.51|
> |DRKPV (Nyström, 500 landmarks)|10000|0.0059±0.0014|306.7±0.02|
> |DRKPV (Nyström, 100 landmarks)|10000|0.0062±0.0022|299.5±0.00|
>
> [R4] Williams, C. K. I. and M. Seeger (2001). “Using the Nyström Method to Speed Up Kernel Machines”. NeurIPS 13.
>
> [R5] Meanti, G., L. Carratino, L. Rosasco, and A. Rudi (2020). “Kernel methods through the roof: handling billions of points efficiently”. NeurIPS 34.
>
> [R6] Liyuan Xu and Heishiro Kanagawa and Arthur Gretton, (2021). "Deep Proxy Causal Learning and its Application to Confounded Bandit Policy Evaluation". NeurIPS 35.
> > **Weaknesses and Questions**: "sensitivity of the estimator’s performance to the choice of kernel function and its parameters"
>
> To evaluate robustness, we conducted further experiments across (i) different kernel length scales (quantiles of pairwise distances) and (ii) different Matern kernels (with varying smoothness parameter $p$).  For Gaussian kernels, our main results used the common median heuristic (0.5 quantile). Tables below report DRKPV and DRPMMR results on synthetic low-dimensional and legalized abortion/crime datasets. We will add these ablation studies in Section F.
> *   Across quantiles, performance remains stable for a broad range, with degradation only at extreme settings (e.g., 0.9 quantile).
> *   Across Matern kernels, our methods maintain robust performance for varying $p$.
>
> Table illustrating performance over different length-scale selections of the RBF kernel in the synthetic low-dimensional dataset.
> | Algorithm |$l$-quantile | MSE ± Std|
> |-|-|-|
> |DRKPV|0.25|0.055±0.035|
> |DRKPV|0.40|0.028±0.016|
> |DRKPV|0.50|0.026±0.015|
> |DRKPV|0.60|0.027±0.018|
> |DRKPV|0.75|0.034±0.019|
> |DRKPV|0.90|0.22±0.096|
> |DRPMMR|0.25|0.026±0.011|
> |DRPMMR|0.40|0.020±0.011|
> |DRPMMR|0.50|0.019±0.012|
> |DRPMMR|0.60|0.024±0.018|
> |DRPMMR|0.75|0.034±0.033|
> |DRPMMR|0.90|0.22±0.130|
>
> Table illustrating performance over different length-scale selections of the RBF kernel in Legalized Abortion and Crime dataset.
> |Algorithm|$l$-quantile|MSE ± Std|
> |-|-|-|
> |DRKPV|0.25|0.024±0.007|
> |DRKPV|0.40|0.020±0.014|
> |DRKPV|0.50|0.018±0.010|
> |DRKPV|0.60|0.019±0.012|
> |DRKPV|0.75|0.023±0.016|
> |DRKPV|0.90|0.021±0.018|
> |DRPMMR|0.25|0.016±0.005|
> |DRPMMR|0.40|0.015±0.008|
> |DRPMMR|0.50|0.016±0.009|
> |DRPMMR|0.60|0.018±0.009|
> |DRPMMR|0.75|0.025±0.013|
> |DRPMMR|0.90|0.022±0.015|
>
> Table illustrating performance over different Matern kernel selections in synthetic low dimensional dataset.
> | Algorithm | $p$| MSE ± Std|
> |-|-|-|
> |DRKPV|0|0.032±0.016|
> |DRKPV|1|0.022±0.013|
> |DRKPV|2|0.022±0.013|
> |DRKPV|3|0.022±0.013|
> |DRKPV|10|0.024±0.015|
> |DRPMMR|0|0.029±0.011|
> |DRPMMR|1|0.019±0.013|
> |DRPMMR|2|0.019±0.013|
> |DRPMMR|3|0.018±0.013|
> |DRPMMR|10|0.022±0.016|
>
> Table illustrating performance over different Matern kernel selections in Legalized Abortion and Crime dataset.
> |Algorithm|$p$|MSE ± Std|
> |-|-|-|
> |DRKPV|0|0.019±0.007|
> |DRKPV|1|0.022±0.015|
> |DRKPV|2|0.020±0.012|
> |DRKPV|3|0.023±0.017|
> |DRKPV|10|0.023±0.014|
> |DRPMMR|0|0.016±0.003|
> |DRPMMR|1|0.019±0.009|
> |DRPMMR|2|0.018±0.010|
> |DRPMMR|3|0.018±0.010|
> |DRPMMR|10|0.021±0.011|
> > **Weaknesses**: The DRKPV estimator, ..., may use data splitting for its different regression stages, but seems less discussed? Or did I miss it?
>
> We agree that adding more discussion would help readers better understand the implementation, and we will devote a subsection in Appendix F to this. Specifically, our estimator uses the same data splitting setup as KPV/KAP. For a dataset of $t$ samples $\{a_i, y_i, z_i, w_i\}_{i= 1}^t$, the methods split data across stages:
> *    KPV: $n_h$ and $m_h$ for its first- and second-stage, respectively,
> *    KAP: $n_\varphi$, $m_\varphi$, and $t_\varphi$ for its first-, second-, and third-stage, respectively.
>
> Using the full dataset per stage still retains consistency (see Corollary 1 in [11] for outcome bridge-based method). Our consistency proof (Appendix E) does not assume disjoint splits, nor do [10] and [19]. However, data splitting becomes useful in practice due to two reasons:
> *    For KAP, tuning the second-stage regularizer (no closed-form LOOCV) requires held-out data. In implementation, we utilize first-stage data as a held-out set to evaluate the validation loss and tune the regularizer in second-stage regression.
> *    Splitting accommodates different observation types: e.g., KPV's first-stage uses $\{A, W, Z\}$, while the third-stage uses $\{A, Y, Z\}$, enabling estimation when only partial marginals are available rather than the full joint distribution.
> > **Paper Formatting Concerns**: Figures 2, 3 may be difficult to read when printed or viewed on small screens.
>
> We will enlarge the font sizes in the figures to ensure readability.

---

### Decision · Program_Chairs · 2025-09-17

**Decision:**

Accept (poster)

**Comment:**

This elegant paper proposed two doubly-robust kernel-based estimators for causal inference in the presence of unobserved confounders.  It builds on two different proxy approaches, when proxies are available for unobserved confounders, focusing on either the influence on treatment (propensity) or influence on outcome.  The paper does a nice job of reviewing the relevant recent literature, situating its contributions in the recent literature, and showing its advantages and limitations empirically and theoretically.

The paper should be proofread again carefully; for example, on line 159 the actual heading for section 2.3 says "Doubly bobust..."

The reviewers and authors had a robust discussion, so the authors should also please incorporate items from that discussion to improve clarity of the paper.